# A fast and simplified subglacial hydrological model for the Antarctic Ice Sheet and outlet glaciers

Elise Kazmierczak[1,*], Thomas Gregov[1,2,*], Violaine Coulon[1], and Frank Pattyn[1]

[1]Laboratoire de Glaciologie, Université libre de Bruxelles (ULB), Brussels, Belgium
[2]Aérospatiale et Mécanique, Université de Liège (ULiège), Liège, Belgium
[*]These authors contributed equally to this work.

**Correspondence:** Elise Kazmierczak (elise.kazmierczak@ulb.be)

**Abstract.** We present a novel and computationally efficient subglacial hydrological model that represents in a simplified way both hard and soft bed rheologies as well as an automatic switch between efficient and inefficient subglacial discharge, designed for the Antarctic ice sheet. The subglacial model is dynamically linked to a regularized Coulomb friction law, allowing for a coupled evolution of the ice sheet on decadal to centennial time scales. It does not explicitly simulate the details of water conduits at the local scale and assumes that subglacial hydrology is in quasi-static equilibrium with the ice sheet, which makes the computations very fast. The hydrological model is tested on an idealized marine ice sheet and subsequently applied to the drainage basin of Thwaites Glacier, West Antarctica, that is composed of a heterogeneous (hard/soft) bed. We find that accounting for subglacial hydrology in the sliding law accelerates the grounding-line retreat of Thwaites Glacier under present-day climatic conditions. Highest retreat rates are obtained for hard bed configurations and/or inefficient drainage systems. We show that the sensitivity is particularly driven by large gradients in effective pressure, more so than the value of effective pressure itself, in the vicinity of the grounding line. Therefore, we advocate for a better understanding of the subglacial system with respect to both the spatial and temporal variability in effective pressure and the rheological conditions/properties of the bed.

## 1 Introduction

Due to the ubiquitous and permanent ice cover, subglacial environments in Antarctica are hard to observe. The lack of direct observations results in major uncertainties in ice-dynamical behavior, especially since the ice-bed interface is one of the main boundary layers that influence the overall dynamics of the ice sheet. In ice-sheet models, basal processes are generally translated in a basal sliding law, that, for the sake of simplicity, is largely parameterized. However, recent studies have shown that the level of plasticity of basal sliding, which mainly depends on the bed rheology, is a highly controlling factor in terms of mass change of the Antarctic Ice Sheet (Brondex et al., 2019; Bulthuis et al., 2019; Sun et al., 2020; Kazmierczak et al., 2022). The rheology of the bed broadly covers two categories, i.e., (i) hard beds, composed of bedrock and whose rigidity is greater than the ice, and also considered as non-deformable and (ii) soft beds, or a subglacial till cover, whose rigidity is lower than the ice, and which can easily be deformed (Muto et al., 2019). The rheology is further influenced by the presence of subglacial water, which is present underneath more than 50% of the ice sheet due to the thick ice cover (Robin et al., 1968; Smith et al., 2009;

Pattyn, 2010). The spatial organization of the subglacial drainage system remains poorly known, although recent attempts led to improvements regarding the modeling of subglacial water flow (Livingstone et al., 2013; Willis et al., 2016; Dow et al., 2022; Dow, 2022a; Hager et al., 2022).

The nature of drainage systems and their tendency to organize into channels depend on subglacial water flow as well as bed rheology. The morphology of subglacial water drainage systems influences basal sliding by modifying the basal boundary

conditions (Hewitt, 2011). Furthermore, within drainage systems, water flow and water pressure also vary greatly. Physically, the presence of subglacial water directly influences basal sliding by lubricating the bedrock or by weakening the till strength. Budd et al. (1979) proposed to link subglacial water to basal sliding through the effective pressure (which is the ice overburden pressure, i.e., the downward pressure due to the weight of overlying ice, minus the subglacial water pressure). The link between basal sliding and subglacial hydrology through the effective pressure is used in most glacier and ice sheet studies (Pattyn,

1996; Hoffman and Price, 2014; Beyer et al., 2018; Gagliardini and Werder, 2018). Other approaches are the use of subglacial water thickness (Weertman and Birchfield, 1982; Budd and Jenssen, 1987; Alley et al., 1989; Johnson and Fastook, 2002) or subglacial water flux (Pattyn et al., 2005; Goeller et al., 2013) as a control on basal sliding. In this study, we consider basal sliding a function of effective pressure at the base of the ice sheet.

Generally, bed areas characterized by a low effective pressure correspond to regions of faster ice flow (Iken and Bindschadler,

1986; Iverson et al., 1999) and consequently may become more sensitive (react much faster and/or exhibit more ice change) to climate forcing (Kazmierczak et al., 2022). However, the effective pressure at the base of the ice sheet is difficult to determine as it depends on bed rheology, the presence and distribution of subglacial water, the distribution of subglacial till and its thickness, etc. Furthermore, subglacial processes occur on time scales that can be as small as a few hours (e.g., Clarke, 2005). Such time scales are several orders of magnitude smaller than the typical response times of glaciers and ice sheets, which are at

least of the order of years. This discrepancy hampers numerical coupling between subglacial hydrology and the ice dynamical response. Another limiting factor is the spatial resolution on which these processes occur, hence the computational demand it entails. For instance, state-of-the-art hydrological models such as GlaDS (Werder et al., 2013) are —due to their high spatial and temporal resolution— extremely computationally demanding, and their applications to evolving ice sheets are often limited to the initialization of the ice-sheet system (e.g., McArthur et al., 2023; Pelle et al., 2023). Although there have been recent

attempts to reduce their computational cost using data-driven methods (Verjans and Robel, 2024), physics-based approaches have not yet been widely explored to achieve this goal.

In this paper, we propose a simplified model of the complex subglacial system that allows us to dynamically link subglacial hydrology to basal sliding for various bed types (hard and soft). The model considers different spatially- and temporally-varying subglacial water drainage systems. Their morphologies depend both on the subglacial water flux (distributed or channelized)

and the rheology of the bed. The approach allows us to evaluate the impact of the subglacial hydrological system and its evolution on the ice-sheet response in a hard and a soft bed environment in large-scale models and with reasonable computational times. By large-scale models, we mean models that have a spatial grid size that is orders of magnitude larger than that of the hydrological system. The model also allows us to deal with mixed beds, composed of zones of different rigidity.

In section 2, we describe the hydrological model and its implementation. We first evaluate our model for an idealized marine ice sheet in Section 3, to evaluate the implementation and robustness of the method. Subsequently, we apply our methodology to Thwaites Glacier, West Antarctica (Section 4), generally characterized by a heterogeneous bed composed of soft and hard bed zones (Joughin et al., 2009; Schroeder et al., 2014; Muto et al., 2019). We discuss the impact of the dynamic subglacial hydrology linked to basal sliding as well as the limitations of the model in Section 5. Finally, we conclude on the relevance of our findings in Section 6.

## 2 Model description

### 2.1 Ice-flow model

We employ the Kori-ULB ice-sheet model (previously called f.ETISh; Pattyn, 2017; Seroussi et al., 2019, 2020; Sun et al., 2020; Coulon et al., 2024), which is a vertically integrated thermomechanically-coupled, hybrid ice sheet/ice shelf model. It combines the shallow-ice approximation for ice deformation with the shallow-shelf approximation for basal sliding in a similar way as in Winkelmann et al. (2011). To account for sliding on both a hard and soft bed (Cuffey and Paterson, 2010; Beaud et al., 2022), we employ a regularized Coulomb friction law (Joughin et al., 2019; Zoet and Iverson, 2020). Such law allows for a smooth transition between a power-law behavior at low velocity and a plastic Coulomb behavior at high velocity, and takes the following form:

$$\boldsymbol{\tau}_{\mathrm{b}} = CN \left( \frac{\|\boldsymbol{v}_{\mathrm{b}}\|}{\|\boldsymbol{v}_{\mathrm{b}}\| + v_0} \right)^{\frac{1}{m}} \frac{\boldsymbol{v}_{\mathrm{b}}}{\|\boldsymbol{v}_{\mathrm{b}}\|}, \tag{1}$$

where $\boldsymbol{\tau}_{\mathrm{b}}$ is the basal shear or friction stress, $N$ is the effective pressure, $C$ is a friction coefficient limiting the shear stress to a maximum plastic value $CN$, and $\boldsymbol{v}_{\mathrm{b}}$ is the basal sliding velocity. The velocity threshold $v_0$ is a parameter that controls the onset of plasticity in the friction law. A value of $m = 3$ and $v_0 = 300\,\mathrm{m\,a^{-1}}$ are taken as in Joughin et al. (2019). A complete list of symbols, and their corresponding values and units, can be found in Appendix A.

### 2.2 Hydrological model

Subglacial water in Antarctica essentially stems from basal melting of areas of the ice sheet that are at the pressure melting point (Pattyn et al., 2005; Pattyn, 2010; Beyer et al., 2018; Dow et al., 2022), due to dissipative phenomena at the ice-bedrock interface. It is absent when the basal ice is at a temperature below the pressure melting point (Pattyn, 2010; Livingstone et al., 2013). Limited surface water infiltration towards the bed has been observed in Antarctica, contrary to the Greenland ice sheet (Bell et al., 2018). The presence of subglacial water will lead to a reduction of the contact between the ice and the subglacial substrate, i.e., it will decrease the value of the effective pressure $N = p_{\mathrm{o}} - p_{\mathrm{w}}$, with $p_{\mathrm{o}}$ the ice overburden pressure and $p_{\mathrm{w}}$ the subglacial water pressure. It is generally assumed that the ice overburden pressure is cryostatic, so that $p_{\mathrm{o}} = \rho_{\mathrm{i}} g h$ where $\rho_{\mathrm{i}}$ is the ice density, $g$ is the gravitational acceleration, and $h$ is the ice thickness. For a given ice-sheet geometry, the effective pressure is therefore fully characterized by the subglacial water pressure.

Alternatively, one can consider a description based on potentials. Introducing the hydraulic potential $\phi = \rho_\mathrm{w} g b + p_\mathrm{w}$ and the geometric potential $\phi_0 = \rho_\mathrm{i} g h + \rho_\mathrm{w} g b$, $\rho_\mathrm{w}$ being the density of fresh water and $b$ the bedrock elevation with respect to the local sea-level height, the effective pressure is then the difference between both, i.e., $N = \phi_0 - \phi$. The rationale behind this description is that water flows from regions where the hydraulic potential is high to regions where the potential is low (Shreve, 1972).

In this paper, we attempt to model together efficient and inefficient water flow over both hard and soft beds. Generally, efficient systems transport large water fluxes and are characterized by localized channelized flow, while inefficient systems take the form of distributed water flow. We define two spatial scales: a global scale, which is the same as the one used for the ice-sheet model and that is typically of the order of kilometers, and a local scale, associated with a water conduit, and that is much smaller than the global one (observations suggest that channels are meters to at most a few hundreds meters wide, that maximal width being reached close to the grounding line (Drews et al., 2017)). The computation of the water flow is done at the global scale, while the computation of the effective pressure is done at the local scale. For the latter, we consider a single element of the hydrological system, which we refer generically to as 'conduit'. The term conduit is used for localized drainage systems (such as cavities, channels, canals), as opposed to diffuse drainage systems (such as continuous films, diffuse water flow within sediments). This applies to efficient flow (channel or canal), or to elements of an inefficient system (cavity or patchy film between clasts), applicable to both a hard or a soft bed. In particular, we do not use 'conduit' as a synonym for 'channel', as a conduit can correspond to other types of hydrological elements. Such approach can be seen as a sub-grid parametrization of the effective pressure; a similar procedure was described in Gowan et al. (2023). The idea of unifying both inefficient and efficient water flow has been previously explored in Arnold and Sharp (2002), in Schoof (2010), and in Sommers et al. (2018). Our approach is shown schematically in Figure 1 and described in detail in the next subsections.

### 2.2.1 Simplifying assumptions

Our model is constructed from multiple approximations that simplify the physics describing subglacial hydrology. It differs from recent studies aiming to develop high-resolution models, such as GlaDS (Werder et al., 2013), SHAKTI (Sommers et al., 2018), CUAS (Beyer et al., 2018), and the subglacial hydrology components within PISM (Bueler and van Pelt, 2015) and MALI (Hoffman et al., 2018). These models typically take the form of a series of coupled partial differential equations that are computationally challenging to solve. Furthermore, these models involve a large number of parameters, many being poorly constrained. Finally, due to their high spatial and temporal resolution they are often computationally demanding. The latter may limit their application to drainage basins or single glaciers on time scales of a few years. By contrast, our model is computationally cheap, with the computational time associated with the subglacial hydrology calculation representing only a small fraction of the computational time associated with the ice-sheet model. This allows us to study the impact of subglacial hydrology on ice dynamics on a large scale and at a limited computational cost, while at the same time keeping the essential features of complex subglacial hydrology models.

The key simplifying assumptions are given by the following:

1. There is limited temporal melt variability so that the hydrological system is in a quasi-static equilibrium with respect to the ice-sheet geometry. Therefore, changes in ice geometry will be the main driver for changes in subglacial water variability (both spatial and temporal).

2. A few kilometers upstream of the grounding line, the hydraulic gradient is approximated by the geometrical gradient.

3. The drainage density, that is, the number of conduits per grid cell, is uniform.

4. The effective-pressure distribution is not calculated at the sub-grid (local) level.

The first assumption is based on several studies of subglacial hydrology in Antarctica (Le Brocq et al., 2009; Pattyn, 2010; Kazmierczak et al., 2022), among others, that demonstrate that —contrary to the Greenland ice sheet— there is limited surface meltwater infiltration. Hence, changes in hydrology are primarily due to changes in ice geometry. Since the time scales associated with water flow are much smaller than those associated with ice flow, subglacial hydrology automatically adapts to any change in ice geometry and reaches the associated equilibrium. The second assumption is motivated by a scaling analysis through an estimation of the dimensionless ratio $\eta := [\nabla N]/[\nabla \phi_0]$, where $[\nabla N]$ is the scale of the spatial gradients for the effective pressure and $[\nabla \phi_0]$ is the characteristic scale for the geometric potential gradient. For the former we take $[\nabla N] = [N]/[x]$, with $[N] = 1\,\mathrm{MPa}$ and $[x] = 10^3\,\mathrm{km}$. For the latter we take $[\nabla \phi_0] = 5 \times 10^{-2}\,\mathrm{MPa\,km^{-1}}$, which is a plausible value for ice sheets (Hewitt, 2011). This results in $\eta = 2 \times 10^{-2} \ll 1$, suggesting that $\|\nabla N\| \ll \|\nabla \phi_0\|$ and $\nabla \phi \approx \nabla \phi_0$. We further note that profiles obtained with a high-resolution subglacial hydrology model suggest that $\nabla \phi$ and $\nabla \phi_0$ have a correlation of at least $\sim 80\%$ for a region that is several kilometers upstream of the grounding line (see Supplementary Material S1). Finally, the third and fourth assumptions follow from our modeling approach, where we do not describe the effective pressure at the sub-grid scale and where we assume the same number of conduits in each grid cell, similar to Gowan et al. (2023). However, the effective pressure within a channel may well differ from its value away from the channel, which is something that is not taken into account. Consequently, these last assumptions are the most likely to be debatable.

### 2.2.2 Subglacial water routing

Let us denote by $\Omega$ the grounded ice sheet domain where subglacial water is routed. This domain evolves over time according to both internal conditions (e.g., changes in ice velocity) and external conditions (e.g., changes in sub-shelf melt). Its boundary is partitioned into non-overlapping subsets $\Gamma_\mathrm{d}$ and $\Gamma_\mathrm{gl}$, representing the divides of the considered basin and the grounding line, respectively. The subglacial water flux, $\boldsymbol{q}_\mathrm{w}$ (m$^2$ s$^{-1}$), is determined from a mass balance equation that takes the form of a steady-state continuity equation:

$$
\begin{cases}
\nabla \cdot \boldsymbol{q}_\mathrm{w} = \dfrac{\dot{m}}{\rho_\mathrm{w}}, & \text{in } \Omega, \\[2mm]
\boldsymbol{q}_\mathrm{w} \cdot \boldsymbol{n} = 0, & \text{on } \Gamma_\mathrm{d},
\end{cases}
$$

(2a)

(2b)

in which $\nabla$ is the horizontal spatial gradient, $\boldsymbol{n}$ is the outward normal to the boundary $\Gamma_\mathrm{d}$, and $\dot{m}$ is the melt rate (kg m$^{-2}$s$^{-1}$). The latter is computed from the energy balance within the ice sheet and includes effects of geothermal heat flux, frictional

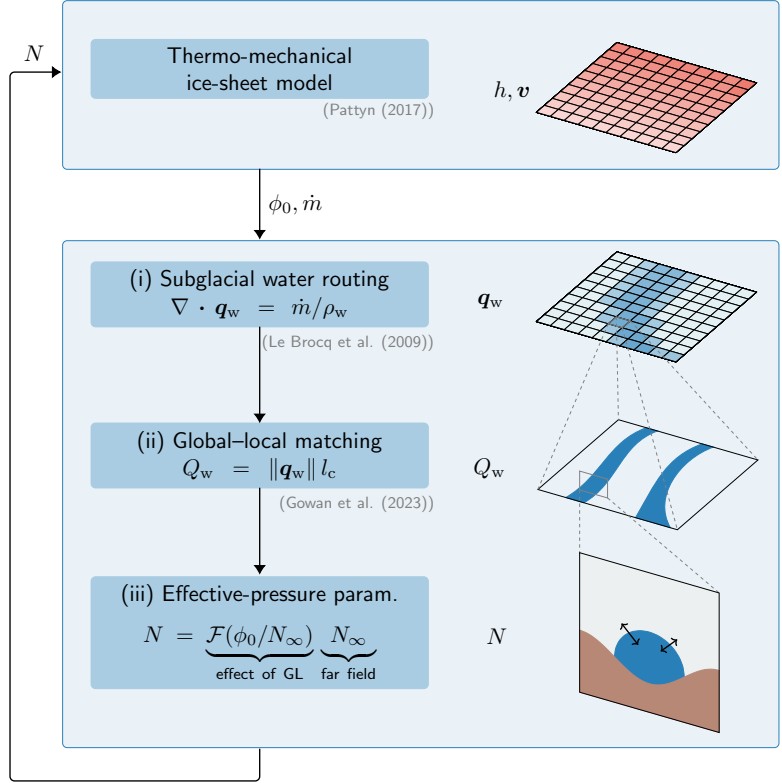

**Figure 1.** Flowchart of the dynamical linkage between the ice sheet and the subglacial hydrology. At each time step, the ice-sheet model provides the basal melt rate $\dot{m}$ and the geometrical potential $\phi_0$. Based on these, the effective pressure is computed in three steps: (i) The globally distributed subglacial water flux $\boldsymbol{q}_w$ is computed according to Le Brocq et al. (2009); (ii) a connection between both global and local (conduit) scale is obtained by specifying the distance $l_c$ between the conduits (Gowan et al., 2023), yielding a volumetric water flux $Q_w$ in each conduit; (iii) the effective pressure $N$ is computed for each conduit via a parametrization where $\mathcal{F}(\phi_0/N_\infty) = \mathrm{erf}[(\sqrt{\pi}/2)\phi_0/N_\infty]$ serves as a correction factor for the impact of the grounding line (GL), and where $N_\infty$ is the effective pressure far upstream of the grounding line. This effective pressure is then used by the large-scale ice-sheet model and is the same for all conduits that belong to the same grid cell.

heating due to the motion of both ice and subglacial water, and thermal conduction, i.e.,

$$\dot{m} = \frac{G + \boldsymbol{\tau}_{\mathrm{b}} \cdot \boldsymbol{v}_{\mathrm{b}} - q_T}{\mathcal{L}_{\mathrm{w}}} + \dot{m}_{\mathrm{w}}, \tag{3}$$

where $G$ is the geothermal heat flux, $q_T$ is the thermal conduction flux, $\mathcal{L}_{\mathrm{w}}$ is the latent heat for ice, and $\dot{m}_{\mathrm{w}} = |\boldsymbol{q}_{\mathrm{w}} \cdot \nabla \phi| / \mathcal{L}_{\mathrm{w}}$ is the water melt rate due to the dissipated energy from the subglacial water conduits. However, we do not include this last term in the computation of the subglacial water in our simulations as it was found to be negligible compared to the other terms. Note that by writing the system of equations (2), we have assumed that the subglacial water system is in steady state with respect to a given ice-sheet geometry. As previously mentioned, we justify this assumption by the observation that the time scales associated with subglacial water flow are much smaller than the ones associated with ice flow. By construction, this inhibits oscillations in the coupled system formed by the ice sheet and the subglacial hydrology, which are known to exist for ice streams on time scales of thousands of years (Robel et al., 2013), well beyond the time scales considered here.

Equation (2) is solved at the global scale, using the water routing of Le Brocq et al. (2009) to compute the water flux based on the geometric potential gradient $\nabla \phi_0$. As the subglacial water flux $\boldsymbol{q}_{\mathrm{w}}$ is in fact proportional to $\nabla \phi$ instead of $\nabla \phi_0$, we should use the potential gradient $\nabla \phi$ itself. Nonetheless, in view of the assumption that $\nabla \phi \approx \nabla \phi_0$ over most of the domain, we choose not to do so as this allows us to decouple the water routing solver from the effective-pressure calculation.

### 2.2.3 Subglacial effective pressure

The distributed water flux $\boldsymbol{q}_{\mathrm{w}}$ needs to be converted to the local volumetric water flux $Q_{\mathrm{w}}$ ($\mathrm{m}^3\,\mathrm{s}^{-1}$) within the water conduits. Since the distance between the conduits is given by $l_{\mathrm{c}}$, it follows that there are $\Delta x / l_{\mathrm{c}}$ conduits within each square grid cell of size $\Delta x \times \Delta x$ of the ice-sheet mesh. Hence, the local water flux is given by (Gowan et al., 2023):

$$Q_{\mathrm{w}} = \frac{\|\boldsymbol{q}_{\mathrm{w}}\| \Delta x}{\Delta x / l_{\mathrm{c}}} = \|\boldsymbol{q}_{\mathrm{w}}\| l_{\mathrm{c}}. \tag{4}$$

We take $l_{\mathrm{c}} = 10\,\mathrm{km}$, which is similar to the value considered in Gowan et al. (2023) based on observations of distances between eskers formed under the Laurentide Ice Sheet (Storrar et al., 2014). However, we acknowledge that this distance is likely to be a function of the drainage system, but leave this to be investigated in future work.

While the water mass balance is defined at the global scale, conduits must evolve at the local scale, which requires water flow to be resolved at this scale, irrespective of whether it is associated with a distributed or a localized flow pattern, similarly to what is done in Arnold and Sharp (2002), Schoof (2010), and Gowan et al. (2023). Let us denote by $S$ the cross-section area in a conduit, with characteristic width and thickness $L$ and $H$ (Figure 2), so that $S = HL$. The equations governing the geometry of the conduits and the flow of water within them in a quasi-static regime are given by the following:

$$\begin{cases} Q_{\mathrm{w}} = K S^{\alpha} \|\nabla \phi\|^{\beta - 1}, \text{ in } \Omega, & (5\mathrm{a}) \\[2mm] \|\boldsymbol{v}_{\mathrm{b}}\| h_{\mathrm{b}} + \dfrac{Q_{\mathrm{w}} \|\nabla \phi\|}{\rho_{\mathrm{i}} \mathcal{L}_{\mathrm{w}}} = 2\, n^{-n} A L^2 |N|^{n-1} N, \text{ in } \Omega, & (5\mathrm{b}) \\[2mm] N = 0, \text{ on } \Gamma_{\mathrm{gl}}, & (5\mathrm{c}) \end{cases}$$

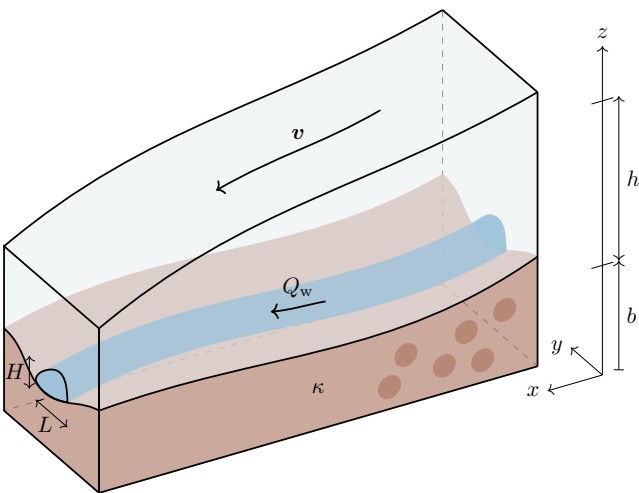

**Figure 2.** Schematic representation of a conduit (here, of a channel) in the subglacial hydrological system, characterized by a cross-sectional width $L$ and thickness $H$ and by a water flux $Q_{\mathrm{w}}$. The ice sheet has a thickness $h$, is moving at a velocity $\boldsymbol{v}$, and overlays the bedrock whose upper surface is located at $z = b$. The bedrock type is parameterized by $\kappa$, with $\kappa = 1$ corresponding to a soft bed, $\kappa = 0$ corresponding to a hard bed, and $0 < \kappa < 1$ corresponding to a mixed bed.

where $h_{\mathrm{b}}$ is the characteristic height of bed obstacles, and $A$ and $n$ are the viscosity parameters in Glen's flow law (Glen, 1955; Paterson, 1994), respectively. The first equation is a Darcy–Weisbach constitutive equation for the water flow with $K$ a conductivity coefficient, and $\alpha$ and $\beta$ exponents. Following Schoof (2010), we assume a turbulent flow, with $\alpha = 5/4$, $\beta = 3/2$, and $K = (2/\pi)^{1/4}\sqrt{(\pi + 2)/(\rho_{\mathrm{w}} f)}$, where $f$ is a friction coefficient (e.g., Clarke, 1996). The choice of a turbulent flow is justified for large water fluxes, which is the case for converging subglacial channels near the grounding line. Other

choices have been considered for subglacial hydrology in the literature; we refer to Hewitt (2011) and Werder et al. (2013) for laminar parametrizations, and to Hill et al. (2023) for a discussion of the transition between laminar and turbulent flow and their range of validity. The second equation describes the equilibrium between opening and closure rates of the conduits, assuming that the hydrological system is at equilibrium. In general, opening and closing of subglacial water systems are due to various mechanisms depending on the drainage system and bed type involved. These mechanisms include, amongst others,

melt of the subglacial conduit walls, sliding over bed protrusions, erosion of sediments, regelation, creep of ice, and creep of sediments (Hewitt, 2011; Bueler and van Pelt, 2015). Here, we consider opening rates associated with sliding over bed obstacles, melting of the conduit walls, and a closure rate due to ice creep (Nye, 1953). The bed obstacles correspond to bed protrusions if the bed is hard, and to clasts if the bed is soft, and our model treats these cases the same. Note that the first type of opening rate is associated with an inefficient drainage system, while the second one is associated with an efficient one. Finally,

the third equation comes from the equality between the subglacial water pressure and the sea-water pressure at the grounding line (Drews et al., 2017), which holds because we are considering marine-terminated ice sheets.

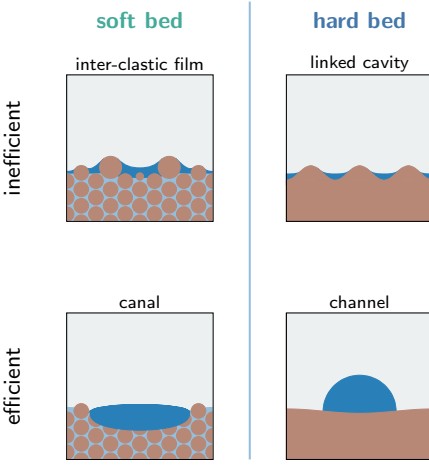

**Figure 3.** Schematic representation of the different types of drainage systems considered in this study: efficient and inefficient drainage systems on soft and hard beds.

The above model (5) is similar to the one that was used by Schoof (2010) to describe both linked-cavity systems (i.e., inefficient systems) and channels (i.e., efficient systems). Linked cavities have first been described by Kamb (1987), following earlier theoretical developments related to sliding with cavitation (Lliboutry, 1968; Kamb, 1970; Lliboutry, 1979; Iken, 1981; Fowler, 1986, 1987). For larger water fluxes, the flow in cavities localizes into channels and the system becomes efficient, forming so-called R-channels (Röthlisberger, 1972), defined as semi-circular tunnels formed within the ice of the glacier. Such conduits allow water to flow rapidly and more efficiently (Kamb et al., 1985; Iken et al., 1993; Hubbard et al., 1995; Lappegard et al., 2006).

Our model is also analogous to those that describe ice flow over soft beds. In a soft-bed system, water can infiltrate the till and weaken its strength, hence increasing basal motion. In Antarctica, till properties indicate a low porosity and hydraulic conductivity, obstructing the water circulation in the till itself and leading to water saturation (Tulaczyk et al., 2000a, b). Indeed, as the drainage rate from the till towards a subglacial aquifer is much smaller than the basal melt rate, subglacial till is supposed to be water-saturated (Bueler and van Pelt, 2015; Kazmierczak et al., 2022). Water that cannot infiltrate the till will take the form of a water film that flows around clasts higher than the water thickness (Creyts and Schoof, 2009; Kyrke-Smith et al., 2014). For large subglacial water fluxes, the film becomes unstable and conduits form a channelized network (Walder, 1982). For ice sheets, efficient conduits on a soft bed take the form of canals, which are incised in the till. They are typically much wider and shallower compared to hard-bed channels (Walder and Fowler, 1994). These different types of drainage systems, following (5), are schematized in Figure 3. Despite the qualitative differences between soft and hard bed hydrology, the governing equations are quite similar and mainly differ in their geometry, i.e., how width $L$ and thickness $H$ of the conduits are related to their cross-sectional area $S$.

To compute the effective pressure within each conduit, we combine the Darcy–Weisbach equation (5a) with the opening-closing equation (5b). This allows us to eliminate $S$ and obtain an equation for $N$ only. However, the resulting equation takes the form of a non-linear differential equation, which is not easy to solve. The complexity stems from the fact that $\nabla\phi$ depends on $N$ through $\nabla\phi = \nabla\phi_0 - \nabla N$. However, given our second simplifying assumption, we have $\nabla\phi \approx \nabla\phi_0$ outside the vicinity of the grounding line. We then obtain algebraic equations for the effective pressure and the cross-sectional area far from the grounding line, $N_\infty$ and $S_\infty$:

$$N_\infty = \left[ \left( \frac{H(S_\infty)}{S_\infty} \right)^2 \frac{\rho_i \mathcal{L}_w \|\boldsymbol{v}_b\| h_b + Q_w \|\nabla\phi_0\|}{2n^{-n}\rho_i \mathcal{L}_w A S_\infty} \right]^{\frac{1}{n}}, \tag{6a}$$

$$S_\infty = K^{-\frac{1}{\alpha}} \|\nabla\phi_0\|^{\frac{1-\beta}{\alpha}} Q_w^{\frac{1}{\alpha}}. \tag{6b}$$

Here, we have written $H(S)/S$ instead of $1/L$ to emphasize that $N_\infty$ depends on the way that $H$ depends on $S$, which is a function of the bed type.

The assumption that $\nabla\phi \approx \nabla\phi_0$ breaks down close to the grounding line because $N$ must reach a zero value there for water to be connected to the ocean, as given by (5c). Hence, the effective pressure decreases significantly in that area, leading to strong gradients in $N$. A boundary-layer analysis actually reveals that $N \approx \phi_0$ close to the grounding line, and suggests that the effective pressure can be approximated over the whole domain by

$$N = \mathrm{erf}\left[ \frac{\sqrt{\pi}}{2} \frac{\phi_0}{N_\infty} \right] N_\infty, \tag{7}$$

where $\mathrm{erf}(x)$ is the Gauss error function (see Appendix B for more details). This approximation has been verified by comparing it with numerical solutions of the differential equation for the effective pressure. This equation reveals that the assumption that $N \approx N_\infty$ becomes valid when $\phi_0$ becomes of the order of $N_\infty$, which for Thwaites Glacier is typically a few kilometres from the grounding line.

### 2.2.4 Bed rheology

One element that is lacking from the equations describing conduits is the definition of their geometry, e.g., through a relation between their thicknesses and their cross-sectional areas, $H = H(S)$. For **hard-bed** systems, we follow Schoof (2010) by assuming that $L = H = \sqrt{S}$, i.e., we consider conduits that are equally wide and thick, even though we acknowledge that the theory of linked cavities by Kamb (1987) was initially developed in the context of shallow cavities. For **soft-bed** systems, the geometry of conduits is more challenging. For small subglacial water fluxes, water takes the form of a patchy film. When the film gets thicker due to an increased water flux, its height will exceed the thickness of the smallest clasts, so that the film will be flowing in between larger clasts that are separated by a larger distance (Kyrke-Smith and Fowler, 2014; Kyrke-Smith et al., 2014). This means that $L$ increases with $H$. The relation between both depends on the spatial distribution of the clasts, as well as their thickness (Creyts and Schoof, 2009). Here, we assume $L \sim \sqrt{S}$, which is not implausible, as this corresponds to the parametrization used for cavities on a hard bed. However, a soft bed is qualitatively different from a hard bed, as the till can be

**Table 1.** Scaling for the geometry of the conduits of the hydrology system.

| Type of bed | Scaling | $H = H(S)$ |
|---|---|---|
| Hard | $L \sim H$ | $H = \sqrt{S}$ |
| Soft (inter-clastic film) | $L/H \sim F_{\text{till}}^2$ | $H = \sqrt{S}/F_{\text{till}}$ |
| Soft (canal) | $H \sim H_0$ | $H = H_0$ |

deformed. To take into account this difference, we introduce a factor $F_{\text{till}}$, defined as

$$L = F_{\text{till}}\sqrt{S}. \tag{8}$$

This deformation factor depends on the difference between ice and till viscosity, as well as the till thickness, and increases with the ability of the till to deform, provided it is sufficiently thick. For a factor $F_{\text{till}} > 1$, the effective pressure is lower compared to hard-bed systems (Beaud et al., 2022). For this reason, we here consider $F_{\text{till}} = 1.1$. For larger subglacial water fluxes, water flow channelizes into canals, for which we prescribe a thickness $H = H_0$. Here, we take $H_0 = 0.1\,\text{m}$ as prescribed in Walder and Fowler (1994) for a sand/silt sediment type located under an ice sheet . For both inefficient and efficient cases, we then set

$$H(S) = H_0 + \left(\sqrt{S}/F_{\text{till}} - H_0\right)\exp\left(-Q_{\text{w}}/Q_{\text{c}}\right), \tag{9}$$

with $Q_{\text{c}}$ a critical water flux value. Then $H \approx \sqrt{S}/F_{\text{till}}$ if $Q_{\text{w}} \ll Q_{\text{c}}$ and $H \approx H_0$ if $Q_{\text{w}} \gg Q_{\text{c}}$. In our simulations, we take $Q_{\text{c}} = 1\,\text{m}^3\,\text{s}^{-1}$ which corresponds to the scale of the water flux considered in Walder and Fowler (1994).

Finally, a **mixed bed** is composed of regions of various stiffness. We define the state of the bed through a spatial field $\kappa = \kappa(\boldsymbol{x})$ that describes the proportion of hard ($\kappa = 0$) and soft ($\kappa = 1$) bed. An intermediate value of $0 < \kappa < 1$ corresponds to a variation of the till thickness or stiffness granting it posseses intermediate rigidity compared to those attributed to the hard and soft cases. For a mixed bed, the conduit-evolution relation is the same as the one used for the hard and soft cases, where we set $H = (1 - \kappa)H_{\text{hard}} + \kappa H_{\text{soft}}$, with $H_{\text{hard}}$ (resp. $H_{\text{soft}}$) the thickness associated with a hard (resp. soft) bed. A mixed-bed system is therefore able to cover the case of linked-cavities, channels, inter-clastic films, and canals, depending on the value of $\kappa$ and of the subglacial water flow intensity. A summary of the differences between these cases can be found in Table 1.

The dependence of the far-field effective pressure $N_\infty$ with respect to the magnitude of the conduit water flux $Q_{\text{w}}$ is shown in Figure 4 for hard and soft-bed systems. It clearly shows that the effective pressure decreases with water flux for inefficient systems. For efficient systems, hard and soft-bed systems differ, where channels lead to an increase in the effective pressure, contrary to canals. Note that this last distinction can be explained as follows. For large flux values, the opening-closing relation (5b) essentially becomes a balance between the opening due to melt, which is proportional to the water flux, and ice creep. Therefore, if the flux further increases, the ice-creep term must also increase. For channels, the factor $L^2$ in the ice-creep term of (5b) increases with the water flux, but at an insufficient rate; hence, the effective-pressure factor must also increase. By

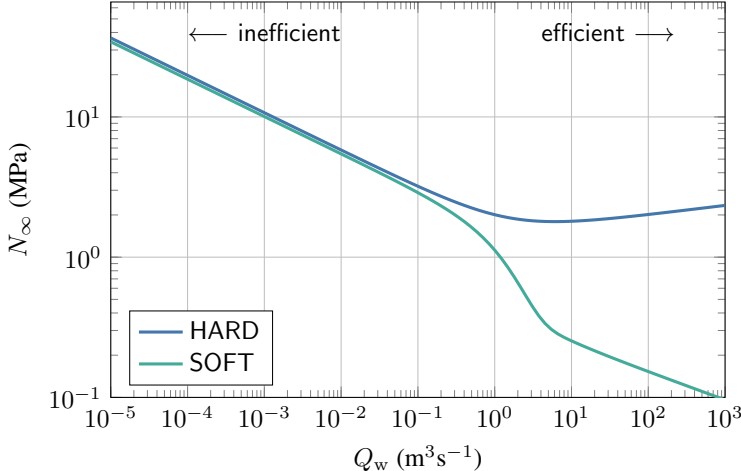

**Figure 4.** Relation between the value of the effective pressure $N_\infty$ far from the grounding line and the magnitude of the conduit water flux $Q_w$, for hard (in blue) and soft (in green) beds. The curves correspond to equation (6a) coupled with the geometric relations shown in table 1. The physical parameters are the ones displayed in the appendix A, with $A = 2.4 \times 10^{-24}\,\mathrm{Pa}^{-3}\,\mathrm{s}^{-1}$, $\|\boldsymbol{v}_b\| = 0.5 \times 10^{-5}\,\mathrm{m\,s}^{-1}$, and $\|\nabla\phi_0\| = 100\,\mathrm{Pa\,m}^{-1}$.

contrast, canals, because of their shallower form, are such that this factor increases at a much faster rate. As a consequence, the effective pressure must decrease in that case.

Besides soft, hard, and mixed beds, we also consider entirely efficient and inefficient drainage systems to gauge the sensitivity of both separately, independent of the subglacial water flux. By default, our model is such that the subglacial system naturally transitions from one to another depending on the subglacial water flux. This transition happens because the melting term, which is proportional to the water flux, becomes dominant over the sliding term in the left-hand side of (5a) as the water flux increases. By removing the opening term associated with the sliding over obstacles, $\|\boldsymbol{v}_b\|h_b$, from equations (5b) and (6a), it is possible to force the model to produce an entirely efficient drainage system. In this case, we also set $Q_c = \infty$, which guarantees that the conduit geometry is the one of an inefficient system for soft beds. Similarly, to force an entirely inefficient system, the efficient component, $Q_w\|\nabla\phi\|/\rho_i\mathcal{L}_w$, can be removed from (5b), together with the condition that $Q_c = 0$.

## 3 Idealized experiments

### 3.1 Experimental setup

As a first test of the hydrological model, we consider a flowband geometry for a marine ice sheet, taken from the benchmark projects MISMIP and MISMIP3d (Pattyn et al., 2012, 2013), and which consists of a linearly downward-sloping bed towards the ocean (Figure 5a). On this bed topography a marine ice sheet is developed with a spatial resolution of $500\,\mathrm{m}$, following the

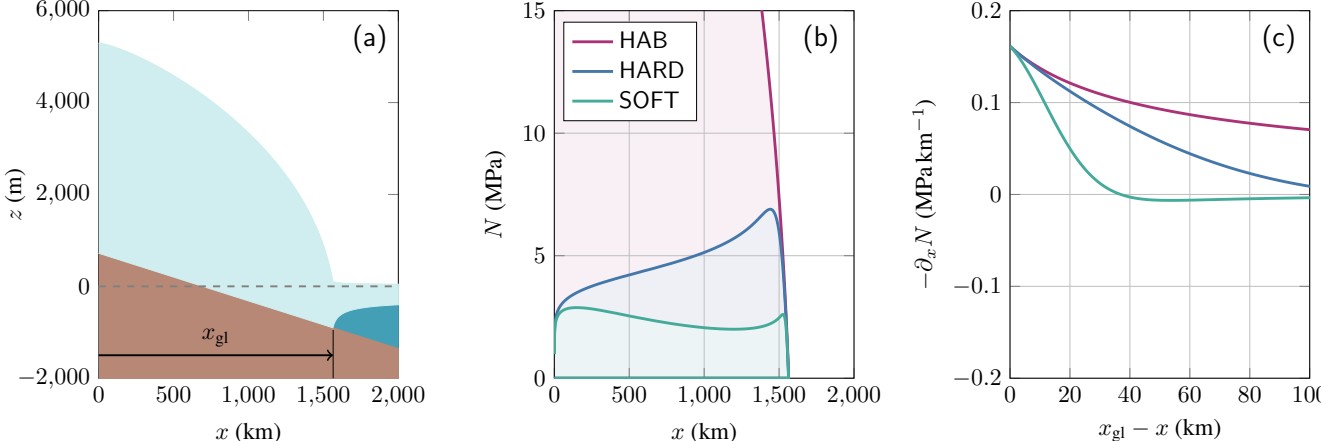

**Figure 5.** Initial state for the experiments on idealized conditions, based on the MISMIP geometry (Pattyn et al., 2012). (a) The initial MISMIP steady-state ice-sheet geometry; $x_{\mathrm{gl}}$ corresponds to the distance to the grounding line. (b) Flowband profiles of the subglacial effective pressure for HAB (in purple), HARD (in blue), and SOFT (in green) hydrological models. (c) Gradient of the subglacial effective pressure near the grounding line for HAB (in purple), HARD (in blue), and SOFT (in green) hydrological models.

set-up described in the EXP1 of the MISMIP experiments (Pattyn et al., 2012, see Figure 5a). The steady state obtained with these conditions is considered to be the 'reference state'.

In our experiments, we use a regularized Coulomb friction law combined with hydrological models, while the reference state from the MISMIP set-up has been obtained with a Weertman friction law. To guarantee that the thickness and velocity fields obtained in the reference state are still compatible with a steady state, we modify the friction coefficient at each position, following the method of Brondex et al. (2017, 2019). In practice, an iterative nudging method is used so that the basal friction matches the basal friction obtained in the reference state. Here, the subglacial hydrologies are generated with a uniform basal melt rate underneath the grounded ice sheet of $\dot{m}/\rho_{\mathrm{w}} = 5 \times 10^{-3}\,\mathrm{m\,a^{-1}}$, which corresponds to the mean basal melt rate of the Antarctic Ice Sheet (Pattyn, 2010). By construction, this method yields initial states that are steady states and in which both the geometry and the velocity field are identical for each type of hydrology, allowing a direct comparison between them. The initial ice-sheet effective-pressure profiles are shown in Figure 5b.

We consider a uniform hard and soft bed and compare these to an experiment where (i) the effective pressure is determined according to the HAB model (for 'height-above-buoyancy'), which assumes a perfect connection with the ocean, and (ii) basal sliding being independent of subglacial pressure or bed rheology (NON). The effective pressure in the HAB model is therefore simply defined by

$$N = \rho_{\mathrm{i}} g h - \rho_s g \max(0, -b), \tag{10}$$

where $\rho_s$ is the density of sea water. It is the most common parametrization of $N$ used (e.g., Tsai et al., 2015).

For all models, the effective pressure is high in the interior and decreases towards the grounding line, where it becomes zero by definition. For the HAB model, the horizontal gradient of the effective pressure is the highest, mainly governed by the geometry of the bedrock and surface slopes. For HARD and SOFT, representing the hard-bed and soft-bed systems, respectively, the effective pressure varies due to variations in both the subglacial water flux and the cross-sectional size of the conduits, according to equation (6a) (Figure 5b).

Besides, we also compare the impact of the drainage efficiency, by comparing the cases where only efficient (eff) or inefficient (ineff) systems are allowed to develop. Note that, by default, the switch between both systems (efficient/inefficient) is determined based on the subglacial water flux magnitude.

## 3.2  Results: the efficient to inefficient switch

A first experiment aims at understanding the switch between efficient and inefficient drainage systems. We force the MISMIP flowband setup with a sinusoidal variation in subglacial meltwater that is then routed across a hard bed (Figure 6a). The response in effective pressure and sliding velocity do not simply follow the same sinusoidal pattern, but the ice velocity accelerates and decelerates as a function of whether the system evolves into an efficient or inefficient drainage system (Figure 6b and c). Sudden speedups correspond to sudden drops in the effective pressure and occur when either the amount of meltwater diminishes or increases. Note that these sudden changes do not appear if the model is entirely efficient or entirely inefficient. Similar characteristics of the subglacial hard-bed system have been described by Schoof (2010) and are shown in Figure 6d. Although our model is a relatively crude parametrization of the subglacial system, the intrinsic instability related to the switch between efficient and inefficient drainage systems is captured.

## 3.3  Results: perturbation experiment

For each hydrological model, the initial steady state is perturbed by instantaneously reducing the net mass accumulation rate to $0.2\,\mathrm{m\,a^{-1}}$. The ice sheet is then allowed to evolve for a period of 20,000 years with a time step of 5 years, eventually reaching a (near) steady state. The hydrological model is updated at each time step (see also Appendix C). The first series of experiments consists of comparing the response of the different models (NON, HAB, HARD, SOFT), as well as the fully efficient (eff) and fully inefficient (ineff) cases for HARD and SOFT models to this perturbation. The reduction in surface accumulation generally leads to a slowdown of the ice velocity as well as a smaller ice sheet. This reduction also results in a slight grounding-line retreat (NON in Figure 7). However, linking basal sliding to *any* of the hydrological models that are a function of effective pressure at the base of the ice sheet, leads to a significant grounding-line retreat and overall grounded mass loss (Figure 7). Of all subglacial models, HAB is the most sensitive to the forcing, i.e., leading to the highest mass loss after forcing. Several factors may be responsible for this, as has been shown in Figure 5c: it is not only that the effective pressure is low near or at the grounding line, but that also the effective-pressure gradient upstream of the grounding line is the highest for the HAB model. The fact that the sensitivity of grounding-line retreat increases with increasing gradient in effective pressure near the grounding line is shown in Figure 5c. While this gradient is equally high near the grounding line for all model configurations, it drops quite quickly with distance from the grounding line for soft-bed systems, resulting in a minimal grounding-line retreat. Since per definition the

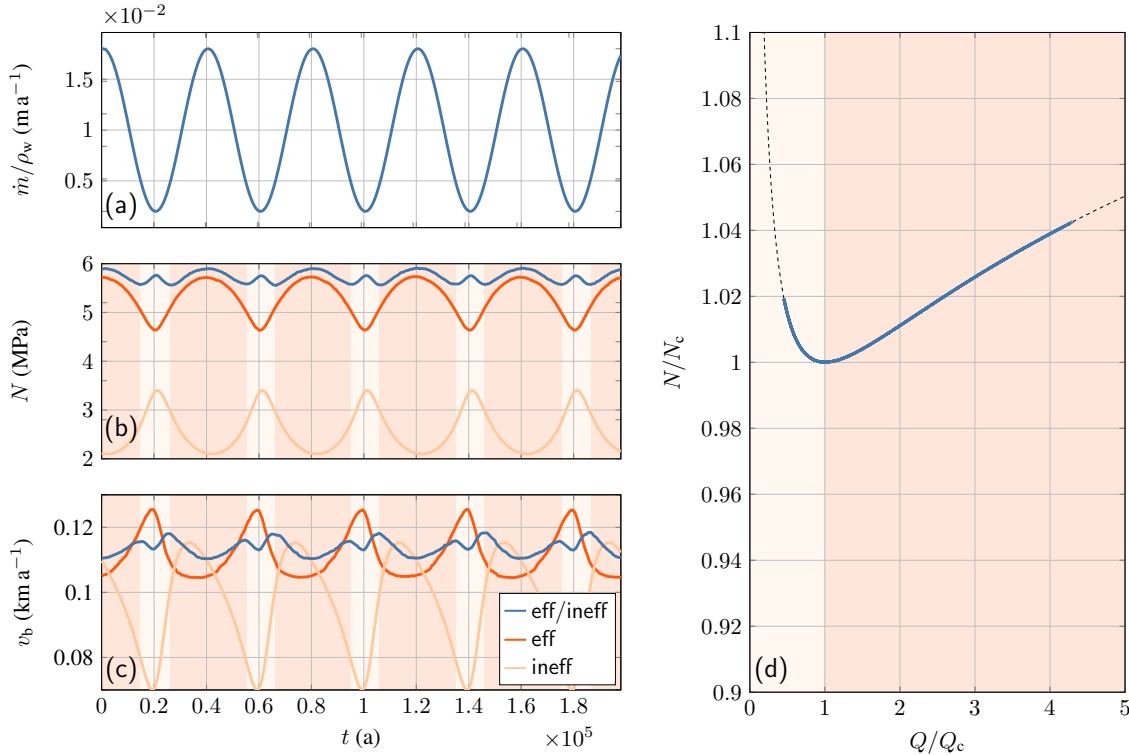

**Figure 6.** (a) Forcing of the MISMIP geometry with a sinusoidal variation in subglacial meltwater, for a hard-bed system. (b) Response to the forcing in effective pressure, and (c) basal sliding velocity (blue: efficient/inefficient; red: entirely efficient; beige: entirely inefficient). (d) Relationship between subglacial water flux and effective pressure for both the efficient/inefficient drainage system. The dashed line corresponds to equation (6a), and $N_c$ and $Q_c$ are critical values for a hard-bed system as defined in Schoof (2010). The light pink (resp. dark pink) areas correspond to regions in which the efficient/inefficient system is an inefficient (resp. efficient) regime.

NON model, representing the solution independent of effective pressure, has a zero gradient in effective pressure, it therefore leads to the least grounding-line retreat due to forcing compared to the other hydrological models, as shown hereunder.

## 4 Application to Thwaites Glacier

### 4.1 Experimental setup

For Thwaites Glacier, necessary input data are the present-day ice-sheet surface and bed geometry from BedMachine v2 (Morlighem et al., 2019), surface mass balance and temperature from RACMO2.3p2 (van Wessem et al., 2018), and a prescribed field for the geothermal heat flux (Shapiro and Ritzwoller, 2004). All datasets were resampled at a spatial resolution of 2 km. The simulations are performed at that spatial resolution and with a time step of 0.05 years.

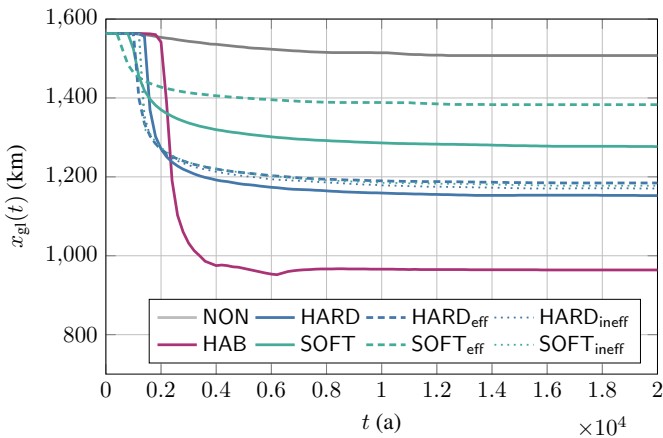

**Figure 7.** Grounding-line evolution after a sudden reduction in surface accumulation rate for the MISMIP setup for NON (in grey), HAB (in purple), HARD (in blue), SOFT (in green), HARD efficient (blue dashed line), HARD inefficient (blue dotted line), SOFT efficient (green dashed line) and SOFT inefficient (green dotted line) hydrological models.

Similar to the idealized experiments, a reference state is obtained with a Weertman friction law. To obtain this state, the
basal friction field is iteratively adapted to obtain a steady-state ice sheet with grounded ice thickness as close as possible to the observed thickness using a nudging method (Pollard and DeConto, 2012; Pattyn, 2017; Coulon et al., 2024). For the floating ice shelves, the sub-shelf melt/accretion is adjusted to keep the ice thickness comparable to observed (Coulon et al., 2024). In a second step, the initial friction field corresponding to a given hydrological model is obtained with the same method as explained for the idealized experiments. Initial bedrock and surface topographies, as well as the ice velocity field can be found
in the Supplementary Material S2, the friction coefficient fields after initialization in the Supplementary Material S3 and the effective pressure values for HAB, HARD and SOFT in the Supplementary Material S4. To evaluate the model drift of this initialization, the model is run forward in time according to the different hydrological models, leading to a mass change after 100 years corresponding to 1–2 mm of sea-level rise. Thanks to this small model drift, the control run does not have to be subtracted from the forcing runs, as was the case in Kazmierczak et al. (2022). It also demonstrates the improvements that have
been made in terms of model initialization (Coulon et al., 2024).

All simulations start from this initial steady state, corresponding to the 2015 conditions. We then run the model forward until 2100, under present-day climate conditions (atmospheric and oceanic). Sub-shelf melt rates are calculated by the PICO model (Reese et al., 2018) with an effective heat exchange velocity value of $3 \times 10^{-5}\,\mathrm{m\,s^{-1}}$, which gives a realistic melt pattern for present-day ocean temperatures and salinity. Since these dynamic melt rates replace the optimized basal melting/accretion
field, the ice-sheet system is no longer in steady state anymore and the grounding line reacts to the applied oceanic forcing by a small retreat over the course of this century (Reference experiment NON in Figure 8a). Note that the mass loss for the NON experiment results in sea-level rise on the order of 10 mm by 2100, which is an order of magnitude larger than the model drift ($\sim$1 mm).

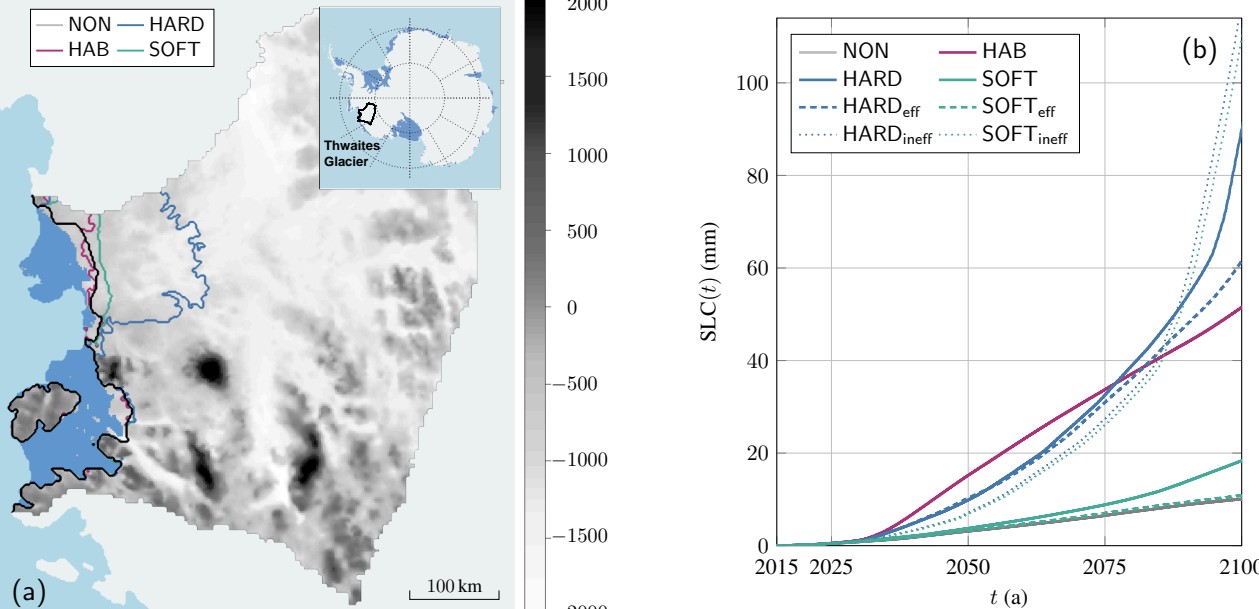

**Figure 8.** Effect on the subglacial hydrological models in the Thwaites Glacier set-up. Present-day climate conditions (atmospheric and oceanic) are applied from 2015 to 2100. (a) Grounding-line positions of Thwaites Glacier (bedrock elevation (m) in the background) in 2100 under constant present-day climate conditions (atmospheric and oceanic) for the NON (black), HAB (purple), HARD (blue) and SOFT (green) models. The inset in the upper right corner shows the position of Thwaites Glacier within Antarctica. (b) Sea-level contribution from Thwaites Glacier from 2015 to 2100 under constant present-day climate conditions (atmospheric and oceanic) for the NON (grey), HAB (purple), HARD (blue), SOFT (green), HARD efficient (blue dashed line), HARD inefficient (blue dotted line), SOFT efficient (green dashed line) and SOFT inefficient (green dotted line).

This simulation under present-day forcing conditions is repeated for the different subglacial hydrological models HAB,
HARD and SOFT. Akin to this, the spatial variability of the HARD and SOFT bed models is assessed by considering heteroge-
neous and/or mixed beds for Thwaites Glacier (Muto et al., 2019; Alley et al., 2022). By heterogeneous we mean that the spatial
field is composed of soft and hard bed portions, while for a mixed bed, we consider that a particular grid cell is composed of a
mix of hard and soft bed, or $0 < \kappa < 1$. We can therefore have different cases: homogeneous uniform (the whole bed domain
with either $\kappa = 0$ or $\kappa = 1$) or mixed (the whole bed domain with a constant value $0 < \kappa < 1$), as well as heterogeneous uni-
form (the domain composed of portions of hard bed and soft bed) and mixed ($0 \leq \kappa \leq 1$). For the heterogeneous uniform beds,
Joughin et al. (2009) and Muto et al. (2019) suggest that soft beds are mainly found in subglacial depressions and hard beds
generally on topographic highs, which allows us to make the separation between both based on the subglacial topography, with
the soft layer occupying the deeper basins. In a first experiment, we set the transition between soft and hard bed at a bedrock
elevation of 1000 m below sea level (Figure 9c). In a second experiment a transition zone is considered (heterogeneous mixed),
where $\kappa$ is linearly varied between a depth of 500 and 1500 m below sea level (Figure 9d). We tested the influence of the

nature of the drainage system itself by applying only inefficient, efficient or both drainage systems for the different bed types described above.

## 4.2 Results: subglacial hydrology on homogeneous beds

As mentioned above, even under present-day atmospheric and oceanic forcing, the reference experiment NON leads to a slight retreat of the grounding line over the period 2015-2100 that continues thereafter. This is in line with large-scale model experiments (Coulon et al., 2024) showing that Thwaites Glacier may collapse, i.e., that it will continue to retreat even if the forcing is completely stopped, under present-day climatic conditions on time scales of several centuries.

The same retreat behavior is observed for the experiment including subglacial hydrology. However, in all cases, subglacial hydrology accelerates the retreat of the grounding line by 2100 (Figure 8a). For instance, by 2100, we observe a sea-level contribution of around $50\,mm$ for HAB, $95\,mm$ for HARD, $20\,mm$ for SOFT, while only $10\,mm$ for NON (Figure 8b). It is important to note that for all the subglacial models a collapse of Thwaites Glacier is engaged under present-day climate forcing conditions. However, only the hard-bed model (HARD) results in a collapse in less than $100\,years$. Not all major mass losses coincide with a distinct grounding-line retreat, as the HAB model exhibits significant thinning of the ice for a modest grounding-line retreat compared to the other subglacial models.

The efficiency of subglacial drainage is tested for both a uniform homogeneous soft and hard bed (Figure 9a and b). As demonstrated in the idealized experiments, it is also possible to force the drainage to be efficient (eff) or inefficient (ineff). The results corroborate the previous experiment, i.e., that for a hard bed, a large amount of ice is lost by 2100 compared to the soft bed. The only exception is that a similar amount of (high) mass loss is observed for the inefficient drainage systems both for soft and hard beds (Figure 9a and b).

## 4.3 Results: subglacial hydrology on heterogeneous beds

Figure 9c and d show the grounding-line positions for a heterogeneous bed, where the subglacial basins of Thwaites are considered to be soft bed, and the topographic highs hard bed. As previously mentioned, the limit between both is set at a depth of $1000\,m$ below present-day sea level. A peculiarity of this setting is that the present-day grounding line is situated on a hard bed, and that the soft bed region is only reached further inland. In the first experiment (Figure 9c) the transition between both bed rheologies is sharp (uniform heterogeneous); in the second experiment (Figure 9d) there is a transition zone (mixed heterogeneous). Despite the hard bed conditions in the present-day grounding zone, a larger mass loss by 2100 is observed for the mixed case. Similar to the previous experiment, the largest mass loss is observed for inefficient drainage, irrespective of the bed rheology.

The results on mixed heterogeneous bed experiments show that the nature of the bed near the grounding line determines the sensitivity of the glacier. With a sharp transition, the motion of the grounding line over the hard-to-soft bed system will lead to a stabilization effect delaying grounding-line retreat. Such stabilization is less pronounced for a mixed bed, where hard-bed physics also play a role in the transition zone. Similar results are presented in the Supplementary S4 with the hard and soft bed zones locations exchanged.

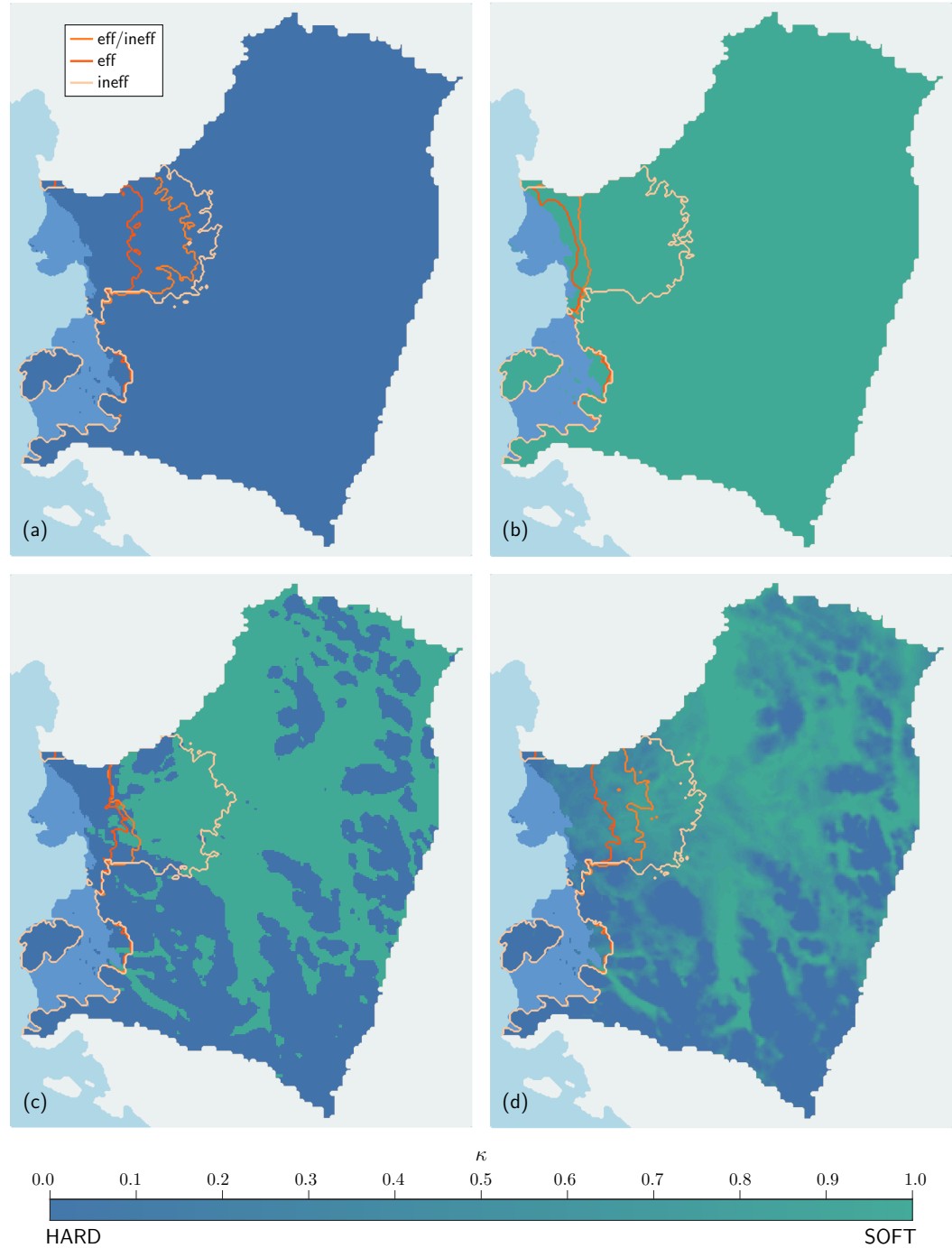

**Figure 9.** Grounding-line positions of Thwaites Glacier in 2100 under constant present-day climate conditions (atmospheric and oceanic) for the subglacial model (inefficient and efficient, entirely efficient, and entirely inefficient) on a) a uniform homogeneous hard bed; b) a uniform homogeneous soft bed; c) a heterogeneous uniform bed, with a sharp transition and d) a heterogeneous bed, with a transition zone of mixed bed.

## 5    Discussion

We have developed a novel and unified subglacial hydrological model that incorporates efficient and inefficient drainage and that applies to both soft and hard beds. It represents different discharge types, ranging from channels and canals to cavities. While the model is a simplification compared to higher-resolution hydrological models, it seems to capture the main characteristics of subglacial flow, and incorporates the feedbacks associated with basal sliding and the ice dynamical response.

In this section, we first discuss the influence of the hydrological model and the bed type. We then describe the hydrological

feedback that may explain the increased sensitivity of the ice sheet due to subglacial hydrology. Finally, we comment on the limitations of the model and suggest improvements to our model.

### 5.1    Influence of subglacial conditions

Our experimental results show that subglacial hydrology and the rheology of the bed (soft, hard and its related spatial variability) have a major impact on the dynamics of marine ice sheets and the West Antarctic Ice Sheet in particular. Taking into

account subglacial hydrology systematically leads to a higher ice-sheet sensitivity for a given climate forcing. This is mainly due to the reduction of the effective pressure near the grounding line, which migrates upstream with a retreating grounding line.

Traditionally, large scale ice sheet models tend to calculate effective pressure at the base of the ice sheet using the HAB parametrization, based on the height above flotation. This model assumes that sea water infiltrates at the grounding line,

increasing the water pressure and reducing the effective pressure in the grounding zone. This leads to significant increases in basal sliding near the grounding line. The idealized experiments clearly demonstrate that this model leads to the highest mass loss after a perturbation and probably overestimates the contribution of ice sheets to climate forcing.

However, in more complex settings, such as Thwaites Glacier, the HAB model remains sensitive, but also allows the grounding line to stabilize during its retreat on subglacial ridges. These bed peaks are known to have a strong impact on grounding-line

dynamics, and their influence on ice-sheet stability is the subject of recent research (Robel et al., 2022a).

Results of our simplified hydrological model show that sliding over a hard bedrock (HARD) leads to the largest reduction of the grounded domain and the highest sea-level contribution for a given forcing, while sliding over a soft bed (SOFT) yields the smallest grounding-line retreat and sea-level contribution (Figure 8). In terms of drainage efficiency, the concentration of water flow in efficient conduits, especially in canals, but also for channels, slows down the retreat of the grounding line,

which aligns with one of the conclusions of Schoof (2010). This can be explained by a higher basal friction due to the higher effective pressure in hard bed channels (Hager et al., 2022), and by the lower effective pressure variation in canals, leading to a reduced impact of subglacial hydrology on the basal friction field. It is interesting to note that in the case of hard beds, our effective-pressure results are similar to those obtained by Hager et al. (2022) with the MALI model (Hoffman et al., 2018), using a subglacial drainage model built on cavities and channels.

Besides the hard bed, inefficient systems also lead to the highest mass loss of Thwaites Glacier over this century. We observe that the largest mass loss occurs when the effective pressure gradient is large towards the grounding line and less so when

effective pressure is low, such as in canals (Figure 8b, see video supplements). This observation aligns with the work of Iken (1981), specifying that the highest velocities is not observed where effective pressures is lowest, but rather when cavities enlarge due to an increase in subglacial water pressure. Therefore, inefficient systems on both soft and hard beds show very

similar results, which is consistent with the representation of the considered drainage systems. Such systems are quite similar as bumps in the hard system correspond to the clasts in the soft system. However, the lower effective pressure in the soft-bed system, associated with the deformation of saturated till, slows down grounding-line retreat.

In general, a soft bed near the grounding line slows down its retreat under climatic forcing. However, transitions between bed types also influence the speed of the grounding-line retreat. A binary switch from hard to soft is more likely to stabilize

the grounding-line position than a smooth transition. Moreover, taking the total proportion between soft and hard beds and applying it uniformly across the entire domain does not allow to capture the variation introduced by the spatial variability of bed rheology and the associated drainage system.

## 5.2 Hydrological feedback

The increased sensitivity observed with hydrological models can be explained by a positive feedback between grounding-line

migration and the reduction in basal friction at or near the grounding line. Formally, frictional stress $\boldsymbol{\tau}_b$ can be split into two components: $\tilde{\boldsymbol{\tau}}_b$, which is the value of the friction stress with the initial effective-pressure field, and $\Delta\tilde{\boldsymbol{\tau}}_b$, which is the deviation with respect to this value:

$$\boldsymbol{\tau}_b = \tilde{\boldsymbol{\tau}}_b + \Delta\tilde{\boldsymbol{\tau}}_b, \tag{11}$$

where

$$\tilde{\boldsymbol{\tau}}_b(t) = C N_0 \left( \frac{\|\boldsymbol{v}_b\|}{\|\boldsymbol{v}_b\| + v_0} \right)^{\frac{1}{m}} \frac{\boldsymbol{v}_b}{\|\boldsymbol{v}_b\|}, \tag{12a}$$

$$\Delta\tilde{\boldsymbol{\tau}}_b(t) = C \left[ N(t) - N_0 \right] \left( \frac{\|\boldsymbol{v}_b\|}{\|\boldsymbol{v}_b\| + v_0} \right)^{\frac{1}{m}} \frac{\boldsymbol{v}_b}{\|\boldsymbol{v}_b\|}, \tag{12b}$$

and where $N_0 = N(t=0)$ is the effective pressure for the initial state. Because of the initialization procedure, $\tilde{\boldsymbol{\tau}}_b$ is initially the same for every hydrological model. In particular, it is also equal to the absence of subglacial hydrology (NON). Therefore, $\Delta\tilde{\boldsymbol{\tau}}_b$ stems from the evolution of the subglacial system and its influence on basal friction. In other words, $\Delta\tilde{\boldsymbol{\tau}}_b$ is associated

with the spatial and temporal evolution of the effective pressure.

Due to the evolution of subglacial hydrology in time, an instability mechanism may appear near the grounding line, as the effective pressure is always low, and the gradient of the effective pressure is the largest (Figure 5). The zone of low effective pressure migrates with a migrating grounding line. Such migration obviously does not take place when the subglacial hydrological field is kept constant or when subglacial hydrology is not linked to basal sliding (or not considered; NON). For a

retreating grounding line, such linkage actually amplifies grounding-line retreat, as the friction stress close to the grounding line is also reduced following this retreat, leading to a positive feedback mechanism. This reduction in $\boldsymbol{\tau}_b$ stems from a reduction of $\tilde{\boldsymbol{\tau}}_b$, but most importantly from a large value in the magnitude of $\Delta\tilde{\boldsymbol{\tau}}_b$, which is typically negative. This essentially explains

the distinction between the HAB and the HARD/SOFT models. The HAB model is purely local as it depends on the geometry of the ice sheet at the position where the effective-pressure is evaluated. By contrast, the HARD/SOFT models also depend on the subglacial water flux and on the distance with respect to the grounding line. This distinction allows for a perturbation near the grounding line to 'propagate' upstream for the latter models. As this is not the case for HAB models, it potentially enables the grounding line to stabilise near a ridge, for instance. This distinction can be observed in the video supplement.

The qualitative assessment can be quantified in the case of a flowline according to the shallow-shelf approximation. Following previous work (Weertman, 1974; Schoof, 2007a, 2012), a steady-state marine ice sheet is unstable if

$$\partial_s q - a < 0, \tag{13}$$

in which $q$ is the flux at the grounding line, $s$ is a coordinate parameterizing the position along the ice sheet, and $a$ is the net mass accumulation rate. For a large class of friction laws (Schoof, 2007b; Tsai et al., 2015; Gregov et al., 2023), the flux at the grounding line can be approximated as

$$q = q_0 \, C^{-\frac{1}{m+1}} \left[ -(\rho_\mathrm{w}/\rho_\mathrm{i})b \right]^r, \tag{14}$$

where $q_0, r > 0$. It follows that for a uniform friction coefficient, a steady-state position on a up-sloping (retrograde) bed is always unstable, i.e.,

$$\partial_s q - a = -rq(\rho_\mathrm{w}/\rho_\mathrm{i}) \left[ -(\rho_\mathrm{w}/\rho_\mathrm{i})b \right]^{-1} \partial_s b - a < 0. \tag{15}$$

However, an ice-sheet on a downward-sloping bed can be stable. For a spatially variable friction coefficient $C = C(s)$, the instability condition becomes

$$-\frac{qC^{-1}\partial_s C}{m+1} - rq(\rho_\mathrm{w}/\rho_\mathrm{i}) \left[ -(\rho_\mathrm{w}/\rho_\mathrm{i})b \right]^{-1} \partial_s b - a < 0. \tag{16}$$

This inequality can be achieved more easily for a downward-sloping (prograde) bed, compared to equation (15). Indeed, if $\partial_s C$ is positive and large at the grounding line, then the left-hand side of (16) is reduced, and the instability condition is easier to fulfill. The initialization produces this condition for the HAB, HARD and SOFT models, contrary to the NON case, where $C(s)$ has to increase considerably close to the grounding line to compensate for the vanishing effective pressure in order to obtain a frictional stress similar to the one obtained by the absence of hydrology. Overall, this implies that the ice sheet is less stable, and therefore exhibits a higher sensitivity to external forcings. This instability was explored in greater details with a similar model in Lu and Kingslake (2023).

### 5.3 Model limitations

Although our simulations for hard, soft, and mixed beds allow to better assess the variability of the response of ice sheets to a climate forcing, there remain some limitations. Our subglacial hydrology models do not include variations of drainage density or of effective pressure below the resolution of the ice-sheet discretization. This is a clear limitation, as we have shown that the spatial variability plays an important role in the numerical experiments. From a physical perspective, improvements could

be made on the representation of physical processes. For example, till water pressure is omitted in the soft bed model, and till deformation is only crudely parameterized. Water flow within the till and exchanges with the neighbouring area (especially in the case of a variation in ice thickness) could well modify subglacial water flow and therefore ice-sheet dynamics (Robel et al., 2023). Recent studies have also shown that sea-water intrusion may impact grounding-line dynamics through modifying the subglacial hydrology, hence increasing the instability (Robel et al., 2022b; Bradley and Hewitt, 2024). This also suggests that the grounding line should be considered a grounding zone that allows for such intrusion, in agreement with recent observations (Rignot et al., 2024). Furthermore, erosion, deposition and sedimentary transport processes that are not taken into account could play a role in the variability of effective pressure at the base of the ice sheet (Ng, 2000; Hewitt and Creyts, 2019; Stevens et al., 2022). Finally, even if our results remain qualitatively valid if the parameter settings are modified (see Appendix D), the latter could be subject to more scrutiny, ideally within a probabilistic framework (Bulthuis et al., 2019; Verjans et al., 2022; Coulon et al., 2024). This analysis could then be used to quantify the uncertainties in the projections obtained in the simulations.

## 6 Conclusions

We developed a novel and simplified model of subglacial hydrology that applies to both soft and hard beds, thereby representing both efficient and inefficient discharge types. The hydrological model is dynamically linked to an ice-sheet model (Kori-ULB) via a regularized Coulomb friction law. Despite its relative simplicity, our model allows to obtain results that agree with multiple previous studies. Our experiments are in agreement with Kazmierczak et al. (2022), showing that the type of subglacial hydrology modulates the basal sliding and that considering subglacial hydrology enhances the ice-sheet response to sliding. Our tests on the spatial and temporal variability of bed rheology also show that a soft-bed system in the grounding zone tends to stabilize a grounding line more easily compared to other bed rheologies. By investigating various drainage efficiencies, our results concur with those of Schoof (2010) by showing that a channelization leads to ice deceleration as well as grounding-line stabilization. The switch between efficient and inefficient drainage has clearly been shown in our experiments where subglacial water input has been varied. Moreover, our results agree with Iken (1981) by the fact that the highest sliding is not occurring at the highest subglacial water pressure, but rather where basal cavities are growing (i.e., when the basal water pressure is increasing downstream). Furthermore, we obtain the largest response in grounding-line retreat for those subglacial conditions where the gradient in effective pressure is the largest, not where its value is the lowest. Therefore, highly saturated grounding zones on soft beds seem to be less responsive than hard-bed systems, where such large gradients in the vicinity of the grounding line occur. While our results for Thwaites Glacier for a hard bed are qualitatively similar to those of Hager et al. (2022), the ability of model to reproduce such results could be studied in more detail.

Overall, our study also emphasizes the necessity for more accurate data and observations of the bed rheology of the Antarctic Ice Sheet at different spatial scales (Parizek et al., 2013; Koellner et al., 2019; Muto et al., 2019; Alley et al., 2022; Li et al., 2022; Aitken et al., 2023). Similarly, the observation of efficient and inefficient subglacial water drainage systems and a connection with numerical results is required (Schroeder et al., 2014; Hager et al., 2022; Dow, 2022b).

*Code and data availability.* The code and reference manual of the Kori-ULB ice-sheet model are publicly available on GitHub via https: //github.com/FrankPat/Kori-dev. The specific Kori-ULB model version used in this study, the simulations outputs, and the scripts needed to produce the figures and tables are hosted on Zenodo (https://zenodo.org/records/13895589, Kazmierczak et al., 2024). All datasets used in this study are freely accessible through their original references.

*Video supplement.* Videos of the evolution of Thwaites until its collapse for the NON, HAB, HARD, and SOFT cases are accessible at https://github.com/tgregov/thwaitesVideos/.

*Author contributions.* EK and TG conceived the study in collaboration with VC and FP. EK and TG constructed and implemented the model and performed the simulations. All authors contributed to the analysis and interpretation of the results. The manuscript was written by EK, TG, and FP with numerous feedbacks from VC.

*Competing interests.* The authors declare that they have no conflict of interest.

*Acknowledgements.* This publication was supported by PROTECT. This project has received funding from the European Union's Horizon 2020 research and innovation programme under grant agreement No 869304, PROTECT contribution number 134. EK acknowledges financial support by PROTECT. TG is supported by the Fonds de la Recherche Scientifique (F.R.S.-FNRS, Belgium) through a Research Fellowship. EK and TG acknowledge the Fonds David et Alice Van Buuren and the Fondation Jaumotte-Demoulin. TG would like to thank George Lu for interesting discussions related to flux conditions and subglacial hydrology, and Matthew Hoffman, Trevor Hillebrand, and Mauro Perego for feedback on an earlier version of this work.

# Appendix A: List of symbols

**Table A1.** List of symbols used for the model (Greek alphabet).

| Symbol | Description | Units | Value |
|---|---|---|---|
| $\alpha$ | Exponent in Darcy–Weisbach relation | - | $5/4$ |
| $\beta$ | Exponent in Darcy–Weisbach relation | - | $3/2$ |
| $\Gamma_d$ | Boundary of the basin | - | - |
| $\Gamma_{gl}$ | Grounding line | - | - |
| $\kappa$ | Indicator of the heterogeneity content of the bed | - | - |
| $\rho_i$ | Density of ice | $kg\,m^{-3}$ | $9.17 \times 10^3$ |
| $\rho_s$ | Density of sea water | $kg\,m^{-3}$ | $1.03 \times 10^3$ |
| $\rho_w$ | Density of fresh water | $kg\,m^{-3}$ | $1.00 \times 10^3$ |
| $\boldsymbol{\tau}_b$ | Basal shear stress | Pa | - |
| $\phi$ | Hydraulic potential | Pa | - |
| $\phi_0$ | Geometric potential | Pa | - |
| $\Omega$ | Grounded ice domain | - | - |

**Table A2.** List of symbols used for the model (Latin alphabet).

| Symbol | Description | Units | Value |
|---|---|---|---|
| $A$ | Viscosity parameter in Glen's flow law | $\mathrm{Pa}^{-n}\mathrm{s}^{-1}$ | - |
| $b$ | Bedrock elevation | m | - |
| $C$ | Friction coefficient | - | - |
| $f$ | Friction coefficient for a turbulent flow | - | 0.10 |
| $F_{\text{till}}$ | Factor for the conduits geometry in a till | - | 1.10 |
| $g$ | Gravitational acceleration | $\mathrm{m\,s}^{-2}$ | 9.81 |
| $G$ | Geothermal heat flux | $\mathrm{W\,m}^2$ | - |
| $h$ | Ice thickness | m | - |
| $h_{\text{b}}$ | Thickness of obstacles | m | 0.10 |
| $H$ | Thickness of conduits | m | - |
| $H_{\text{hard}}$ | Thickness of conduits over a hard bed | m | - |
| $H_{\text{soft}}$ | Thickness of conduits over a soft bed | m | - |
| $H_0$ | Thickness of canals | m | 0.10 |
| $K$ | Conductivity coefficient in Darcy–Weisbach relation | $\mathrm{kg}^{1-\beta}\,\mathrm{m}^{2\beta-2\alpha+1}\,\mathrm{s}^{2\beta-3}$ | - |
| $L$ | Width of conduits | m | - |
| $l_{\text{c}}$ | Distance between conduits | m | $1.00 \times 10^4$ |
| $\mathcal{L}_{\text{w}}$ | Latent heat of fusion of water | $\mathrm{J\,kg}^{-1}$ | $3.35 \times 10^5$ |
| $m$ | Power-law exponent | - | 3.00 |
| $\dot{m}$ | Melt rate | $\mathrm{kg\,m}^{-2}\,\mathrm{s}^{-1}$ | - |
| $\dot{m}_{\text{w}}$ | Melt rate associated with the subglacial water flow | $\mathrm{kg\,m}^{-2}\,\mathrm{s}^{-1}$ | - |
| $n$ | Exponent in Glen's flow law | - | 3.00 |
| $\boldsymbol{n}$ | Normal vector to a boundary | - | - |
| $N$ | Effective pressure | Pa | - |
| $N_\infty$ | Far-field effective pressure | Pa | - |
| $\boldsymbol{q}_{\text{w}}$ | Subglacial water flux | $\mathrm{m}^2\,\mathrm{s}^{-1}$ | - |
| $q_T$ | Thermal conduction flux | $\mathrm{W\,m}^2$ | - |
| $Q_{\text{w}}$ | Volumetric water flux in a conduit | $\mathrm{m}^3\,\mathrm{s}^{-1}$ | - |
| $Q_{\text{c}}$ | Critical water flux in a conduit | $\mathrm{m}^3\,\mathrm{s}^{-1}$ | 1.00 |
| $S$ | Cross-sectional area of conduits | $\mathrm{m}^2$ | - |
| $S_\infty$ | Far-field cross-sectional area of conduits | $\mathrm{m}^2$ | - |
| $\boldsymbol{v}$ | Ice velocity | $\mathrm{m\,s}^{-1}$ | - |
| $\boldsymbol{v}_{\text{b}}$ | Basal ice velocity | $\mathrm{m\,s}^{-1}$ | - |
| $v_0$ | Velocity threshold in the friction law | $\mathrm{m\,s}^{-1}$ | $9.51 \times 10^{-6}$ |
| $\boldsymbol{x}$ | Position | m | - |

## Appendix B:  The effective pressure near the grounding line: a boundary-layer analysis

In this appendix, we apply a boundary-layer analysis of the hydrology system close to the grounding line, and derive an approximate expression of the effective pressure in the area.

### B1  Problem statement

We consider a streamline of subglacial water parametrized by a parameter $s \in [0, s_{gl}]$, where $s = s_{gl}$ corresponds to the grounding-line position. All the parameters are fixed and the magnitude of the geometric potential gradient, $\Psi = \|\nabla \phi_0\|$, is assumed to be constant and known. The governing equations of the hydrology system are then expressed as

$$
\begin{cases}
N = \phi_0 - \phi, & \text{(B1a)} \\[2mm]
\partial_t S + \partial_s Q_w = \dfrac{\dot{M}}{\rho_w}, & \text{(B1b)} \\[2mm]
Q_w = -K S^\alpha |\partial_s \phi|^{\beta - 2} \partial_s \phi, & \text{(B1c)} \\[2mm]
\partial_t S = \|\boldsymbol{v}_b\| h_b + \dfrac{|Q_w \partial_s \phi|}{\rho_i \mathcal{L}_w} - 2 n^{-n} A L^2 |N|^{n-1} N, & \text{(B1d)}
\end{cases}
$$

where $\dot{M}$ is the net melt rate, associated with the amount of water that reaches the conduit. As boundary conditions, we require a zero water flux at the ice divide, i.e., $Q_w = 0$ at $s = 0$, and a continuity of the subglacial water pressure with the ocean water, i.e., $N = 0$ at $s = s_{gl}$. We consider hard beds, for which $L(S) = \sqrt{S}$. For canals, $L(S) = S/H_0$, the derivation and results are quite similar.

### B2  Dimensionless equations

We first make the problem and the unknowns dimensionless. We therefore write

$$
\hat{s} = \frac{s}{[s]}, \quad \hat{t} = \frac{t}{[t]}, \quad \hat{\phi} = \frac{\phi}{[\phi]}, \quad \hat{N} = \frac{N}{[N]}, \quad \hat{Q}_w = \frac{Q_w}{[Q_w]}, \quad \hat{S} = \frac{S}{[S]}. \tag{B2}
$$

The scales are chosen as follows. First, we set $[s] = s_{gl}$ and $[\phi] = \Psi[s]$. We further choose $[t]$ to be a time scale associated with ice flow. The other scales, $[N]$, $[Q_w]$, and $[S]$, are chosen such that

$$
\frac{[Q_w]}{[s]} = \frac{\dot{M}}{\rho_w}, \quad [Q_w] = K[S]^\alpha \Psi^{\beta-1}, \quad \frac{1}{\rho_i \mathcal{L}_w}[Q_w]\Psi = 2 n^{-n} A[S][N]^n. \tag{B3}
$$

This leads to the following dimensionless formulation of the problem:

$$
\begin{cases}
\eta \hat{N} = -\hat{s} - \hat{\phi}, & \text{(B4a)} \\[2mm]
\tau_1 \, \partial_{\hat{t}} \hat{S} + \partial_{\hat{s}} \hat{Q}_w = 1, & \text{(B4b)} \\[2mm]
\hat{Q}_w = -\hat{S}^\alpha |\partial_{\hat{s}} \hat{\phi}|^{\beta-2} \partial_{\hat{s}} \hat{\phi}, & \text{(B4c)} \\[2mm]
\tau_2 \, \partial_{\hat{t}} \hat{S} = \nu + |\hat{Q}_w \, \partial_{\hat{s}} \hat{\phi}| - \hat{S} |\hat{N}|^{n-1} \hat{N}, & \text{(B4d)}
\end{cases}
$$

for $0 < \hat{s} < 1$, with boundary conditions $\hat{Q}_w(\hat{s} = 0) = 0$ and $\hat{N}(\hat{s} = 1) = 0$. Four dimensionless ratios appear here:

$$\eta := \frac{[N]/[s]}{\Psi}, \quad \nu := \frac{\rho_i \mathcal{L}_w \|\boldsymbol{v}_b\| h_b}{[Q_w]\Psi}, \quad \tau_1 := \frac{[S][s]}{[t][Q_w]}, \quad \tau_2 := \frac{\rho_i \mathcal{L}_w [S][s]}{[t][Q_w][\phi]}. \tag{B5}$$

The first ratio compares the magnitude of the spatial variation of $N$ with respect to the geometric potential gradient $\Psi$. It thus

follows that if $\eta \ll 1$, then $\partial_s \phi \approx \Psi$, while for $\eta \gg 1$, $\partial_s \phi \approx -\partial_s N$. The second parameter compares the two possible terms that lead to an opening of the cavities or of the channels: it compares the opening due to sliding over bumps of the bedrock and the melt of the conduit boundaries. The two last ratios compare the characteristic times related to changes in the water flux and in the channel thickness with respect to the characteristic time of ice flow. In particular, if $\tau_1, \tau_2 \ll 1$, which we anticipate, then the time dependency disappears from the problem and the hydrological system is in a steady state.

In what follows, we drop the hats on the dimensionless variables to simplify the notations.

## B3  Outer solution

For commonly used parameters, we obtain $\eta, \tau_1, \tau_2 \ll 1$ and $\nu \sim 1$. This suggests to treat $\eta$, $\tau_1$, and $\tau_2$ as small parameters of the problem. The leading-order solution is then given by

$$Q_w = s, \quad S = s^{\frac{1}{\alpha}}, \quad \phi = -s, \quad N = s^{-\frac{1}{n\alpha}}(\nu + s)^{\frac{1}{n}}. \tag{B6}$$

This effective pressure has originally been obtained by Schoof (2010). It is such that $N(s = 1) = \nu^{1/n}$, so the Dirichlet boundary condition at the grounding line is not fulfilled. This suggests that there exists a boundary layer close to the grounding line, in which $N$ quickly decreases to reach a zero value (Figure B1a). We therefore refer to the leading-order solution (B6) as the outer solution, and the inner solution, associated with the boundary layer, must be found to obtain an acceptable expression of the effective pressure.

## B4  Inner solution

We first eliminate $Q_w$ and $S$ from the dimensionless system of equations (B4) to get the following equation for $N$ only:

$$\nu + s|1 + \eta \partial_s N| = s^{\frac{1}{\alpha}}|1 + \eta \partial_s N|^{\frac{1-\beta}{\alpha}}|N|^{n-1}N, \tag{B7}$$

for $0 < s < 1$ and with $N(s = 1) = 0$. The boundary layer at the grounding line suggests the introduction of a scaling in which $\partial_s N$ becomes of order $\mathcal{O}(1)$. We therefore introduce $\mathcal{X} = \eta^{-1}(1 - s)$ and rename $\mathcal{N} = N$. Then, at leading order,

$$\nu|1 - \partial_{\mathcal{X}} \mathcal{N}|^{\frac{\beta-1}{\alpha}} + |1 - \partial_{\mathcal{X}} \mathcal{N}|^{\frac{\beta+\alpha-1}{\alpha}} = |\mathcal{N}|^{n-1}\mathcal{N}, \tag{B8}$$

for $\mathcal{X} > 0$ and with $\mathcal{N}(\mathcal{X} = 0) = 0$. Because the inner solution must join the outer solution, we also have the compatibility condition $\mathcal{N} \to (\nu + 1)^{\frac{1}{n}}$ as $\mathcal{X} \to +\infty$. Finding the solution of this leading-order problem is not trivial because of its non-linearity. Nonetheless, we approximate it by an expression $\tilde{\mathcal{N}}$. We require that this approximate has the correct behavior over the boundaries of the boundary layer, that is, $\tilde{\mathcal{N}} \sim \mathcal{X}$ as $\mathcal{X} \to 0$ and $\tilde{\mathcal{N}} \sim (\nu + 1)^{\frac{1}{n}}$ as $\mathcal{X} \to +\infty$. We find that the following

expression is a good approximation of $\mathcal{N}$ for $\nu \lesssim 1$, see Figure B1b:

$$\tilde{\mathcal{N}} = \text{erf}\left[\frac{\sqrt{\pi}}{2}\frac{\mathcal{X}}{(\nu+1)^{\frac{1}{n}}}\right](\nu+1)^{\frac{1}{n}}, \tag{B9}$$

where $\text{erf}(x) = (2\pi)^{-1/2}\int_0^x \exp(-t^2)\,dt$ is the Gauss error function.

## B5   Composite solution

The composite solution, which is valid over the whole domain, is obtained by summing the inner and outer solutions and subtracting the overlap in the matching area. This leads to the following expression:

$$N(s) = \text{erf}\left[\frac{\sqrt{\pi}}{2}\frac{\eta^{-1}(1-s)}{(\nu+1)^{\frac{1}{n}}}\right](\nu+1)^{\frac{1}{n}} + s^{-\frac{1}{n\alpha}}(\nu+s)^{\frac{1}{n}} - (\nu+1)^{\frac{1}{n}}. \tag{B10}$$

It is possible to go back to the original variables to obtain the expression of the effective pressure as a function of the original parameters. To do so, we introduce

$$N_\infty(s) = \left[\frac{\rho_i\mathcal{L}_w\|\boldsymbol{v}_b\|h_b + (\rho_w^{-1}\dot{M}s)\Psi}{2\,n^{-n}\rho_i\mathcal{L}_w A(\rho_w^{-1}\dot{M}s)^{\frac{1}{\alpha}}K^{-\frac{1}{\alpha}}\Psi^{\frac{1-\beta}{\alpha}}}\right]^{\frac{1}{n}}, \tag{B11}$$

which is the outer solution written in its dimensional form. The effective pressure is then given by

$$N(s) = \text{erf}\left[\frac{\sqrt{\pi}}{2}\frac{\Psi s_{gl}}{N_\infty(s_{gl})}\left(1-\frac{s}{s_{gl}}\right)\right]N_\infty(s_{gl}) + N_\infty(s) - N_\infty(s_{gl}). \tag{B12}$$

Because $N_\infty$ does not change much over the boundary layer, the previous expression can be replaced by

$$N(s) = \text{erf}\left[\frac{\sqrt{\pi}}{2}\frac{\Psi s_{gl}}{N_\infty(s)}\left(1-\frac{s}{s_{gl}}\right)\right]N_\infty(s) \tag{B13a}$$

$$= \text{erf}\left[\frac{\sqrt{\pi}}{2}\frac{\phi_0(s)}{N_\infty(s)}\right]N_\infty(s). \tag{B13b}$$

This composite solution closely matches the numerical solution to the problem (Figure B1c).

## Appendix C:  Effect of the coupling frequency between the hydrological and ice-sheet models

Here, we investigate the effect of the coupling frequency between the hydrological and the ice-sheet models. We show results for both MISMIP and Thwaites setups, and show that the hydrological model must be updated at a frequency that is at least of the same order of magnitude as the update frequency of the ice-sheet model. As a consequence, no subglacial hydrology model, no matter how complex, can improve ice-sheet simulations involving grounding-line motion if it is not updated at a sufficiently high frequency. In particular, considering a hydrological model during the initialization of an ice-sheet model but not during a forward simulation is virtually useless as the impact of the hydrological model is then almost nonexistent.

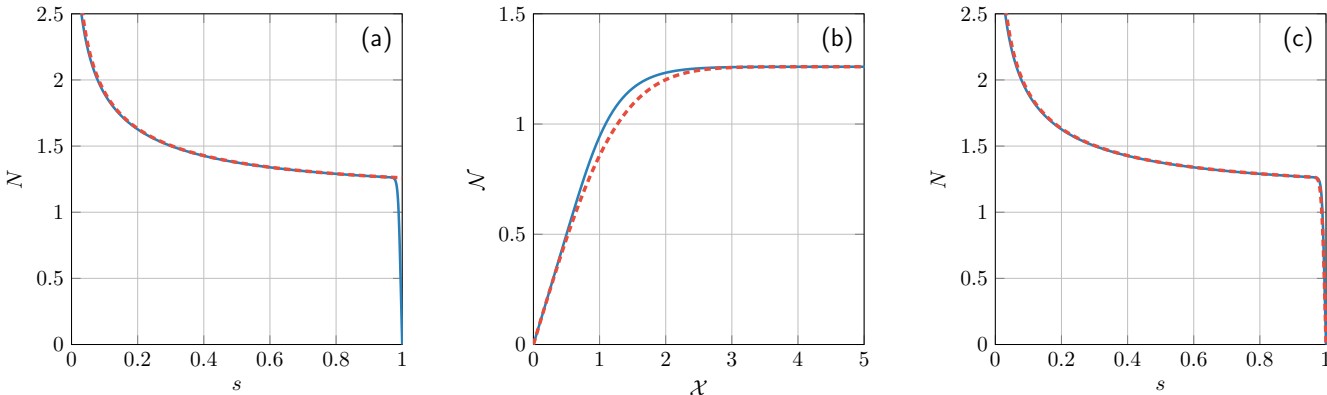

**Figure B1.** Inner, outer, and composite solutions of the dimensionless problem, for $\eta = 10^{-2}$ and $\nu = 1$. (a) Outer solution: numerical solution to the system of equations (B4) (continuous line) and leading-order solution (B6) to the outer problem (dashed line). (b) Inner solution: numerical solution to the equation (B8) (continuous line) and approximate solution (B9) (dashed line). (c) Composite solution: numerical solution to the system of equations (B4) (continuous line) and composite solution (B10) (dashed line).

## C1 MISMIP

Figure C1a shows the grounding-line position after the same forcing as the one that was considered in section 3, for various update frequencies of the hydrological model. If the hydrology model is not regularly updated, then the results resemble the no-subglacial-hydrology case NON. This last case is evidently not affected by the chosen time steps. A higher sensitivity of the subglacial hydrological model also requires higher update frequency rates.

## C2 Thwaites

We can make the same observations in Figure C1b as those made for the MISMIP configuration. A difference remains in the time scales considered for the two studies, and by the fact that HAB does not exhibits the largest sensitivity.

## Appendix D: Influence of the unconstrained parameters of the hydrological model

We performed a sensitivity analysis of the least constrained parameters of our model, i.e., $l_c$, $Q_c$, and $F_{\text{till}}$ (Figure D1). It can be observed that $l_c$ has only a limited effect for hard beds, while it has a more pronounced impact for soft beds. From equation (4), a change in $l_c$ results in a change in the water flux $Q_w$, which will be important if water flow transitions from an efficient to an inefficient flow (or the reverse). However, for hard beds, the entirely efficient or inefficient cases yield similar results (Figure 8b). On the contrary, for soft beds, the difference between the entirely efficient or inefficient cases is more pronounced (Figure 8b), and it follows that there is a stronger dependence with respect to $l_c$. For $Q_c$ and $F_{\text{till}}$, the impact is limited. Finally, it can be noted the spread in the results increase over a time.

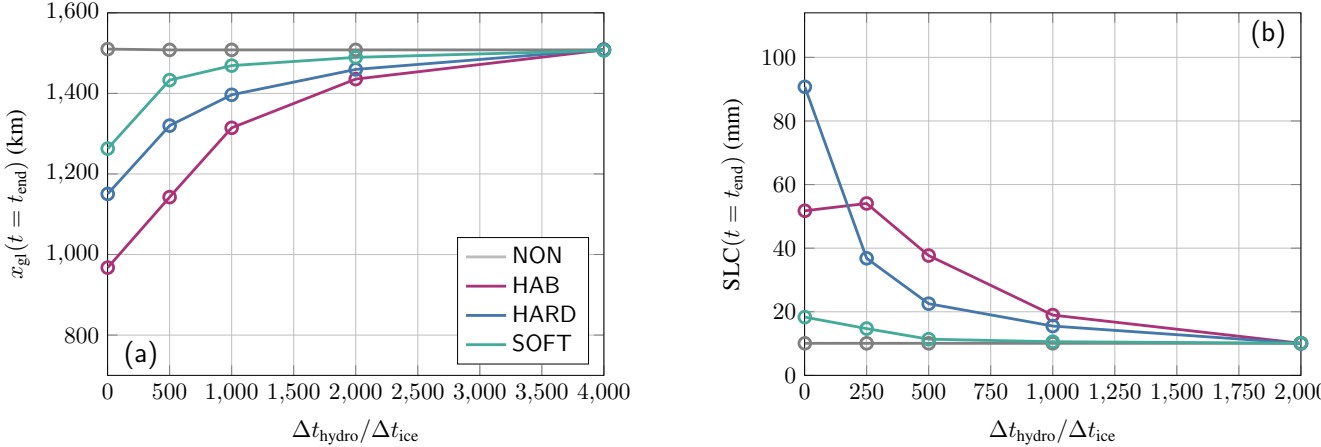

**Figure C1.** Effect of the coupling frequency between the hydrological and ice-sheet models: grounding-line position after the forcing as a function of the ratio between the time step $\Delta t_{\text{hydro}}$ of the hydrological model and the time step $\Delta t_{\text{ice}}$ of the ice-sheet model. Practically, we fix the ice-sheet time step and increase the time step of the hydrological model to obtain several values for $\Delta t_{\text{hydro}}/\Delta t_{\text{ice}}$. (a) MISMIP configuration: grounding-line position after the forcing. (b) Thwaites configuration: sea-level contribution after the forcing.

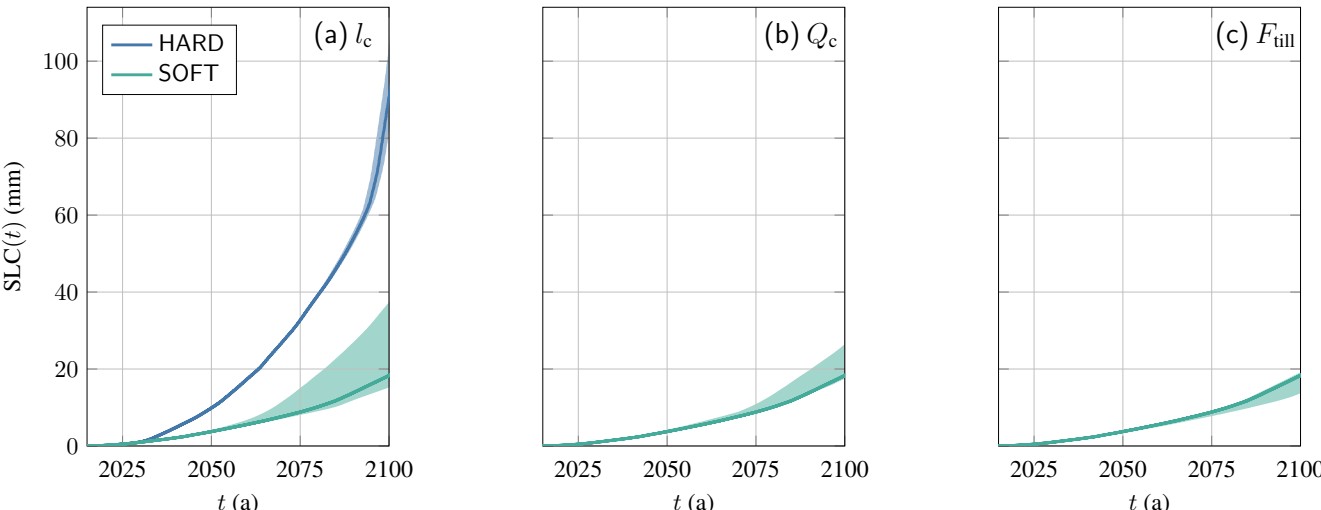

**Figure D1.** Sensitivity analysis of the results with respect to the parameters $l_c$, $Q_c$, and $F_{\text{till}}$. The set-up is the same as the one described in the forcing experiments over Thwaites (subsection 4.2; subglacial hydrology on homogeneous beds), except that different values of these parameters are chosen. The shaded areas correspond to the ranges $l_c \in [5, 15]\,\text{km}$, $Q_c \in [0.5, 1.5]\,\text{m}^3\,\text{s}^{-1}$, and $F_{\text{till}} \in [1, 2]$, and the lines correspond to the nominal values considered in the original experiment.

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
