# Peer review of "A fast and simplified subglacial hydrological model for the Antarctic Ice Sheet and outlet glaciers"

_EGUsphere, 2024_

## Referee Comment (RC1)

**Review of A fast and unified subglacial hydrological model applied to Thwaites Glacier, Antarctica, Kazmierczak et al.**

In this work, the authors develop a new hydrological model for Antarctica, and then apply it to some example cases, including modelling the retreat rate of Thwaites Glacier. I am reading the examples as test-cases of the model implementation, rather than fully fledged investigations into the likely future behaviour of Thwaites, and I appreciate that the abstract and conclusions respect this level of preliminarity (although the title might make one think otherwise).

The authors make some interesting modelling assumptions in the setup of the hydrology model, some of which are also found in Gowan et al (2023), a paper I will admit I was not familiar with. The current manuscript presents itself as not proposing too much beyond simplifications that are already present somewhere in the literature. However, given the number of different hydrological models currently out there, it would be good to compile a clear list of the simplifying assumptions at play in this work, so that future users can quickly assess if their use-case fits in this framework.

As I read it, the modelling assumptions are

- The hydrological system is always in steady state, i.e. the timescale of basal melt and channel development is fast compared to the timescale of forcing changes - likely a good assumption for Antarctica, less so for seasonal meltwater input in Greenland (so figure 7 seems a bit of a perverse/misleading test case - although here the timescale appears to be thousands of years, so perhaps this is not supposed to investigate seasonality, just a demonstration of the non-monotonicity of figure 5?)

- Gradients in hydraulic potential are primarily geometric, since $N$ is slowly varying, except at the grounding line, so when converting between $Q_w$ and $S$ using (5a), we can ignore gradients in $N$. This seems reasonable, but I don't quite understand the paragraph at l.126 - "so we choose not to do this" (do what?). Isn't $\boldsymbol{q}_w$ being computed directly from (2) without any specification of what gradient it is proportional to? Perhaps the way (2) is solved could be made more explicit - no expression for $\boldsymbol{q}_w$ is given in the manuscript.

  Note there is an extra factor of $S_\infty$ in (6a), but I assume this is just a typo, since the plots of $N_\infty$ in figure 5 show the correct behaviour from (5b).

- Close to the grounding line, $N$ must go to zero, so by eye, the authors pick an error function to approximate this transition. Per appendix B, this is not the solution to any local inner form of the ODE, but just a function that has the right gradient at the grounding line.

- Drainage density, regardless of the nature of the basal hydrology, is constant in space and time, and thus the flux through a drainage element is some constant, large, multiple of the flux through the area it represents. This one I find harder to wrap my head around, particularly since inefficient drainage is often imagined as slow flow everywhere (so what even is a drainage element in this case?) and models such as GlaDS and SHAKTI show dynamically evolving channel networks and drainage densities over time. This really is a big simplification, and the one that allows for the shift in scale, and I'm saddened that it is not discussed further (the choice of value for $l_c$, the drainage density, is not discussed at all).

- Effective pressure within the drainage elements (a small proportional of the domain) is equal to the effective pressure everywhere else - despite how strongly models that resolve the channels show them as being local lows in the hydraulic potential. (Not discussed)

- Specific choices about how $H$, $L$, and $S$ depend on the type of bed, which are well-discussed and clear.

- Specific choices about how $Q_\mathrm{w}$ depends on $S$ and $\nabla\phi$, which have quite a lot of precedent in the literature, although I might have expected a non-turbulent parametrisation for the inefficient system, and it's not clear why $K$ should be the same for all geometries.

I'm also confused about the basal melt production. No expression for the $\dot{m}_w$ in equation (3) is given, the term driving feedback between routing and meltwater production. The channelised version of the expression is given in (5b), but it's not clear if/how this is included in the routing algorithm. This feedback also seems to be missing in (B1b), with the meltwater input to the channel assumed constant (scaled to 1 in B4b) and not dependent on local melt.

I have not read too closely into the experiments, and model results, nor provided specific line-by-line minor comments, because I would like more clarity on the model setup first. I hope this is ok. I do think this is potentially quite an interesting approach to modelling Antarctic hydrology, but I would like to see more justification from the authors for the assumptions of their model.

---

## Referee Comment (RC2)

Review of "A fast and unified subglacial hydrological model applied to Thwaites Glacier, Antarctica" by Kazmierczak and others, submitted to *The Cryosphere*

Overview

This manuscript presents a subglacial hydrology model that represents inefficient and efficient subglacial drainage in the context of hard and soft beds, coupled to an ice dynamics model. The model is demonstrated with application to an idealized setting based on the MISMIP experimental setup and to Thwaites Glacier to investigate the influence of subglacial hydrology on its future behavior.

I am glad to see this work being done, combining different pieces of subglacial hydrology modeling in a way that is more practical for large-scale and long-term ice sheet simulations than many previous efforts. While I am enthusiastic about the paper's topic and findings, it will benefit from some revisions to strengthen it before publishing.

Please see the specific comments below. In general, the model description needs additional work for completeness and clarity, as already highlighted by another reviewer's comments. The structure of the paper could also be improved upon for easy navigation, to clearly indicate where results are presented versus experimental descriptions. The lengthy Discussion section would benefit from being broken up into subsections for each theme addressed within it.

As a final comment, much of the analysis of Thwaites behavior focuses on location and migration of the grounding line. How would this change by considering a grounding zone rather than a distinct grounding line? Some brief mention or discussion about this would be helpful.

I look forward to seeing this work being refined to make an impactful publication. It is an important effort to improve representation of subglacial hydrology in large ice-sheet models, and the work presented here is a great contribution toward this aim.

Specific Comments
Lines 40-42: It may be helpful to some readers to give example ranges of the typical temporal and spatial scales discussed here, for both hydrology and ice dynamics.

Line 49: Clarify what "various bed types" means.

Lines 55-56: Suggest changing wording to avoid using "evaluate" twice in the same sentence.

Figure 1: This figure could be combined with another figure as an inset.

Line 89: It would be useful to include a brief description of what you mean by 'efficient' and 'inefficient' drainage before this.

Line 91: How small is the local scale? Order of sub-meter, meter, tens of meters, hundreds of meters?

Line 99: The SHAKTI model also combines inefficient and efficient drainage, with a continuum approach. Sommers, A., Rajaram, H., and Morlighem, M.: SHAKTI: Subglacial Hydrology and Kinetic, Transient Interactions v1.0, Geosci. Model Dev., 11, 2955–2974, https://doi.org/10.5194/gmd-11-2955-2018, 2018.

Figure 2: I am slightly confused by this figure and the flow shown. A more thorough description of the coupling in the text would probably help.

Line 108: How cheap? Give some illustrative value to back up this claim, probably based on domain size, resolution, time step, simulation time, number of processors, wall-clock time.

Line 112: Depth-integrated subglacial water flux?

Lines 119-120: For completeness, describe how the melt rate due to dissipation ($m_w$) is calculated. Do you include this dissipation term everywhere? This is worth clarifying because of the legacy of models that only include it for channel components.

Lines 126-129: Intriguing to use the simple routing scheme – I'm interested to see the results that support the claim that $phi_o$ is approximately equal to $phi$ over most of the domain. Perhaps pointing to a figure would be good, rather than simply saying "in anticipation of what follows". It seems like a strange thing to want to decouple the water routing from effective pressure when you are interested in modeling subglacial hydrology, given that water flow is driven by gradients in potential, which obviously changes depending on effective pressure.
Line 132: How is $l_c$ chosen? How sensitive are results to this value?

Line 143: Do you always assume turbulent flow in the model?

Lines 149-150: Is the opening by sliding over obstacles the same for hard and soft beds? It isn't clear from this sentence whether the model treats these the same or differently, or if this means that the physical interpretation is simply different.

Lines 150-151: Why isn't melt opening associated with both inefficient and efficient drainage systems? Similarly to the previous comment, is this sentence purely commentary on physical interpretation, or describing a coded switch in melt equations applied to different parts of the model domain?

Lines 175-176: It would be helpful to justify the assumption that effective pressure is "fairly constant" far from the grounding line, perhaps with a plot either in the main text or in a supplement or appendix. How far from the grounding line?

Equation6: Are N_infinity and S_infinity the effective pressure and conduit cross-sectional area far from the grounding line? That's what I infer, but they should be explicitly defined.

Line 181: How close to the grounding line?

Section 2.2.3: How sensitive are results to these geometric assumptions (the relationships between L, H, and S, also the value of F_till)? These are nicely explained here, but are still mostly unconstrained by observations and are somewhat arbitrary, so it would be more thorough to consider their influence on model results.

Line 204: missing a period

Line 207: How is the critical water flux value Q_c selected?

Line 209: I am curious as to how confident we can be in prescribing which regions are hard bedded and which soft bedded, particularly as these can be highly heterogeneous spatially. Maybe this is coming later in the application to Thwaites.

Line 226: Do you mean "entirely efficient" (instead of "entirely effective")?

Line 229: Similarly, should this be "entirely inefficient"?

Line 229: It is not clearly justified why the dissipation term should be removed in the inefficient system. Is this based on similar earlier models that needed this for numerical stability? Is this necessary in your model formulation? I'm not convinced that it makes sense physically to ignore the dissipative contribution to melt if you can help it.

Lines 225-230: This section about switches imposed in the model needs to be clarified. It's great to represent inefficient and efficient systems and systems that don't fall cleanly into either category. But it is not entirely clear from reading what the thresholds are for selecting different forms of the equations. Are these manually set based on preference of the modeler and the problem of interest? Or are there criteria that automatically trigger these switches?

Lines 237-238: It would be helpful to comment on why Weertman was selected as the sliding law, and why a uniform value for the friction coefficient, and why that value. (You have to make some choices, just curious about the rationale behind these selections).

Line 238: should be "upper boundary condition" (singular, not plural for grammatical agreement)

Line 241: This is confusing about the spatially variable friction coefficient used here, when it was just stated in the previous paragraph that a uniform friction coefficient was used.

Line 246: So did N change throughout the inversion here? A brief description of that would help clarify this, i.e. what initial distribution of N was assumed, and how was it altered through the iterative nudging process?

Line 257: Some observations have suggested low effective pressure in the interior. Can you comment on this here or elsewhere?

Line 263: This statement about the default switch between efficient and inefficient drainage equations would be helpful to include above (see comment about Lines 225-230).

Line 279: With what size time step is the ice sheet model run for 20,000 years? Is the hydrology model also run for 20,000 years?

Line 280: It would be helpful to include a brief reminder of how the hydrology and ice sheet models are coupled here.

FIgure 7: Why is the sliding velocity for entirely inefficient not included in panel c? It would also be useful to plot the effective pressure response for entirely efficient and entirely inefficient in panel b to help strengthen your points about the importance of the switching behavior.

Lines 341-342: What is the criterion to be considered a collapse?

Sections 4.1-4.3: The structure of this section needs some improvement. The titles of 4.2 and 4.3 may be renamed to make clear that the first section is the experimental description for Thwaites, and the second and third sections are presenting results. I was a bit confused by this structure while reading. The information about the threshold for hard-to-soft transition in line 328 seems to be repeated in line 352. We also seem to be missing information in the experimental setup on Thwaites about model resolution(s) and time step size(s), which would be interesting to know.

Discussion: I recommend separating this into some subsections with corresponding headings.

Tables A1 and A2 could be combined into a single table.

---

## Referee Comment (RC3)

**Review of "A fast and unified subglacial hydrological model applied to Thwaites Glacier, Antarctica"**

By  Kazmierczak et al.

Submitted to The Cryosphere

In this paper, the authors present a novel subglacial hydrology model that includes efficient and inefficient drainage systems on bed rheologies that range continuously from soft to hard. The model can be run on the same spatial scales as ice flow, which makes the model computationally efficient. The model is applied to Thwaites Glacier in Antarctica and the response to present day climate forcings is analyzed for hard, soft, and mixed beds for efficient and inefficient drainage systems.

I think the development of a model with these capabilities is an exciting and important step in modeling coupled ice flow and subglacial hydrology on the ice sheet scale. There are a few things that I believe should be clarified and adapted in the writing before publication.

**General comments:**

Although assuming hydrologic equilibrium are most likely appropriate for Antarctica, it should be made clear that this model would not be appropriate for modeling places with more highly variable water fluxes such as mountain glaciers or the Greenland ice sheet (or atleast the margins) where the subglacial hydrological system is often not in equilibrium due to external meltwater input which varies on much shorter timescales than changes in ice geometry. There could be an argument that this model could be applied to the Greenland Ice Sheet on long timescales (that neglect short term changes resulting from subglacial hydrology), but this would need to be thoroughly addressed and justified. Add to the abstract and conclusion that this model is appropriate for modeling subglacial hydrology where changes in water flux are on the same timescales as changes in ice geometry.

Does the thermomechanical ice flow model determine where there is basal melting? If so, does this evolve in time? How is water routed through areas with a frozen bed? Is there refreezing? Elaborate more on this, perhaps in section 2.1.

There are a few places in the manuscript where references are made to literature with different timescales and rates of change being discussed than in this paper. In this model, the assumptions around hydrologic equilibrium (melting balances closure in Eq. 5b) imply that the changes in time in the subglacial hydrologic system are small enough to neglect ($dS/dt = 0$). However, the manuscript contains references to Schoof (2010) and Iken (1981) which are specifically referring to when $dS/dt$ is not zero. In line 267, there is a reference to Schoof (2010) and the importance of the meltwater variability rather than meltwater input, but these processes are time-dependent changes that occur on timescales of hours to days, at most months. From

Schoof (2010), "Further acceleration must then be driven instead by **short-term temporal variability** in water supply." In this model, it is assumed to be in steady state which means that these short term increases in water pressure due to varying water flux on short timescales that result in conduit growth are inherently neglected. The same is true for the reference in line 292 in the manuscript, which I believe is referring to the following statement in Iken (1981). "It has been seen that the effect of a water pressure pw on the sliding velocity is largest at the instant of separation and then gradually decreases **until steady cavities have formed**." This is again talking about when the system is not in steady state, unlike the model presented here. To be clear, I think the assumption of steady-state is reasonable given the context of Antarctica, where changes in water flux occur on long-timescales, but these references and associated statements (specifically noted in the line-by-line comments) are not applicable in this context.

**Line-by-line comments:**

Line 5: This is worded strangely. Perhaps, "We find that accounting for subglacial hydrology in the sliding law accelerates the grounding line retreat of Thwaites Glacier under present-day climatic conditions." or something similar .

Line 13: 'behavior' since using American spelling elsewhere.

Fig 1. Figure could be made part of Fig. 9 or moved to before Fig. 9, but I don't feel strongly about this.

Line 71: introduce variable before introducing product of variables "... N is the effective pressure, C is a friction coefficient limiting the shear stress to a maximum plastic value CN, .."

Line 90: Replace "that is much smaller than the global one" with " that is on the order of meters" or what the scale actually is.

Line 93: The term 'conduit' may be confusing, as it is often synonymous with channels and efficient drainage in the subglacial hydrology literature. Although, I don't have a better suggestion…

Lines 105 - 106: Bueler and van Pelt (2015) ran the model efficiently on the Greenland ice sheet, but these statements are generally true.

Line 111: Elaborate on what these conditions are. "This domain evolves over time according to internal and external conditions."

Eq 3: Lw is confusing with L being the length of conduit. Use cursive L or something else to denote latent heat.

Line 120-123: This assumption is only reasonable in Antarctica and locations where changes in water flux are only through changes in ice geometry which happen on long time scales. Make this more clear in the text.

Line 128: Delete "anticipating what follows".

Fig 4: This is how the channel might look in theory. In the caption of Fig 4, it should also be stated that this is what you are trying to capture and not the actual geometry you have described in your model (which is square channel?).

Line 220-224: Elaborate on how this relates to theory and observations. Are your model results what we expect for these cases?

Fig 5:

Wouldn't we expect much higher effective pressures for that high of water flux in an efficient system? Or is this because it is the average of effective pressure over a larger scale than the channel? This could be made more clear either in the text or in the figure caption. On first glance at the figure, I find this result surprising.

On a related note, do you ever observe this high of water fluxes in channels far from the grounding line in the model? How exactly is the water fluxes on the local scale in the efficient system related to the effective pressure which is an average over a larger area presumably?

On what scale are these calculated?

Line 226-229: You mean efficient/inefficient, not effective/ineffective. The system still transports water, so it isn't ineffective. It just doesn't transport it efficiently.

A discussion about how effective pressure behavior differs near the grounding line and far from the grounding line would be helpful.

Line 236: Does this mean the whole bed is temperate in this experiment since melting is uniform?

Fig 6b. Why does the effective pressure go down by 2 MPa in both the soft and hard bed cases at x=0?

Fig 7: Are the light pink and red areas in (a)-(c) related to the colors in (d)? If so, it is not clear how. Add something to the caption about this (or remove from the background?).

Fig. 9a: It would be helpful to have a sense of scale in the 2D either by adding a scale bar or axis.

Line 267: This does not apply on the timescales you are analyzing. The theory from Schoof (2010) on meltwater variability is referring to time-dependent changes that occur on timescales of hours to days, at most months. In this model, it is assumed to be in steady state which means that these short term increases in water pressure due to varying water flux are inherently neglected.

Line 282: What do you mean by "the latter". If referring to a smaller ice sheet results in grounding line retreat, replace "the latter" with "consequently" or similar. Both slower velocities and a smaller ice sheet can result in grounding line retreat.

Line 318: This is not a unit of mass, but a unit of length. Rephrase to say something like, "Note that the mass loss for the NON experiment results in sea level rise on the order of 10 mm by 2100…"

Line 341: All of the models result in the collapse of Thwaites within how many years?

Line 392: As mentioned before, I have questions about how the assumption of steady state allows you to relate to the time variability of other work such as in the comment that follows. "This observation aligns with the work of Iken (1981), specifying that the highest velocities is not observed where effective pressures is lowest, but rather when cavities enlarge due to an increase in subglacial water pressure." I think this statement should be removed.

Line 396: How does a lower effective pressure in the soft bed system slow down grounding line retreat? Maybe I am missing something.

Line 434: add "(retrograde)".

Line 440: "(prograde)"

Line 465: add 'considering' or 'modeling' so that it reads "considering subglacial hydrology enhances the ice-sheet response to sliding".

Line 467: This reference to Schoof (2010) is appropriate.

Line 470: Making the connection to changes when the system is explicitly not in steady state ("when basal cavities are growing") does not make sense here since you assume steady state and therefore neglect time-dependent changes in cavity growth.

---

## Author Comment (AC1)

**Response to Referee 1 on "A fast and unified subglacial hydrological model applied to Thwaites Glacier, Antarctica" by Kazmierczak, Gregov, Coulon & Pattyn.**

Dear Referee,

We would like to thank you for the time you have already devoted to reviewing our manuscript. In order to clarify some elements and to enable you to continue your review, we provide hereafter additional comments related to the assumptions made within the original manuscript. Our comments are written in blue. The revised manuscript will be modified so that this information is more complete and clear.

Best regards,

On behalf of the authors,
Thomas Gregov

**Response to the Referee's comments**

In this work, the authors develop a new hydrological model for Antarctica, and then apply it to some example cases, including modelling the retreat rate of Thwaites Glacier. I am reading the examples as test-cases of the model implementation, rather than fully fledged investigations into the likely future behaviour of Thwaites, and I appreciate that the abstract and conclusions respect this level of preliminarity (although the title might make one think otherwise).

We agree with the Referee – we will change the title in order to be more generic. Thwaites is used as a particularly interesting test case, as it is thought to be composed of both hard and soft regions.

The authors make some interesting modelling assumptions in the setup of the hydrology model, some of which are also found in Gowan et al. (2023), a paper I will admit I was not familiar with. The current manuscript presents itself as not proposing too much beyond simplifications that are already present somewhere in the literature. However, given the number of different hydrological models currently out there, it would be good to compile a clear list of the simplifying assumptions at play in this work, so that future users can quickly assess if their use-case fits in this framework.

We thank the Referee for the suggestion of making the assumptions clearer. As those are not necessarily common within other hydrological models, this should indeed help potential readers identify more easily whether or not our model is suited for specific applications. We will also clarify the differences with the assumptions made in Gowan et al. (2023).

As I read it, the modelling assumptions are

- The hydrological system is always in steady state, i.e. the timescale of basal melt and channel development is fast compared to the timescale of forcing changes - likely a good assumption for Antarctica, less so for seasonal meltwater input in Greenland (so figure 7 seems a bit of a perverse/misleading test case - although here the timescale appears to be thousands of years, so perhaps this is not supposed to investigate seasonality, just a demonstration of the non-monotonicity of figure 5?).

  That is correct. Our model is meant for Antarctica, and should probably not be used for Greenland. Figure 7 is associated with subsection 3.2 ('the efficient to non-efficient switch'), so the role of this figure is indeed to investigate the non-monotonicity of the relation between the flux and the effective pressure, rather than the effect of seasonality.

- Gradients in hydraulic potential are primarily geometric, since $N$ is slowly varying, except at the grounding line, so when converting between $Q_w$ and $S$ using (5a), we can ignore gradients in $N$. This seems reasonable, but I don't quite understand the paragraph at l.126 - "so we choose not to do this" (do what?). Isn't $\boldsymbol{q}_w$ being computed directly from (2) without any specification of what gradient it is proportional to? Perhaps the way (2) is solved could be made more explicit - no expression for $\boldsymbol{q}_w$ is given in the manuscript.

This requires a bit of explanation. Equation (2) is given by

$$
\left\{
\begin{array}{ll}
\nabla \cdot \boldsymbol{q}_w = \dfrac{\dot{m}}{\rho_w}, & \text{in } \Omega, \\[2mm]
\boldsymbol{q}_w \cdot \boldsymbol{n} = 0, & \text{on } \Gamma_d.
\end{array}
\right.
\tag{R1a}
$$
$$
\tag{R1b}
$$

By itself, this system of equations alone cannot be used to determine the subglacial water flux $\boldsymbol{q}_w$. Indeed, if $(q_x, q_y)$ are the components of $\boldsymbol{q}_w$, then equation (R1a) is explicitly given by

$$
\frac{\partial q_x}{\partial x} + \frac{\partial q_y}{\partial y} = \frac{\dot{m}}{\rho_w}, \quad \text{in } \Omega,
\tag{R2}
$$

that is, one equation for two unknowns. To determine $\boldsymbol{q}_w$, we therefore need some additional information. Physically, equation (R1a) constrains how the subglacial water can evolve: in each arbitrary region (and, in particular, in each grid cell), there is an imbalance between the inflow and outflow of subglacial water, the imbalance being caused by the basal-melt term $\dot{m}/\rho_w$. However, (R1a) does not specify where (i.e., in which direction) the excess of water goes. It is for that reason that we need to specify the direction of $\boldsymbol{q}_w$.

This can be made more explicit by considering an integral version of (R1a): integrating this equation over a grid cell $\omega$ of the mesh, one gets

$$
\psi_{out} = \psi_{in} + \int_\omega \frac{\dot{m}}{\rho_w} \, d\omega,
\tag{R3}
$$

in which $\psi_{in}$ and $\psi_{out}$ are the inflow and outflow integrated scalar fluxes, respectively. Explicitly, these are given by

$$
\psi_{in} = -\int_{\partial\omega_{in}} \boldsymbol{q}_w \cdot \boldsymbol{n} \, dl \quad \text{and} \quad \psi_{out} = \int_{\partial\omega_{out}} \boldsymbol{q}_w \cdot \boldsymbol{n} \, dl,
\tag{R4}
$$

with $\partial\omega_{in}$ (resp. $\partial\omega_{out}$) the part of cell boundary associated to an inflow of subglacial water, i.e., $\boldsymbol{q}_w \cdot \boldsymbol{n} < 0$ (resp. $\boldsymbol{q}_w \cdot \boldsymbol{n} > 0$). Note that, by construction, $\psi_{in}$ and $\psi_{out}$ are non-negative. Equation (R3) can then be iteratively solved by determining the value of the scalar flux in a cell, and then 'propagating' the outgoing flux to neighboring cells that are in the direction of $\boldsymbol{q}_w$. This is the method we follow to solve this equation; specifically, we use the method of Le Brocq et al. (2009), as stated in the original manuscript. This method takes the form of a routing algorithm, and is based on earlier developments (Budd and Warner, 1996; Le Brocq et al., 2006). It has been used in the context of the computation of subglacial water flow in other studies, e.g., in Pattyn (2010) and in Kazmierczak et al. (2022).

It remains to clarify the question of the direction of $\boldsymbol{q}_w$. Physically, this direction is the same as the direction of the hydraulic potential gradient $\nabla\phi$, as water flows from high to low values of the hydraulic potential $\phi$. We also have

$$
\nabla\phi = \nabla\phi_0 - \nabla N,
\tag{R5}
$$

in which, at this point in the manuscript, $\nabla\phi_0$ is known, but $\nabla N$ is not. However, we anticipate the result that $\nabla\phi \approx \nabla\phi_0$ in most of the domain. Therefore, we state that the

direction of $\boldsymbol{q}_\mathrm{w}$ is the one of $\nabla\phi_0$, and not the one of $\nabla\phi$. We therefore choose not to include the term $\nabla N$ in the computation of the direction of the flow.

Note there is an extra factor of $S_\infty$ in (6a), but I assume this is just a typo, since the plots of $N_\infty$ in figure 5 show the correct behaviour from (5b).

Indeed, thanks!

- Close to the grounding line, $N$ must go to zero, so by eye, the authors pick an error function to approximate this transition. Per appendix B, this is not the solution to any local inner form of the ODE, but just a function that has the right gradient at the grounding line.

  It is true that the error function is not the analytical solution to the inner problem. Nonetheless, it is reasonably close to the numerical solution to that problem (see figure B1(b)). Hence, its use has the advantage of leading to a practical closed-form expression: the error function is easy to compute and available in most scientific computing libraries. At the same time, its use leads to a relatively small error.

- Drainage density, regardless of the nature of the basal hydrology, is constant in space and time, and thus the flux through a drainage element is some constant, large, multiple of the flux through the area it represents. This one I find harder to wrap my head around, particularly since inefficient drainage is often imagined as slow flow everywhere (so what even is a drainage element in this case?) and models such as GlaDS and SHAKTI show dynamically evolving channel networks and drainage densities over time. This really is a big simplification, and the one that allows for the shift in scale, and I'm saddened that it is not discussed further (the choice of value for $l_\mathrm{c}$, the drainage density, is not discussed at all).

  We agree with the Referee that this is a strong assumption in the model, and we apologize for not discussing it further.

  In our model, we took $l_\mathrm{c} = 10\,\mathrm{km}$, which is similar to the value used in Gowan et al. (2023). Although the choice of taking a value of $l_\mathrm{c}$ that is both uniform and constant is made for simplicity, the value chosen in Gowan et al. (2023) is based on some observations of the distance between eskers. The goal of our study is to provide a model that is capable of representing the essential physics of different types of hydrological components (efficient/inefficient, hard/soft) at a relatively large scale, i.e., at a resolution that is of the order of at least a few kilometers. In that sense, we use a 'lumped-element' approach, i.e., we parametrize complex and distributed flows using simplified relations that aim to reproduce the overall behavior of the system. With that in mind, it seems to us that developing models that are able to incorporate these different types of hydrological components is particularly important, which is why we have focused our study on including these, rather than prescribing or tuning parameters. This is corroborated by our results, which are noteworthy; they suggest that, even for a a relatively simple model (e.g., with a constant and uniform value for $l_\mathrm{c}$), (i) including subglacial hydrology that is coupled to ice flow greatly increases the sensitivity of marine ice sheets to external forcings and (ii) there is a strong dependency on the efficiency of the system and on the type of bed.

  Nonetheless, we acknowledge that the drainage density, in general, should not be a constant in space or in time. As such, it could be particularized to the type of drainage system or the type of bed, e.g., by tuning it against a high-resolution hydrological model such as GlaDS or SHAKTI. We leave this for future work. We still want to emphasize that the value of $l_\mathrm{c}$ does not change the fundamental dynamics that govern water flow, in

the sense that, for example, the effective pressure is an increasing or decreasing function of the subglacial water flux is mainly unchanged if the value of $l_c$ is modified. In the end, friction coefficients (and, possibly, other parameters), will be tuned so that some computed quantities fit observations. Again, with that in mind, it seems more important to obtain the right dynamical relations between the variables of the system rather than determining exactly one of the parameters, as many choices of parameters can potentially lead to a good fit with observations. By contrast, fixing correctly parameters does not guarantee the correct relation between the effective pressure and the flux, for example.

This discussion should also, hopefully, respond to the Referee's question about the nature of an inefficient drainage component: it is a component that is such that the relation between $N$ and $Q$ in a grid cell is the one prescribed by an inefficient system (i.e., equation (6) with $H \sim \sqrt{S}$); no more, no less.

- Effective pressure within the drainage elements (a small proportion of the domain) is equal to the effective pressure everywhere else - despite how strongly models that resolve the channels show them as being local lows in the hydraulic potential. (Not discussed)

  We agree with the Referee's comment that our model is not able to include any spatial variation of the effective pressure at a scale that is smaller than the grid scale. That is an inherent limitation of the approach that we have pursued. Although this has been mentioned in the discussion section, we will discuss it in more detail in the revised manuscript.

- Specific choices about how $H$, $L$, and $S$ depend on the type of bed, which are well-discussed and clear.

  Ok.

- Specific choices about how $Q_w$ depends on $S$ and $\nabla\phi$, which have quite a lot of precedent in the literature, although I might have expected a non-turbulent parametrisation for the inefficient system, and it's not clear why $K$ should be the same for all geometries.

  Here we have followed Schoof (2010) and Gowan et al. (2023). We recognize that there is quite an important missing portion of the literature that we have omitted in our paper, and will include the relevant references in the revised version.

  Fundamentally, there is no reason why $K$ should be the same for all geometries. Our rationale is the same as the one described earlier for the choice of values for $l_c$: in our paper, we focused on the right relations rather than the right parameters, although we acknowledge that the choice of parameters could, and should, be studied in more details in future work.

I'm also confused about the basal melt production. No expression for the $\dot{m}_w$ in equation (3) is given, the term driving feedback between routing and meltwater production. The channelised version of the expression is given in (5b), but it's not clear if/how this is included in the routing algorithm.

Indeed, the expression for $\dot{m}_w$ is missing from the manuscript. It is given by

$$\dot{m}_w = \frac{|\boldsymbol{q}_w \cdot \nabla\phi|}{L_w} = \frac{1}{l_c}\frac{Q_w\|\nabla\phi\|}{L_w} = \frac{1}{\Delta x}\left[\frac{\Delta x}{l_c}\frac{Q_w\|\nabla\phi\|}{L_w}\right]. \tag{R6}$$

Here, $\Delta x/l_c$ is the number of conduits per cell, and $Q_w\|\nabla\phi\|/\rho_i L_w$ is the melt production per conduit. In our numerical experiments, the term $\dot{m}_w$ was found to be relatively small in the

total melt production.

This feedback also seems to be missing in (B1b), with the meltwater input to the channel assumed constant (scaled to 1 in B4b) and not dependent on local melt.

Indeed, we have assumed a constant meltwater input. In fact, it turns out that the local melt is not an important contribution to the total melt in a channel. We refer to Lu and Kingslake (2023) – in which the authors describe a model analogous to ours – that show through a scaling analysis that this term can be dropped at leading order for marine ice sheets.

I have not read too closely into the experiments, and model results, nor provided specific line-byline minor comments, because I would like more clarity on the model setup first. I hope this is ok. I do think this is potentially quite an interesting approach to modelling Antarctic hydrology, but I would like to see more justification from the authors for the assumptions of their model.

We thank the Referee for their encouraging comment, and hope that the justifications provided here will allow them to continue their review.

**References**

Budd, W. F. and Warner, R. C. (1996). A computer scheme for rapid calculations of balance-flux distributions. *Annals of Glaciology*, 23:21–27.

Gowan, E. J., Hinck, S., Niu, L., Clason, C., and Lohmann, G. (2023). The impact of spatially varying ice sheet basal conditions on sliding at glacial time scales. *Journal of Glaciology*, 69(276):1056–1070.

Kazmierczak, E., Sun, S., Coulon, V., and Pattyn, F. (2022). Subglacial hydrology modulates basal sliding response of the antarctic ice sheet to climate forcing. *The Cryosphere*, 16(10):4537–4552.

Le Brocq, A., Payne, A., Siegert, M., and Alley, R. (2009). A subglacial water-flow model for West Antarctica. *Journal of Glaciology*, 55(193):879–888.

Le Brocq, A. M., Payne, A. J., and Siegert, M. J. (2006). West Antarctic balance calculations: Impact of flux-routing algorithm, smoothing algorithm and topography. *Computers & Geosciences*, 32(10):1780–1795.

Lu, G. and Kingslake, J. (2023). Coupling between ice flow and subglacial hydrology enhances marine ice-sheet retreat. *EGUsphere [preprint]*.

Pattyn, F. (2010). Antarctic subglacial conditions inferred from a hybrid ice sheet/ice stream model. *Earth and Planetary Science Letters*, 295(3–4):451–461.

Schoof, C. (2010). Ice-sheet acceleration driven by melt supply variability. *Nature*, 468(7325):803–806.

---

## Author Comment (AC2)

**Addundum to the Response to Referee 1 on "A fast and unified subglacial hydrological model applied to Thwaites Glacier, Antarctica" by Kazmierczak, Gregov, Coulon & Pattyn.**

Dear Referee,

Following the comments of all the Referees, we have made changes and improvements to several sections of our manuscript. In this addendum to our previous response, we would like to describe in more detail the additional changes that are relevant to your comments, as we believe they may be valuable. This addendum is divided into four parts:

   A. Assumptions of the hydrological model

   B. Clarification of the expression of the melt rate

   C. Form of the Darcy-like flow equation

   D. Justification and impact of the drainage density

The modifications and additions to the initial manuscript are written in blue.

Best regards,

On behalf of the authors,
Thomas Gregov

**A. Assumptions of the hydrological model**

In the new structure of the model description, we have tried to improve the description of the hydrological model by stating the main assumptions prior to the derivation of the model. This has resulted in an additional subsection to the Model section, which is now structured as follows:

   1. Ice-flow model

   2. Hydrological model

     2.1. Simplifying assumptions
     2.2. Subglacial water routing
     2.3. Subglacial effective pressure
     2.4. Bed rheology

Specifically, the following discussion has been added to the new 'Simplifying assumptions' subsection:

*"The key simplifying assumptions are given by the following:*

   *1. There is limited temporal melt variability so that the hydrological system is in a quasi-static equilibrium with respect to the ice-sheet geometry. Therefore, changes in ice geometry will be the main driver for changes in subglacial water variability (both spatial and temporal).*

   *2. A few kilometers upstream of the grounding line, the hydraulic gradient is approximated by the geometric gradient.*

   *3. The drainage density is uniform and the effective pressure is not calculated at a sub-grid level.*

*The first assumption is based on several studies of subglacial hydrology in Antarctica (Le Brocq et al., 2009; Pattyn, 2010; Kazmierczak et al., 2022), among others, that demonstrate that — contrary to the Greenland ice sheet— there is limited surface meltwater infiltration. Hence,*

*changes in hydrology are primarily due to changes in ice geometry. Since the time scales associated with water flow are much smaller than those associated with ice flow, subglacial hydrology automatically adapts to any change in ice geometry and reaches the associated equilibrium. The second assumption is motivated by a scaling analysis through an estimation of the dimensionless ratio $\eta := [\nabla N]/[\nabla \phi_0]$, where $[\nabla N]$ is the scale of the spatial gradients for the effective pressure and $[\nabla \phi_0]$ is the characteristic scale for the geometric potential gradient. For the former we take $[\nabla N] = [N]/[x]$, with $[N] = 1\,MPa$ and $[x] = 10^3\,km$. For the latter we take $[\nabla \phi_0] = 5 \times 10^{-2}\,MPa\,km^{-1}$, which is a plausible value for ice sheets (Hewitt, 2011). This results in $\eta = 2 \times 10^{-2} \ll 1$, suggesting that $\|\nabla N\| \ll \|\nabla \phi_0\|$ and $\nabla \phi \approx \nabla \phi_0$. We further note that profiles obtained with a high-resolution subglacial hydrology model suggest that $\nabla \phi$ and $\nabla \phi_0$ have a correlation of at least $\sim 80\%$ for a region that is several kilometers upstream of the grounding line (see Supplementary Material S1). Finally, the third assumption follows from our modeling approach, where we do not describe the effective pressure at the sub-grid scale and where we assume the same number of conduits in each grid cell, similar to Gowan et al. (2023).*"

The Supplementary Material S1 refers to the assessment of the assumption that $\nabla \phi \approx \nabla \phi_0$ outside the vicinity of the grounding line based on data. Here is the content of this addition to the supplementary materials:

*"Here, we provide additional data to underpin the validity of the assumption that $\nabla \phi \approx \nabla \phi_0$ outside the range of influence of the grounding line, which is a few kilometers from it. Since there are no direct observations of the effective-pressure field in Antarctica, we have to rely on high-resolution models. A first test case comes from Lu and Kingslake (2023) who uses a high-resolution model that couples ice-sheet dynamics and subglacial hydrology for hard beds. Potential limitations of that study is that it considers a flow line and a smooth bedrock. The assumption that $\nabla \phi \approx \nabla \phi_0$ a few kilometers upstream of the grounding line is confirmed numerically (Figure R1).*

[Figure]

(a) Numerical results.    (b) Computed gradients.

Figure R1: Data derived from Figure 4 of Lu and Kingslake (2023).

*A second test case comes from Hager et al. (2022) who applied the high-resolution model MALI (Hoffman et al., 2018) to Thwaites Glacier. They also consider a hard-bed hydrology. The computed effective pressures along a center-line transect are shown in Figure R2. Note that the signals are much more noisier compared to the first test case. This noise can be attributed to the model resolution, but also to the presence of localized hydrological features that cross the center-line transect at which the effective pressures are evaluated, therefore resulting in very localized variations. However, we observe a good correlation between $\partial_s \phi$ and $\partial_s \phi_0$ out of the*

*vicinity of the grounding line (Figure R2): $\sim 80\%$ over the range $[10, 400]$ km, suggesting that the assumption that $\nabla \phi \approx \nabla \phi_0$ is valid in this region."*

[Figure]

(a) Numerical results.

(b) Computed gradients.

Figure R2: Data derived from Figure 8 of Hager et al. (2022).

**B. Clarification of the expression of the melt rate**

The expression for $\dot{m}_{\mathrm{w}}$ was missing from the manuscript. We have modified the introduction of the melt rate as follows:

*"The latter is computed from the energy balance within the ice sheet and includes effects of geothermal heat flux, frictional heating due to the motion of both ice and subglacial water, and thermal conduction, i.e.,*

$$\dot{m} = \frac{G + \boldsymbol{\tau}_{\mathrm{b}} \cdot \boldsymbol{v}_{\mathrm{b}} - q_T}{L_{\mathrm{w}}} + \dot{m}_{\mathrm{w}}, \tag{R1}$$

*where $G$ is the geothermal heat flux, $q_T$ is the thermal conduction flux, $L_{\mathrm{w}}$ is the latent heat for ice, and $\dot{m}_{\mathrm{w}} = |\boldsymbol{q}_{\mathrm{w}} \cdot \nabla \phi|/L_{\mathrm{w}}$ is the water melt rate due to the dissipated energy from the subglacial water conduits. However, we do not include this last term in our simulations as it was found to be negligible compared to the other terms."*

**C. Form of the Darcy-like flow equation**

Regarding of the use of the Darcy–Weisbach equation equation for the relation between $Q_{\mathrm{w}}$, $S$, and $\nabla \phi$, we now mention that we follow the approach taken in Schoof (2010), although we acknowledge the possibility of considering other parametrizations:

*"Following Schoof (2010), we assume a turbulent flow, with $\alpha = 5/4$, $\beta = 3/2$, and $K = (2/\pi)^{1/4}\sqrt{(\pi + 2)/(\rho_{\mathrm{w}}f)}$, where $f$ is a friction coefficient (e.g., Clarke, 1996). Other choices have been considered for subglacial hydrology in the literature; we refer to Hewitt (2011) and Werder et al. (2013) for laminar parametrizations, and to Hill et al. (2023) for a discussion of the transition between laminar and turbulent flows and their range of validity."*

**D. Justification and impact of the drainage density**

We have made several changes to the manuscript with respect to the uniform drainage density. It is now explicitly mentioned in the additional 'Simplifying assumptions' subsection (see comment in part A of this addendum).

Moreover, we have added the following after sentence the introduction of the parameter $l_{\mathrm{c}}$:

*"We take $l_c = 10\,km$, which is similar to the value considered in Gowan et al. (2023) based on observations of distances between eskers formed under the Laurentide Ice Sheet (Storrar et al., 2014)."*

Finally, we have added an appendix with a sensitivity analysis with respect to $l_c$, and also with respect to the other unconstrained parameters, namely, $Q_c$ and $F_{\text{till}}$:

*"We performed a sensitivity analysis of the least constrained parameters of our model, i.e., $l_c$, $Q_c$, and $F_{\text{till}}$ (Figure R3). It can be observed that $l_c$ has only a limited effect for hard beds, while it has a more pronounced impact for soft beds. From equation (4), a change in $l_c$ results in a change in the water flux $Q_w$, which will be important if water flow transitions from an efficient to an inefficient flow (or the reverse). However, for hard beds, the entirely efficient or inefficient cases yield similar results (Figure 9b). On the contrary, for soft beds, the difference between the entirely efficient or inefficient cases is more pronounced (Figure 9b), and it follows that there is a stronger dependence with respect to $l_c$. For $Q_c$ and $F_{\text{till}}$, the impact is limited. Finally, it can be noted the spread in the results increase over a time."*

[Figure]

(a) Effect of $l_c$.        (b) Effect of $Q_c$.        (c) Effect of $F_{\text{till}}$.

Figure R3: Sensitivity analysis of the results with respect to the parameters $l_c$, $Q_c$, and $F_{\text{till}}$. The set-up is the same as the one described in the forcing experiments over Thwaites (subsection 4.2, Subglacial hydrology on homogeneous beds), except that different values of these parameters are chosen. The shaded areas correspond to the ranges $l_c \in [5, 15]\,\text{km}$, $Q_c \in [0.5, 1.5]\,\text{m}^3/\text{s}$, and $F_{\text{till}} \in [1, 2]$, and the lines correspond to the nominal values considered in the original experiment.

**References**

Clarke, G. K. C. (1996). Lumped-Element Analysis of Subglacial Hydraulic Circuits. *J. Geophys. Res.*, 101:17547–17599.

Gowan, E. J., Hinck, S., Niu, L., Clason, C., and Lohmann, G. (2023). The impact of spatially varying ice sheet basal conditions on sliding at glacial time scales. *Journal of Glaciology*, 69(276):1056–1070.

Hager, A. O., Hoffman, M. J., Price, S. F., and Schroeder, D. M. (2022). Persistent, extensive channelized drainage modeled beneath Thwaites Glacier, West Antarctica. *Cryosphere*, 16(9):3575–3599.

Hewitt, I. J. (2011). Modelling distributed and channelized subglacial drainage: the spacing of channels. *Journal of Glaciology*, 57(202):302–314.

Hill, T., Flowers, G. E., Hoffman, M. J., Bingham, D., and Werder, M. A. (2023). Improved representation of laminar and turbulent sheet flow in subglacial drainage models. *Journal of Glaciology*, page 1–14.

Hoffman, M. J., Perego, M., Price, S. F., Lipscomb, W. H., Zhang, T., Jacobsen, D., Tezaur, I., Salinger, A. G., Tuminaro, R., and Bertagna, L. (2018). MPAS-Albany land ice (MALI): a variable-resolution ice sheet model for earth system modeling using voronoi grids. *Geosci. Model Dev.*, 11(9):3747–3780.

Kazmierczak, E., Sun, S., Coulon, V., and Pattyn, F. (2022). Subglacial hydrology modulates basal sliding response of the antarctic ice sheet to climate forcing. *The Cryosphere*, 16(10):4537–4552.

Le Brocq, A., Payne, A., Siegert, M., and Alley, R. (2009). A subglacial water-flow model for West Antarctica. *Journal of Glaciology*, 55(193):879–888.

Lu, G. and Kingslake, J. (2023). Coupling between ice flow and subglacial hydrology enhances marine ice-sheet retreat. *EGUsphere [preprint]*.

Pattyn, F. (2010). Antarctic subglacial conditions inferred from a hybrid ice sheet/ice stream model. *Earth and Planetary Science Letters*, 295(3–4):451–461.

Schoof, C. (2010). Ice-sheet acceleration driven by melt supply variability. *Nature*, 468(7325):803–806.

Storrar, R. D., Stokes, C. R., and Evans, D. J. (2014). Morphometry and pattern of a large sample (>20, 000) of Canadian eskers and implications for subglacial drainage beneath ice sheets. *Quaternary Science Reviews*, 105:1–25.

Werder, M. A., Hewitt, I. J., Schoof, C. G., and Flowers, G. E. (2013). Modeling channelized and distributed subglacial drainage in two dimensions. *Journal of Geophysical Research: Earth Surface*, 118(4):2140–2158.

---

## Author Comment (AC3)

**Response to Referee 2 on "A fast and unified subglacial hydrological model applied to Thwaites Glacier, Antarctica" by Kazmierczak, Gregov, Coulon & Pattyn.**

Dear Referee,

We would like to thank you for you detailed review; your numerous constructive comments are much appreciated. We are convinced that your comments helped improving our manuscript significantly. Below, you will find a point-by-point response to your remarks, written in blue. We hope that our responses will be satisfactory.

Best regards,

On behalf of the authors,
Thomas Gregov

**Response to the Referee's comments**

Overview

This manuscript presents a subglacial hydrology model that represents inefficient and efficient subglacial drainage in the context of hard and soft beds, coupled to an ice dynamics model. The model is demonstrated with application to an idealized setting based on the MISMIP experimental setup and to Thwaites Glacier to investigate the influence of subglacial hydrology on its future behavior.

I am glad to see this work being done, combining different pieces of subglacial hydrology modeling in a way that is more practical for large-scale and long-term ice sheet simulations than many previous efforts. While I am enthusiastic about the paper's topic and findings, it will benefit from some revisions to strengthen it before publishing.

Please see the specific comments below. In general, the model description needs additional work for completeness and clarity, as already highlighted by another reviewer's comments. The structure of the paper could also be improved upon for easy navigation, to clearly indicate where results are presented versus experimental descriptions. The lengthy Discussion section would benefit from being broken up into subsections for each theme addressed within it.

These are valuable suggestions and we thank the Referee for them. Following their suggestion, we have added a new subsection to the Model section. It is now structured as follows:

1. Ice-flow model

2. Hydrological model

    2.1. Simplifying assumptions
    2.2. Subglacial water routing
    2.3. Subglacial effective pressure
    2.4. Bed rheology

The additional subsection, 'Simplifying assumptions', should clarify our hypotheses, as also requested by another Referee.

Regarding the sections 3 and 4, we now explicitly mention in the title of each subsection whether we are discussing the experimental set-up or results of simulations.

Finally, we have split the Discussion section into several subsections, as follows:

1. Influence of subglacial conditions

2. Hydrological feedback

3. Model limitations

As a final comment, much of the analysis of Thwaites behavior focuses on location and migration of the grounding line. How would this change by considering a grounding zone rather than a distinct grounding line? Some brief mention or discussion about this would be helpful.

In our model we do not consider sub-shelf melting beyond the grounding line, so partially grounded cells are not affected by sub-shelf melting (see Seroussi and Morlighem (2018) for a more profound discussion on this). Of course, extending sub-shelf melting under grounded ice shelves increases the sensitivity of the model, since you are melting away the grounded ice sheet that is already close to floatation, so the latter is enhanced. In a recent paper, Rignot et al. (2024) make observations of a grounding zone and witness pretty high melting rates in that area, especially when all the heat is used for melting. It leads to values of up to 60 meters per year, which is a lot. Definitely, such high melt rates will increase the sensitivity of grounding-line retreat and it is worth looking into this problem in future research. Furthermore, in a just published paper by Bradley and Hewitt (2024), water intrusion underneath the grounding zone could lead to a further instability. Both mechanisms are not considered in our model, but are now discussed.

I look forward to seeing this work being refined to make an impactful publication. It is an important effort to improve representation of subglacial hydrology in large ice-sheet models, and the work presented here is a great contribution toward this aim.

We would like to thank the Referee for their encouraging comment.

Specific Comments

Lines 40-42: It may be helpful to some readers to give example ranges of the typical temporal and spatial scales discussed here, for both hydrology and ice dynamics.

We have added this information in the revised manuscript – for subglacial hydrology, the spatial and temporal scales can be as small as few meters and a couple of hours, whereas for Antarctic ice-sheet dynamics, areas are hundreds of kilometers wide and the temporal response occurs over spans of several years.

Line 49: Clarify what "various bed types" means.

We have added that this corresponds to the hard/soft distinction.

Figure 1: This figure could be combined with another figure as an inset.

That is a good suggestion. We have included this figure as an inset of the Figure 9(a) of the original manuscript.

Line 89: It would be useful to include a brief description of what you mean by 'efficient' and 'inefficient' drainage before this.

We have added the following here: *"Here we consider that a hydrological system is efficient if it transports large fluxes of water."*.

Line 91: How small is the local scale? Order of sub-meter, meter, tens of meters, hundreds of meters?

We have added the following here: *"observations suggest that channels are meters to at most a few hundreds meters wide, that maximal width being reached close to the grounding line (Drews et al., 2017)"*.

Note that, in our model, this local scale is not explicitly prescribed. Rather, it is the amount of hydrological components (which we refer to as 'conduits' in our manuscript) per grid cell that is prescribed, through the quantity $l_{\mathrm{c}}$.

Line 99: The SHAKTI model also combines inefficient and efficient drainage, with a continuum approach. Sommers, A., Rajaram, H., and Morlighem, M.: SHAKTI: Subglacial Hydrology and Kinetic, Transient Interactions v1.0, Geosci. Model Dev., 11, 2955–2974, `https://doi.org/10.5194/gmd-11-2955-2018`, 2018.

We have added the reference Sommers et al. (2018) here.

Figure 2: I am slightly confused by this figure and the flow shown. A more thorough description of the coupling in the text would probably help.

We have improved the description of the caption of this figure by completing it with additional information. It is now given by the following:

"*Flowchart of the dynamical linkage between the ice sheet and the subglacial hydrology. At each time step, the ice-sheet model provides the basal melt rate $\dot{m}$ and the geometrical potential $\phi_0$. Based on these, the effective pressure is computed in three steps: (i) The global distributed subglacial water flux $\boldsymbol{q}_{\mathrm{w}}$ is computed according to Le Brocq et al. (2009); (ii) a connection between both global and local (conduit) scale is obtained by specifying the distance $l_c$ between the conduits (Gowan et al., 2023), yielding a volumetric water flux $Q_w$ in each conduit; (iii) the effective pressure $N$ is computed for each conduit via a parametrization where $\mathcal{F}(\phi_0/N_\infty) = \mathrm{erf}[(\sqrt{\pi}/2)\phi_0/N_\infty]$ serves as a correction factor for the impact of the grounding line (GL), and where $N_\infty$ is the effective pressure far upstream of the grounding line. This effective pressure is then used by the large-scale ice-sheet model and is the same for all conduits that belong to the same grid cell.*".

Besides, we now clearly mention that the content of the figure is described in further details in the following subsections.

Line 108: How cheap? Give some illustrative value to back up this claim, probably based on domain size, resolution, time step, simulation time, number of processors, wall-clock time.

We have modified this paragraph to the following: "*By contrast, our model is computationally cheap, with the computational time associated with the subglacial hydrology calculation representing only a small fraction of the computational time associated with the ice-sheet model. This allows us to study the impact of subglacial hydrology on ice dynamics on a large scale and at a limited computational cost, while at the same time keeping the essential features of complex subglacial hydrology models*".

Overall, the major computational cost comes from the distributed water flux $\boldsymbol{q}_{\mathrm{w}}$ calculation, which is done efficiently using the method from Le Brocq et al. (2009), combined to a parametrization (i.e., an explicit formula) for the effective pressure – equation (7).

To give an order of magnitude, the non-forced Thwaites experiments on a homogeneous bed with a hard bed (HARD) take $\sim 15\%$ more computing time compared to the the no-hydrology case (NON).

Line 112: Depth-integrated subglacial water flux?

We are not convinced that 'depth-integrated subglacial water flux' is the right name for $\boldsymbol{q}_{\mathrm{w}}$. There are several ways to relate this flux to well-known quantities. One of these is to consider that this water flux is evenly distributed over the whole grid cell, giving rise to a water film of depth $d_{\mathrm{w}}$ (Le Brocq et al., 2009). In that case,

$$\boldsymbol{q}_{\mathrm{w}} = \overline{\boldsymbol{u}}_{\mathrm{w}} \, d_{\mathrm{w}}, \tag{R1}$$

where $\overline{\boldsymbol{u}}_{\mathrm{w}}$ is the depth-averaged horizontal velocity. Hence $\boldsymbol{q}_{\mathrm{w}}$ has units $\mathrm{m}^2/\mathrm{s}$.

Another way is to consider the water flux. Usually, it is defined as the volume of water that crosses a surface per unit of time. The relation between $\boldsymbol{q}_{\mathrm{w}}$ and the total water flux in each cell

$Q_\text{total}$ follows the same reasoning to convert $\boldsymbol{q}_\text{w}$ and $Q_\text{w}$: we have

$$Q_\text{total} = \Delta x \, \|\boldsymbol{q}_\text{w}\|, \tag{R2}$$

where $\Delta x$ is the width of the square grid cell. Hence, $\boldsymbol{q}_\text{w}$ can be interpreted as a water flux per unit length (hence our name 'distributed subglacial water flux'), but not as a depth-integrated water flux.

Lines 119-120: For completeness, describe how the melt rate due to dissipation ($\dot{m}_\text{w}$) is calculated. Do you include this dissipation term everywhere? This is worth clarifying because of the legacy of models that only include it for channel components.

Apologies, the expression for $\dot{m}_\text{w}$ was indeed missing from the manuscript. We have modified this paragraph as follows: "(...), i.e.,

$$\dot{m} = \frac{G + \boldsymbol{\tau}_\text{b} \cdot \boldsymbol{v}_\text{b} - q_T}{L_\text{w}} + \dot{m}_\text{w}\,, \tag{R3}$$

where $G$ is the geothermal heat flux, $q_T$ is the thermal conduction flux, $L_\text{w}$ is the latent heat for ice, and $\dot{m}_\text{w} = |\boldsymbol{q}_\text{w} \cdot \nabla\phi|/L_\text{w}$ is the water melt rate due to the dissipated energy from the subglacial water conduits. However, we do not include this last term in our simulations as it was found to be negligible compared to the other terms."

Lines 126-129: Intriguing to use the simple routing scheme – I'm interested to see the results that support the claim that $\phi_0$ is approximately equal to $\phi$ over most of the domain. Perhaps pointing to a figure would be good, rather than simply saying "in anticipation of what follows". It seems like a strange thing to want to decouple the water routing from effective pressure when you are interested in modeling subglacial hydrology, given that water flow is driven by gradients in potential, which obviously changes depending on effective pressure.

In the new structure of the description of the model, we tried to improve the description of the hydrological model by stating the essential assumptions prior to the derivation of the model. This allowed us to remove this 'in anticipation of what follows'.

Specifically, the following discussion has been added to the new 'Simplifying assumptions' subsection:

"The key simplifying assumptions are given by the following:

1. There is limited temporal melt variability so that the hydrological system is in a quasi-static equilibrium with respect to the ice-sheet geometry. Therefore, changes in ice geometry will be the main driver for changes in subglacial water variability (both spatial and temporal).

2. A few kilometers upstream of the grounding line, the hydraulic gradient is approximated by the geometric gradient.

3. The drainage density is uniform and the effective pressure is not calculated at a sub-grid level.

The first assumption is based on several studies of subglacial hydrology in Antarctica (Le Brocq et al., 2009; Pattyn, 2010; Kazmierczak et al., 2022), among others, that demonstrate that — contrary to the Greenland ice sheet— there is limited surface meltwater infiltration. Hence, changes in hydrology are primarily due to changes in ice geometry. Since the time scales associated with water flow are much smaller than those associated with ice flow, subglacial hydrology automatically adapts to any change in ice geometry and reaches the associated equilibrium. The second assumption is motivated by a scaling analysis through an estimation of the dimensionless ratio $\eta := [\nabla N]/[\nabla\phi_0]$, where $[\nabla N]$ is the scale of the spatial gradients for the effective pressure and $[\nabla\phi_0]$ is the characteristic scale for the geometric potential gradient. For the former we take $[\nabla N] = [N]/[x]$, with $[N] = 1\,MPa$ and $[x] = 10^3\,km$. For the latter we take

$[\nabla\phi_0] = 5 \times 10^{-2}\,MPa\,km^{-1}$, *which is a plausible value for ice sheets (Hewitt, 2011). This results in $\eta = 2 \times 10^{-2} \ll 1$, suggesting that $\|\nabla N\| \ll \|\nabla\phi_0\|$ and $\nabla\phi \approx \nabla\phi_0$. We further note that profiles obtained with a high-resolution subglacial hydrology model suggest that $\nabla\phi$ and $\nabla\phi_0$ have a correlation of at least $\sim 80\%$ for a region that is several kilometers upstream of the grounding line (see Supplementary Material S1). Finally, the third assumption follows from our modeling approach, where we do not describe the effective pressure at the sub-grid scale and where we assume the same number of conduits in each grid cell, similar to Gowan et al. (2023)."*

Note that saying that $\|\nabla N\| \ll \|\nabla\phi_0\|$ is not the same as saying that $N = 0$; rather, we are saying that $N$ varies much more slowly in space compared to $\phi_0$.

The Supplementary Material S1 refers to the assessment of the assumption that $\nabla\phi \approx \nabla\phi_0$ outside the vicinity of the grounding line based on data. Here is the content of this addition to the supplementary materials:

*"Here, we provide additional data to underpin the validity of the assumption that $\nabla\phi \approx \nabla\phi_0$ outside the range of influence of the grounding line, which is a few kilometers from it. Since there are no direct observations of the effective-pressure field in Antarctica, we have to rely on high-resolution models. A first test case comes from Lu and Kingslake (2023) who uses a high-resolution model that couples ice-sheet dynamics and subglacial hydrology for hard beds. Potential limitations of that study is that it considers a flow line and a smooth bedrock. The assumption that $\nabla\phi \approx \nabla\phi_0$ a few kilometers upstream of the grounding line is confirmed numerically (Figure R1).*

[Figure]

(a) Numerical results.  (b) Computed gradients.

Figure R1: Data derived from Figure 4 of Lu and Kingslake (2023).

*A second test case comes from Hager et al. (2022) who applied the high-resolution model MALI (Hoffman et al., 2018) to Thwaites Glacier. They also consider a hard-bed hydrology. The computed effective pressures along a center-line transect are shown in Figure R2. Note that the signals are much more noisier compared to the first test case. This noise can be attributed to the model resolution, but also to the presence of localized hydrological features that cross the center-line transect at which the effective pressures are evaluated, therefore resulting in very localized variations. However, we observe a good correlation between $\partial_s\phi$ and $\partial_s\phi_0$ out of the vicinity of the grounding line (Figure R2): $\sim 80\%$ over the range $[10, 400]\,km$, suggesting that the assumption that $\nabla\phi \approx \nabla\phi_0$ is valid in this region."*

Line 132: How is $l_c$ chosen? How sensitive are results to this value?

That is a good question and apologies that we did not specify this more clearly in the manuscript.

[Figure]

(a) Numerical results.

(b) Computed gradients.

Figure R2: Data derived from Figure 8 of Hager et al. (2022).

We have added the following after its introduction: "*We take $l_c = 10\,km$, which is similar to the value considered in Gowan et al. (2023) based on observations of distances between eskers formed under the Laurentide Ice Sheet (Storrar et al., 2014).*"

Furthermore, we have added an appendix with a simple sensitivity analysis with respect to $l_c$, $Q_c$, and $F_{\text{till}}$ (Figure R3).

[Figure]

(a) Effect of $l_c$.

(b) Effect of $Q_c$.

(c) Effect of $F_{\text{till}}$.

Figure R3: Sensitivity analysis of the results with respect to the parameters $l_c$, $Q_c$, and $F_{\text{till}}$. The set-up is the same as the one described in the forcing experiments over Thwaites (subsection 4.2, Subglacial hydrology on homogeneous beds), except that different values of these parameters are chosen. The shaded areas correspond to the ranges $l_c \in [5, 15]\,\text{km}$, $Q_c \in [0.5, 1.5]\,\text{m}^3/\text{s}$, and $F_{\text{till}} \in [1, 2]$, and the lines correspond to the nominal values considered in the original experiment.

It can be observed that $l_c$ has only a limited effect for hard beds, while it has a more pronounced impact for soft beds. From equation (4), a change in $l_c$ results in a change in the water flux $Q_w$, which will be important if water flow transitions from an efficient to an inefficient flow (or the reverse). However, for hard beds, the entirely efficient or inefficient cases yield similar results (Figure 9b). On the contrary, for soft beds, the difference between the entirely efficient

or inefficient cases is more pronounced (Figure 9b), and it follows that there is a stronger dependence with respect to $l_\mathrm{c}$. For $Q_\mathrm{c}$ and $F_\mathrm{till}$, the impact is limited. Finally, it can be noted the spread in the results increase over a time. This figure and discussion have been added to the additional appendix.

Line 143: Do you always assume turbulent flow in the model?

Yes, although one could argue that in practice the flow could also be laminar. In our manuscript, we have chosen a turbulent parametrization for the Darcy's flow equation; this fixes both the exponents $\alpha$ and $\beta$, as well as the value of the conductivity parameter $K$. In this way, we follow the analysis of Schoof (2010) and Gowan et al. (2023).

However, we now discuss the possibility of having a laminar flow of water in the revised version of our manuscript, together with relevant references from the hydrological literature, in particular Hill et al. (2023).

Lines 149-150: Is the opening by sliding over obstacles the same for hard and soft beds? It isn't clear from this sentence whether the model treats these the same or differently, or if this means that the physical interpretation is simply different.

We treat them similarly; this is now clarified in the revised manuscript, where we have written the following : "*The bed obstacles correspond to bed protrusions if the bed is hard, and to clasts if the bed is soft, and our model treats these cases the same.*".

Lines 150-151: Why isn't melt opening associated with both inefficient and efficient drainage systems? Similarly to the previous comment, is this sentence purely commentary on physical interpretation, or describing a coded switch in melt equations applied to different parts of the model domain?

Physically, we associate melt opening with an efficient drainage system, as this is the system in which it will be the dominant term (as it is proportional to the water flux). However, in general, opening can occur by different mechanisms; in our model we consider both opening by melting and opening by sliding over bed protrusions. By default, both mechanisms (efficient and inefficient) operate in our model, as both terms are included. In particular, there is no switch between the two in the code.

However, for some simulations in the paper we considered either one of them to test the sensitivity.

Lines 175-176: It would be helpful to justify the assumption that effective pressure is "fairly constant" far from the grounding line, perhaps with a plot either in the main text or in a supplement or appendix. How far from the grounding line?

For the "fairly constant" assumption, we refer to our response to the comment of Lines 126-129.

Far from the grounding line, $N \approx N_\infty$, while close to it, $N \approx \phi_0$. As a first approximation, the switch between the two regimes therefore appears when $\phi_0 \approx N_\infty$, which is consistent with our equation (7). Numerically, for the simulations over Thwaites, this corresponds to a few kilometers from the grounding line.

Equation 6: Are $N_\infty$ and $S_\infty$ the effective pressure and conduit cross-sectional area far from the grounding line? That's what I infer, but they should be explicitly defined.

Yes, that is the case. This is now clarified in the revised manuscript: we introduce equation (6) as follows: "*In that case, we obtain algebraic equations for the effective pressure and the cross-sectional area far from the grounding line, $N_\infty$ and $S_\infty$:*".

Line 181: How close to the grounding line?

This corresponds to the region where $\phi_0 \approx N_\infty$, which is typically a few kilometers from the grounding line.

Section 2.2.3: How sensitive are results to these geometric assumptions (the relationships between $L$, $H$, and $S$, also the value of $F_{\text{till}}$)? These are nicely explained here, but are still mostly unconstrained by observations and are somewhat arbitrary, so it would be more thorough to consider their influence on model results.

This is an important point. It is evident from our analysis that the results are very sensitive to these geometric relations (as there is a large variability in the results when comparing hard and soft beds; see e.g. Figures 5 and 9). Overall, this highlights the necessity for more data on the bed rheology of the Antarctic ice sheet, as well as additional studies to compare this data with the results of numerical models.

For the value of $F_{\text{till}}$, we have taken $F_{\text{till}} > 1$ because we physically expect a lower effective pressure for soft beds, compared to hard beds. We have added in the Appendix a sensitivity on the choice of $F_{\text{till}}$ (Figure R3), and results suggest a relatively limited effect.

Line 204: missing a period.

Corrected.

Line 207: How is the critical water flux value $Q_c$ selected?

We added the following sentence: "*In our simulations, we took $Q_c = 1\,m^3\,s^{-1}$ which corresponds to the scale of the water flux considered in Walder and Fowler (1994).*". This value is also of the same order of magnitude as the value of the flux for which the regime transitions from an efficient regime (in which the sliding-over-protrusions opening term dominates) to an efficient regime (in which the melt-opening term dominates) for a hard bed (see Figure 5).

Its sensitivity is further gauged in the Appendix (Figure R3).

Line 209: I am curious as to how confident we can be in prescribing which regions are hard bedded and which soft bedded, particularly as these can be highly heterogeneous spatially. Maybe this is coming later in the application to Thwaites.

That is a good point and, actually, a motivation for our work. If our results were similar over a hard, a soft, and a mixed bed, then the question of the bed type would not be particularly important for ice-sheet simulations. But our results (which indeed are in the Thwaites section) precisely show the opposite: the type of bed is a key parameter that strongly affects ice flow, together with the flow efficiency. Nonetheless, the vast majority of hydrological models consider water flow to take place over beds that are similar to what we describe as 'hard beds', with a mix of cavities and channels (i.e., no canals). This suggests that: (i) ice-sheet models should be able to incorporate water flow over both hard and soft beds and (ii) additional research efforts should address the characterization of the bed, specifically for basins that are susceptible to be retreating in the next centuries. Since the spatial distribution of hard and soft bed is poorly known, our study advocates for the importance to improve observational constraints.

Line 226: Do you mean "entirely efficient" (instead of "entirely effective")?

Yes – this is now corrected.

Line 229: Similarly, should this be "entirely inefficient"?

Yes – this is now corrected too.

Line 229: It is not clearly justified why the dissipation term should be removed in the inefficient system. Is this based on similar earlier models that needed this for numerical stability? Is this

necessary in your model formulation? I'm not convinced that it makes sense physically to ignore the dissipative contribution to melt if you can help it.

We do not remove the dissipation term for a physical or numerical reason, but for a testing purpose, to assess the sensitivity of the system with respect to its different opening mechanisms.

Note that by default, we consider both opening terms: the 'inefficient' one, associated with sliding over bed protrusions/clasts, and the 'efficient' one, associated with melting. It is only in our additional tests that we consider entirely efficient and entirely inefficient cases in which we artificially remove one of the opening terms. We have slightly modified this paragraph so that it is more clear (see response to your next comment).

Lines 225-230: This section about switches imposed in the model needs to be clarified. It's great to represent inefficient and efficient systems and systems that don't fall cleanly into either category. But it is not entirely clear from reading what the thresholds are for selecting different forms of the equations. Are these manually set based on preference of the modeler and the problem of interest? Or are there criteria that automatically trigger these switches?

We have modified the beginning of this paragraph as follows: *"Besides soft, hard, and mixed beds, we also consider entirely efficient and inefficient drainage systems to gauge the sensitivity of both separately, independent of the subglacial water flux. By default, our model is such that the subglacial system naturally transitions from one to another depending on the subglacial water flux. This transition happens because the melting term, which is proportional to the water flux, becomes dominant over the sliding term in the left-hand side of (5a) as the water flux increases. To obtain an entirely efficient system, the opening term associated with the sliding over obstacles, $\|\boldsymbol{v}_\mathrm{b}\| h_\mathrm{b}$, is removed from equation (5a), as well as from the parametrization (6a). We also set $Q_\mathrm{c} = \infty$, which guarantees that the conduit geometry is the one of an inefficient system for soft beds. To obtain an entirely inefficient system, we remove the efficient component, $Q_\mathrm{w} \|\nabla \phi\| / \rho_\mathrm{i} L_\mathrm{w}$, from (5b), together with the condition that $Q_\mathrm{c} = 0$."*

This should make it more clear that in the model, the 'switch' between the efficient and inefficient regimes is naturally included in the equations as a function of subglacial water flux.

Lines 237-238: It would be helpful to comment on why Weertman was selected as the sliding law, and why a uniform value for the friction coefficient, and why that value. (You have to make some choices, just curious about the rationale behind these selections).

We have revised the section on model initialization, as we felt it was not clear enough. In particular, it seemed to us that the initialization procedure would be easier to understand if it were described in words rather than equations. Here is the new version:

*"On this bed topography a marine ice sheet is developed with a spatial resolution of $500\,m$, following the set-up described in the EXP1 of the MISMIP experiments (Pattyn et al., 2012, see Figure 6a). The steady state obtained with these conditions is considered to be the 'reference state'.*

*In our experiments, we use a regularized Coulomb friction law combined with hydrological models, while the reference state has been obtained with a Weertman friction law. To guarantee that the thickness and velocity fields obtained in the reference state are still compatible with a steady state, we modify the friction coefficient at each position, following the method of Brondex et al. (2017, 2019). In practice, an iterative nudging method is used so that the basal friction matches the basal friction obtained in the reference state. Here, the subglacial hydrologies are generated with a uniform basal melt rate underneath the grounded ice sheet of $\dot{m}/\rho_\mathrm{w} = 5 \times 10^{-3}\,m\,a^{-1}$, which corresponds to the mean basal melt rate of the Antarctic ice sheet (Pattyn, 2010). By construction, this method yields initial states that are steady states and in which both the geometry and the velocity field are identical for each type of hydrology, allowing a direct comparison between them. The initial ice-sheet effective-pressure profiles are shown in Figure 6b."*

In this new version, it should be more clear that the choice of a Weertman friction law to obtain the reference state stems from the use of the standard MISMIP set-up (Pattyn et al., 2012).

Line 238: should be "upper boundary condition" (singular, not plural for grammatical agreement).

Thanks for spotting this mistake. As we have changed the presentation of the initialization method, it no longer appears.

Line 241: This is confusing about the spatially variable friction coefficient used here, when it was just stated in the previous paragraph that a uniform friction coefficient was used.

We have revised the paragraph on the initialization method (see response to comment on Lines 237-238). We hope this information is now more clear.

Line 246: So did $N$ change throughout the inversion here? A brief description of that would help clarify this, i.e. what initial distribution of $N$ was assumed, and how was it altered through the iterative nudging process?

Yes, $N$ is allowed to change during the inversion: it is updated at each iteration of the nudging procedure. The initial effective-pressure field is the one that is obtained from the model with the initial guess for the sliding coefficient. Note that in the revised version of this paragraph, we have chosen not to include too many technical details about this procedure (i.e., not too many equations), as it seemed to us that these might be rather confusing to the reader.

Line 257: Some observations have suggested low effective pressure in the interior. Can you comment on this here or elsewhere?

Indeed, and this is why we think (and demonstrate) that the HAB parameterization that is widely used in ice sheet models, fails. In the discussion section we mention that our results are qualitatively similar to those of Hager et al. (2022) for hard beds.

Line 263: This statement about the default switch between efficient and inefficient drainage equations would be helpful to include above (see comment about Lines 225-230).

We have included this statement in the paragraph of Lines 225-230.

Line 279: With what size time step is the ice sheet model run for 20,000 years? Is the hydrology model also run for 20,000 years?

The time step for the idealized simulations is 5 years, while for the Thwaites experiments it is 0.05 years. These time steps are now explicitly mentioned in the text.

The question of the time step of the hydrological model is an important one. This is explained in more details in the Appendix C. The key result is that the hydrological model should be updated at a frequency that is at least of the same order as the one used for the ice-sheet model (i.e., with a time step that it smaller to or similar to the one used by the ice-sheet model).

Line 280: It would be helpful to include a brief reminder of how the hydrology and ice sheet models are coupled here.

We have added the following sentence here: *"The hydrological model is updated at each time step (see also Appendix C)."*.

Figure 7: Why is the sliding velocity for entirely inefficient not included in panel c? It would also be useful to plot the effective pressure response for entirely efficient and entirely inefficient in panel b to help strengthen your points about the importance of the switching behavior.

Following your suggestion, we have added the sliding velocity for the entirely-inefficient case in

panel (c), as well as the effective pressures corresponding to the entirely-efficient and entirely-inefficient case in panel (b).

Lines 341-342: What is the criterion to be considered a collapse?

We have precised what we meant by 'collapse' by modifying the Line 336, which is the first time this word appears, to the following: "*This is in line with large-scale model experiments (Coulon et al., 2023) showing that Thwaites Glacier may collapse, i.e., that it will continue to retreat at an accelerated rate even if the forcing is completely stopped, under present-day climatic conditions on time scales of several centuries*".

Sections 4.1-4.3: The structure of this section needs some improvement. The titles of 4.2 and 4.3 may be renamed to make clear that the first section is the experimental description for Thwaites, and the second and third sections are presenting results. I was a bit confused by this structure while reading. The information about the threshold for hard-to-soft transition in line 328 seems to be repeated in line 352. We also seem to be missing information in the experimental setup on Thwaites about model resolution(s) and time step size(s), which would be interesting to know.

We now make it clear that subsections 4.2 and 4.3 correspond to results – we have renamed these "*Results: subglacial hydrology on homogeneous beds*" and "*Results: subglacial hydrology on heterogeneous beds*", respectively. We also clarified that the limit in line 352 is a repetition of the one in line 328, so that it is clear to the reader that it is not an oversight and that we are simply repeating this information for clarity. We now also mention the resolution (2 km) and the time step (0.05 years).

Discussion: I recommend separating this into some subsections with corresponding headings.

Thanks for the suggestion. We have followed it; it also seemed to us that the Discussion section was rather long and could benefit from further structuring. It is now divided in three subsections:

1. Influence of subglacial conditions
2. Hydrological feedback
3. Model limitations

Tables A1 and A2 could be combined into a single table.

Agreed. However, we have chosen to use two different tables: by separating the Greek and Latin alphabets, we are able to obtain a table that is no longer than one page. If we put them together, we would get a table that would extend over two pages.

**References**

[revised manuscript text omitted]

---

## Author Comment (AC4)

**Response to Referee 3 on "A fast and unified subglacial hydrological model applied to Thwaites Glacier, Antarctica" by Kazmierczak, Gregov, Coulon & Pattyn.**

Dear Referee,

We would like to thank you for the time you have spent reading our paper, and for your comments on it. Your questions should help us to clarify certain points and improve our paper.

You will find below, in blue, our responses to your comments.

Best regards,

On behalf of the authors,
Thomas Gregov

**Response to the Referee's comments**

In this paper, the authors present a novel subglacial hydrology model that includes efficient and inefficient drainage systems on bed rheologies that range continuously from soft to hard. The model can be run on the same spatial scales as ice flow, which makes the model computationally efficient. The model is applied to Thwaites Glacier in Antarctica and the response to present day climate forcings is analyzed for hard, soft, and mixed beds for efficient and inefficient drainage systems.

I think the development of a model with these capabilities is an exciting and important step in modeling coupled ice flow and subglacial hydrology on the ice sheet scale. There are a few things that I believe should be clarified and adapted in the writing before publication.

We would like to thank you for your encouraging comment. You will find the responses to your remarks hereafter.

General comments:

Although assuming hydrologic equilibrium are most likely appropriate for Antarctica, it should be made clear that this model would not be appropriate for modeling places with more highly variable water fluxes such as mountain glaciers or the Greenland ice sheet (or at least the margins) where the subglacial hydrological system is often not in equilibrium due to external meltwater input which varies on much shorter timescales than changes in ice geometry.

Indeed, our model is clearly primarily meant for Antarctica. To clarify this, we have made the following changes to the manuscript:

- The title has been changed to "*A fast, simplified, and unified subglacial hydrological model for the Antarctic ice sheet and outlet glaciers*".

- We have modified the Model section in order to highlight the key assumptions of our model. It is now structured as follows:

   1. Ice-flow model
   2. Hydrological model
      2.1. Simplifying assumptions
      2.2. Subglacial water routing
      2.3. Subglacial effective pressure
      2.4. Bed rheology

   The additional subsection, 'Simplifying assumptions', should clarify our hypotheses. In it, we specify that the equilibrium assumption is based on the study of subglacial hydrology in

Antarctica (Le Brocq et al., 2009; Pattyn, 2010; Kazmierczak et al., 2022), where there is limited surface water infiltration. Thus, changes in hydrology are primarily due to changes in ice geometry. Since the time scales associated with water flow are much smaller than those associated with ice flow, subglacial hydrology automatically adapts to any change in ice geometry and reaches the associated equilibrium.

In this additional subsection, we also discuss the simplification of the hydraulic gradient (the assumption that $\|\nabla N\| \ll \|\nabla \phi_0\|$ far from the grounding line, so that $\nabla \phi \approx \nabla \phi_0$ in that region), and the conduits distribution (uniform drainage density with no sub-grid resolution).

There could be an argument that this model could be applied to the Greenland Ice Sheet on long timescales (that neglect short term changes resulting from subglacial hydrology), but this would need to be thoroughly addressed and justified. Add to the abstract and conclusion that this model is appropriate for modeling subglacial hydrology where changes in water flux are on the same timescales as changes in ice geometry. Does the thermomechanical ice flow model determine where there is basal melting? If so, does this evolve in time? How is water routed through areas with a frozen bed? Is there refreezing? Elaborate more on this, perhaps in section 2.1.

We agree with the referee that the current model cannot be applied to the Greenland ice sheet, since short-time variations in meltwater supply and subglacial water pressure are important and neglected in the current description. As mentioned in the previous response, we have modified the title and added a new subsection to highlight this assumption and its range of validity.

The thermomechanical ice flow determines where basal melting occurs and these regions may evolve over time (Pattyn, 2017). In our idealized simulations, the bed is entirely temperate, and for our simulations over Thwaites, the vast majority of the domain is also temperate, since the geothermal heat flow underneath the West Antarctic ice sheet is rather high and most of the bed remains temperate.

There are a few places in the manuscript where references are made to literature with different timescales and rates of change being discussed than in this paper. In this model, the assumptions around hydrologic equilibrium (melting balances closure in Eq. 5b) imply that the changes in time in the subglacial hydrologic system are small enough to neglect $(\mathrm{d}S/\mathrm{d}t = 0)$. However, the manuscript contains references to Schoof (2010) and Iken (1981) which are specifically referring to when $\mathrm{d}S/\mathrm{d}t$ is not zero. In line 267, there is a reference to Schoof (2010) and the importance of the meltwater variability rather than meltwater input, but these processes are time-dependent changes that occur on timescales of hours to days, at most months. From Schoof (2010), "Further acceleration must then be driven instead by **short-term temporal variability** in water supply." In this model, it is assumed to be in steady state which means that these short term increases in water pressure due to varying water flux on short timescales that result in conduit growth are inherently neglected. The same is true for the reference in line 292 in the manuscript, which I believe is referring to the following statement in Iken (1981). "It has been seen that the effect of a water pressure $p_\mathrm{w}$ on the sliding velocity is largest at the instant of separation and then gradually decreases **until steady cavities have formed**." This is again talking about when the system is not in steady state, unlike the model presented here. To be clear, I think the assumption of steady-state is reasonable given the context of Antarctica, where changes in water flux occur on long-timescales, but these references and associated statements (specifically noted in the line-by-line comments) are not applicable in this context.

It is true that the assumptions in Schoof (2010) apply to the Greenland ice sheet, with, in particular, a strong melt supply variability. Here our model fails to cope with such changes. However, an important aspect of that paper is the ability of water channelization to slow down the ice. Fundamentally, this channelization mechanism is linked to the form of the $Q - N$ curve, and is more specifically due to the fact that $\partial N / \partial Q > 0$ for channels. While the hydrological

system in our manuscript is considered at equilibrium, it does evolve *with respect to the ice-sheet geometry*. Nevertheless, we have removed the reference to Schoof (2010) on Line 267; we agree that it was misleading as it was associated with shorter time scales.

Moreover, the comparison with Iken (1981) remains in our opinion legitimate: while the hydrological system is assumed to be at equilibrium at each time, this does not exclude its evolution over time. In particular, cavities are allowed to grow over time, provided the hydrological system stays in quasi-static equilibrium with respect to the ice-sheet geometry.

Line-by-line comments:

Line 5: This is worded strangely. Perhaps, "We find that accounting for subglacial hydrology in the sliding law accelerates the grounding line retreat of Thwaites Glacier under present-day climatic conditions." or something similar.

Thanks for your suggestion, which has been corrected accordingly.

Line 13: 'behavior' since using American spelling elsewhere.

Corrected.

Fig 1. Figure could be made part of Fig. 9 or moved to before Fig. 9, but I don't feel strongly about this.

Following your suggestion, we have included this figure as an inset of Figure 9.

Line 71: introduce variable before introducing product of variables "... $N$ is the effective pressure, $C$ is a friction coefficient limiting the shear stress to a maximum plastic value $CN$, ..."

Corrected.

Line 90: Replace "that is much smaller than the global one" with " that is on the order of meters" or what the scale actually is.

We have modified this sentence to the following: "*We define two spatial scales: a global scale, which is the same as the one used for the ice-sheet model and that is typically of the order of kilometers, and a local scale, associated with a water conduit, and that is much smaller than the global one; observations suggest that channels are meters to at most a few hundreds meters wide, that maximal width being reached close to the grounding line (Drews et al., 2017).*".

Line 93: The term 'conduit' may be confusing, as it is often synonymous with channels and efficient drainage in the subglacial hydrology literature. Although, I don't have a better suggestion...

We are not entirely happy with this word either, but we could not find a better alternative. However, we now emphasize that it is not synonymous with channels. Specifically, we have added at Line 96 the following sentence: "*In particular, we do not use 'conduit' as a synonym for 'channel', as a conduit can correspond to other types of hydrological elements.*".

Lines 105-106: Bueler and van Pelt (2015) ran the model efficiently on the Greenland ice sheet, but these statements are generally true.

It is true that the implementation described in Bueler and van Pelt (2015) is computationally efficient, as it has been optimized for large-scale simulations. However, it represents a significant computational cost when the subglacial hydrology is coupled to the ice flow, since the equations described in Bueler and van Pelt (2015) take the form of coupled PDEs.

On the contrary, our model is computationally cheap, as the hydrological model boils down to one simple algebraic expression (Equation 7) that does not require the solution to a differential equation.

Therefore, we have added a sentence at the end of the paragraph: "*Finally, due to their high spatial and temporal resolution they are often computationally demanding. The latter may limit their application to drainage basins or single glaciers on time scales of a few years. However, our model is computationally cheap, with the computational time associated with the subglacial hydrology calculation representing only a small fraction of the computational time associated with the ice-sheet model. This allows us to study the impact of subglacial hydrology on ice dynamics on a large scale and at a limited computational cost, while at the same time keeping the essential features of complex subglacial hydrology models.*".

Line 111: Elaborate on what these conditions are. "This domain evolves over time according to internal and external conditions."

We have modified this sentence to the following: "*This domain evolves over time according to both internal conditions (e.g., changes in ice velocity) and external conditions (e.g., changes in sub-shelf melt).*".

Eq 3: $L_\mathrm{w}$ is confusing with $L$ being the length of conduit. Use cursive L or something else to denote latent heat.

We changed $L_\mathrm{w}$ to $\mathcal{L}_\mathrm{w}$ to avoid any confusion.

Line 120-123: This assumption is only reasonable in Antarctica and locations where changes in water flux are only through changes in ice geometry which happen on long time scales. Make this more clear in the text.

We have modified the title of the paper to "*A fast, simplified, and unified subglacial hydrological model for the Antarctic ice sheet and outlet glaciers*". Furthermore, the hydrological model description (subsection 2.2) now starts with a discussion of the simplifying assumptions, in particular this steady-state hypothesis.

Line 128: Delete "anticipating what follows".

The assumption that $\nabla\phi \approx \nabla\phi_0$ far from the grounding line is now explicitly described in the additional subsection 'Simplifying assumption' of the Model section. Hence, we have removed this statement.

Fig 4: This is how the channel might look in theory. In the caption of Fig 4, it should also be stated that this is what you are trying to capture and not the actual geometry you have described in your model (which is square channel?).

Indeed, the conduits shown in Figure 4 are schematics. We do not make a distinction between a square and circular channel. This is why we write '$L \sim H$', in the scaling of Table 1: the width of the channel is similar to its height, but there could be a $\mathcal{O}(1)$ factor here. In other words, what matters is the functional relation, i.e., that $L$ increases linearly with $H$. In a second step, we prescribe the relation between $L$ and $H$ through $H = \sqrt{S}$, which is the exact solution for a square channel, but approximates a circular channel as well, given all other uncertainties in our approach.

Line 220-224: Elaborate on how this relates to theory and observations. Are your model results what we expect for these cases?

This role of this paragraph is to provide an intuitive explanation behind the divergence between the hard-bed and soft-bed trends for efficient systems as shown in Figure 5. It can indeed be observed on this figure that for large water flux values, the effective pressure is an increasing function of the water flux for hard beds (i.e., for channels), while it is a decreasing function of the water flux for soft beds (i.e., for canals). To provide this explanation, we solely rely on the theory, i.e., on the equation (5b).

We agree with the Referee that a comparison with observations would be most welcome. However, such observations are still fairly limited for Antarctica. Nevertheless, there exists interesting studies in the recent literature that aim at validating hydrological models against observational data, e.g., Hager et al. (2022), which we refer to in our Discussion section. Overall, and as mentioned in the conclusion of our paper, it is clear that further research should address this comparison between numerical results and observations.

Fig 5:

Wouldn't we expect much higher effective pressures for that high of water flux in an efficient system? Or is this because it is the average of effective pressure over a larger scale than the channel? This could be made more clear either in the text or in the figure caption. On first glance at the figure, I find this result surprising.

We agree with the Referee that this result might be surprising. However, two reasons may eventually explain this behavior:

- The plot has been made based on standard values for $A$, $\|\boldsymbol{v}_\mathrm{b}\|$, and $\|\nabla\phi_0\|$, whose values are mentioned in the caption. However, the effective pressure depends quite strongly on those variables (specifically, it depends quite significantly on $\|\nabla\phi_0\|$), so modifying these values can lead to both larger and lower values, respectively, in terms of the effective pressure $N_\infty$ far from the grounding line.

- For large flux values, $N_\infty$ is an increasing function of the flux, but the growth is rather slow: we have $N_\infty \propto Q_\mathrm{w}^{1-1/\alpha}$. For $\alpha = 5/4$, this yields $N_\infty \propto Q_\mathrm{w}^{1/5}$.

On a related note, do you ever observe this high of water fluxes in channels far from the grounding line in the model? How exactly is the water fluxes on the local scale in the efficient system related to the effective pressure which is an average over a larger area presumably?

We do not observe water fluxes in our results that are that important, especially far from the grounding line. The maximum water fluxes that we observe are of the order of a few tens of cubic meters per second, and these are obtained close to the grounding line (see Figure S1(d) in the supplementary materials). However, one should realize that our subglacial water routing is of course an approximation of the real subglacial system and we may not necessarily capture all the details of that system.

Again, the plot shown in Figure 5 is obtained for specific values of several parameters, so the effective pressure value could deviate from it, especially if the gradient in the geometric potential, $\|\nabla\phi_0\|$, varies spatially.

In our model, we do not explicitly compute a spatial average of the effective pressure within each grid cell. Once the volumetric water flux $Q_\mathrm{w}$ has been obtained, the value for the effective pressure is computed, using equation (7). This value for the effective pressure is the one that is sent back to the ice-sheet code. The fact that we do not compute explicitly the spatial distribution of the effective pressure within in each grid cell is an inherent limitation of our model. There is no simple way to avoid it, as we explicitly do not want to resolve the dynamics at a sub-grid scale.

To make this limitation more clear, it is now explicitly mentioned in the additional 'Simplifying assumption' subsection which is at the beginning of the Model section. This limitation is also briefly mentioned in the Discussion section (*"Our subglacial hydrology models do not include variations of effective pressure below the resolution of the ice-sheet discretization. This is a clear limitation, as we have shown that the spatial variability plays an important role in the numerical experiments."*). Nonetheless, we have improved the structure of the Discussion section, which is now divided in three subsections:

1. Influence of subglacial conditions
2. Hydrological feedback

3. Model limitations
This should help to highlight the limitations of the model.

On what scale are these calculated?

The effective pressure is computed using equation (7), and the value obtained is sent back to the ice-sheet code. The Thwaites simulations are done with a $2\,\mathrm{km}$ uniform resolution.

Line 226-229: You mean efficient/inefficient, not effective/ineffective. The system still transports water, so it isn't ineffective. It just doesn't transport it efficiently.

Corrected.

A discussion about how effective pressure behavior differs near the grounding line and far from the grounding line would be helpful.

This is explained a bit earlier in the text (around Lines 181-187).

Line 236: Does this mean the whole bed is temperate in this experiment since melting is uniform?

Yes, the whole bed is assumed to be temperate.

Fig 6b. Why does the effective pressure go down by 2 MPa in both the soft and hard bed cases at $x = 0$?

This comes from the fact that for low water flux values, the effective pressures in hard-bed and soft-bed systems are similar (see, e.g., Figure 5). Because there is a uniform melt supply, the subglacial water flux increases linearly with $x$. In particular, close to $x = 0$, it is very small so that we are indeed in an inefficient regime in which hard and soft beds leads to similar results.

The value observed at $x = 0$ (around $2\,\mathrm{MPa}$) is a result of the sampling used for the display of the plot, the discretization error in the numerical simulation, and the way $\boldsymbol{v}_\mathrm{b}$ behaves near the origin. Close to $x = 0$, the water flux is rather small, so that the hydrological system is in an inefficient regime, with an effective pressure given by:

$$N = \left( \frac{\|\boldsymbol{v}_\mathrm{b}\| h_\mathrm{b}}{2n^{-n} A S_\infty} \right)^{\frac{1}{n}} \tag{R1a}$$

$$= \left( \frac{\|\boldsymbol{v}_\mathrm{b}\| h_\mathrm{b}}{2n^{-n} A K^{-\frac{1}{\alpha}} \|\nabla \phi_0\|^{\frac{1-\beta}{\alpha}} Q_\mathrm{w}^{\frac{1}{\alpha}}} \right)^{\frac{1}{n}}, \tag{R1b}$$

which is the particularization of equation (7) for an inefficient regime. Here, we have considered hard beds, soft beds being very similar (the only difference is that there is an additional factor $F_\mathrm{till}$, but this does not change the reasoning).

As $x \to 0$, both $Q_\mathrm{w}$ and $\|\boldsymbol{v}_\mathrm{b}\|$ go towards zero because both quantities must vanish at that position (these are prescribed boundary conditions). Therefore, the value taken by $N$ depends on the way both quantities converge towards zero, i.e., how fast they do so. The flux $Q_\mathrm{w}$ depends on linearly with respect to $x$ as the melt rate is uniform. Hence, the value taken by $N$ depends on the way $\|\boldsymbol{v}_\mathrm{b}\|$ goes towards zero. It seems to us that it does so with a rate that is at least linear, which would suggest that $N \to 0$ as $x \to 0$. As $N$ reaches a small but non-zero value, we speculate that this difference comes from either the sampling used to display the plot, or numerical errors in the simulations. Note that the value taken by $N$ at this position is not really important for our simulations as those are focused around grounding-line dynamics.

Fig 7: Are the light pink and red areas in (a)-(c) related to the colors in (d)? If so, it is not clear how. Add something to the caption about this (or remove from the background?).

Yes, they are related. We now mention in the caption that the light pink (resp. dark pink) areas in the background of the sub-figures (b), (c), and (d) correspond to regions in which the efficient/inefficient system is an inefficient (resp. efficient) regime.

Fig. 9a: It would be helpful to have a sense of scale in the 2D either by adding a scale bar or axis.

Indeed, thanks for the suggestion. We have added a scale bar.

Line 267: This does not apply on the timescales you are analyzing. The theory from Schoof (2010) on meltwater variability is referring to time-dependent changes that occur on timescales of hours to days, at most months. In this model, it is assumed to be in steady state which means that these short term increases in water pressure due to varying water flux are inherently neglected.

We have removed this reference to Schoof (2010).

Line 282: What do you mean by "the latter". If referring to a smaller ice sheet results in grounding line retreat, replace "the latter" with "consequently" or similar. Both slower velocities and a smaller ice sheet can result in grounding line retreat.

We were referring to the applied perturbation, i.e., the reduction in surface accumulation. To make it more clear, we have therefore replaced this sentence by the following: "*This reduction also results in a slight grounding-line retreat (NON in Figure 7).*".

Line 318: This is not a unit of mass, but a unit of length. Rephrase to say something like, "Note that the mass loss for the NON experiment results in sea level rise on the order of 10 mm by 2100..."

Corrected.

Line 341: All of the models result in the collapse of Thwaites within how many years?

This depends on the subglacial hydrology considered but all models lead to a collapse before 2500. These collapses are displayed in the supplementary videos of our manuscript.

Line 392: As mentioned before, I have questions about how the assumption of steady state allows you to relate to the time variability of other work such as in the comment that follows. "This observation aligns with the work of Iken (1981), specifying that the highest velocities is not observed where effective pressures is lowest, but rather when cavities enlarge due to an increase in subglacial water pressure." I think this statement should be removed.

As mentioned in our response to your general comment, it seems to us that this comparison can be included as our model allows cavities to evolve over time, the hydrological system being in a quasi-static equilibrium with respect to the ice-sheet configuration which itself evolve over time.

Line 396: How does a lower effective pressure in the soft bed system slow down grounding line retreat? Maybe I am missing something.

When the grounding line retreats in the region where the drainage system is inefficient, it tends to slow down as the region near the grounding line experiences an effective pressure that is larger to the one it was previously experiencing. This is because of the form of the relation between the effective pressure and the water flux in an inefficient system: $\partial N/\partial Q < 0$ for such a system (see Figure 5).

Because the effective pressure in an inefficient system for a soft bed is slightly inferior to that of a hard bed (thanks to the parameter $F_{\text{till}} > 1$), this increase in effective pressure experienced by the region close to the grounding line is reduced. As a consequence, this slows down the

grounding-line retreat.

Line 434: add "(retrograde)".

Done.

Line 440: "(prograde)".

Done.

Line 465: add 'considering' or 'modeling' so that it reads "considering subglacial hydrology enhances the ice-sheet response to sliding".

Corrected; we now write *"considering subglacial hydrology enhances the ice-sheet response to sliding"*.

Line 467: This reference to Schoof (2010) is appropriate.

Ok.

Line 470: Making the connection to changes when the system is explicitly not in steady state ("when basal cavities are growing") does not make sense here since you assume steady state and therefore neglect time-dependent changes in cavity growth.

Cavities are allowed to grow in our hydrological model. As mentioned in our response to the Referee's general comment, we have added an additional subsection in the Model section, called 'Simplifying assumptions', that aims at describing the assumptions more explicitly, and prior to the actual model description. In it, we emphasize that, because we assume that it is assumed that there is limited melt variability, we consider that the water flow is in a quasi-static equilibrium with respect to the ice-sheet geometry. In other words, the hydrological system is assumed to automatically adapt to each change in the ice-sheet geometry, reaching the corresponding equilibrium position.

We hope that this clarifies the assumptions made in our model, and the our reference to evolving cavities.

**References**

Bueler, E. and van Pelt, W. (2015). Mass-conserving subglacial hydrology in the Parallel Ice Sheet Model version 0.6. *Geoscientific Model Development*, 8(6):1613–1635.

Drews, R., Pattyn, F., Hewitt, I. J., Ng, F. S. L., Berger, S., Matsuoka, K., Helm, V., Bergeot, N., Favier, L., and Neckel, N. (2017). Actively evolving subglacial conduits and eskers initiate ice shelf channels at an antarctic grounding line. *Nature Communications*, 8(1).

Hager, A. O., Hoffman, M. J., Price, S. F., and Schroeder, D. M. (2022). Persistent, extensive channelized drainage modeled beneath Thwaites Glacier, West Antarctica. *The Cryosphere*, 16(9):3575–3599.

Iken, A. (1981). The effect of the subglacial water pressure on the sliding velocity of a glacier in an idealized numerical model. *J. Glaciol.*, 27(97):407–421.

Kazmierczak, E., Sun, S., Coulon, V., and Pattyn, F. (2022). Subglacial hydrology modulates basal sliding response of the antarctic ice sheet to climate forcing. *The Cryosphere*, 16(10):4537–4552.

Le Brocq, A., Payne, A., Siegert, M., and Alley, R. (2009). A subglacial water-flow model for West Antarctica. *Journal of Glaciology*, 55(193):879–888.

Pattyn, F. (2010). Antarctic subglacial conditions inferred from a hybrid ice sheet/ice stream model. *Earth and Planetary Science Letters*, 295(3–4):451–461.

Pattyn, F. (2017). Sea-level response to melting of Antarctic ice shelves on multi-centennial timescales with the fast Elementary Thermomechanical Ice Sheet model (f.ETISh v1.0). *The Cryosphere*, 11(4):1851–1878.

Schoof, C. (2010). Ice-sheet acceleration driven by melt supply variability. *Nature*, 468(7325):803–806.

---

## Referee Report (RR1)

Second review of "A fast and simplified subglacial hydrological model for the Antarctic ice sheet and outlet glaciers"
By Kazmierczak et al.
Submitted to *The Cryosphere*

Overview

The authors of this manuscript have made substantial improvements addressing concerns and questions raised in the first round of reviews. They present a method for simulating subglacial hydrology on large scales (both in space and time) that captures several interesting qualities of potential drainage systems that may arise. The model assumes a steady state hydrological configuration and involves switches between different classifications of drainage types: efficient vs. inefficient, opening by sliding vs. opening by melt, close to the grounding line vs. far from the grounding line, hard bed vs. soft bed. While there are limitations, I think this is a creative approach with demonstrated practical use that has great potential for producing helpful advancements in understanding of the relationship between ice dynamics and subglacial hydrology on broad scales.

The revised manuscript is well organized and well written. I have a few relatively minor comments and questions, listed by line number below, along with a few typographical errors pointed out. With a bit of further refinement, this paper will make a nice contribution to the glaciological literature.

Specific Comments
Title: I like the new title, but suggest "Antarctic Ice Sheet" to follow convention (capitalized when referring to a specific named ice sheet)

Line 41: extra "a" before "small"

Line 42: Some mountain glaciers can surge remarkably fast, advancing significantly in sub-year time scales, and Antarctic ice shelves can break apart rapidly. I suggest adding a qualifier here: "... several orders of magnitude smaller than the typical response time of glaciers and ice sheets…"

Line 65: ice sheet–ice shelf model (replace "-" with "–" for clarity, otherwise this reads as sheet-ice)

Lines 91-92: This is a vague description of efficient criteria – perhaps a more quantitative statement would be better here. What qualifies as a large flux? Or you could describe physical characteristics of an efficient drainage system.

Line 122: This assumption may only be valid if there isn't much water generated. If you have a substantial amount of subglacial water, this won't necessarily be accurate.

Line 123: Does "sub-grid" here refer to your global or local grid scale?

Lines 149-150: Under what conditions was the dissipative melt negligible relative to the other melt terms? This can be an important source of basal meltwater depending on geometry etc., and in fact is the melt term usually responsible for triggering the channelization process. So, I don't think the general dismissal of this term is justified by this statement – and you may be missing an important piece in the hydrology physics if you ignore heat from dissipation. This deserves more acknowledgement.

Lines 166-167: It seems inconsistent to assume fully turbulent flow with large fluxes as described here, but neglect heat from dissipation in Eq. (3). This limitation should at least be mentioned and justified.

Lines 219-220: Another interpretation of boundary conditions at the grounding line applies when it is actually grounded ice. Rather than N=0 as is appropriate for floating ice, in the grounded case the subglacial water pressure could be equal to the pressure of the overlying water where it is discharged. This may be a small difference, but should ideally be tested or at least mentioned here.

Lines 260-261: Melt from dissipation was indicated to be neglected above (text following Eq. 3), so how does large flux lead to higher melt in your model? This is not clear.

Line 262: Clarify that this is only because of the steady-state condition that ice creep must increase to counter increased melt with increased flux. In a dynamic system that is not at steady state, these opening and closing rates are not necessarily in balance.

Line 270: I suggest rewriting slightly to clarify that these are modifications that may be imposed to force the model to produce an entirely efficient or entirely inefficient system. As it is currently written, it can lead to confusion about what is included in the model. Suggested rephrasing: "By removing the opening term associated with the sliding over obstacles, ∥ vb ∥ hb, from equations (5b) and (6a), it is possible to force the model to produce an entirely efficient drainage system. In this case, we also set $Q_c = \infty$, which guarantees that the conduit geometry is the one of an inefficient system for soft beds. Similarly, to force an entirely inefficient system, the efficient component, $Q_w$ ∥ ∇$\phi$ ∥ /$\rho_i L_w$, can be removed from (5b), together with the condition that $Q_c = 0$."

Lines 281-282: Why use a different sliding law with the hydrology simulations (regularized Coulomb) than what was used for generating the reference state (Weertman)? I imagine you could justify this because the effective pressure varies spatially in the hydrology simulations, while in the reference state spin-up you likely assumed some spatially uniform fraction of overburden or something. Please explain.

Lines 301-303: Suggested rephrasing for enhanced clarity: "We also compare the impact of the drainage efficiency, by comparing the cases where only efficient (eff) or inefficient (ineff) systems are allowed to develop. Note that, by default, the switch between both systems (efficient/inefficient) is determined based on the subglacial water flux magnitude."

Line 309: "correspond", not "corresponds"

Line 518: "when basal cavities are growing" should be "*where* basal cavities are growing" (your model is considering steady-state configurations, so this should be a spatial description, not temporal)

---

## Referee Report (RR2)

**Review of A fast and simplified subglacial hydrological model for the Antarctic ice sheet and outlet glaciers, Kazmierczak et al.**

Thanks to the authors for their work restructuring and clarifying the paper, which is a very nice and interesting read. Most of my comments below are quite minor. However, I am still a little unhappy that there are two (rather different) assumptions lumped under 'key point 3' in their new section 2.2.1, where the only justification provided in the text is that the assumption 'follows from [their] modelling approach', which seems circular to me.

I do appreciate that for practical purposes one must prescribe a single value of $N$ for a given grid cell, to then use in the sliding law. My point was that it is usually the case (per high resolution models) that the effective pressure in a channel [as calculated with something like (5b)] is somewhat higher compared to the average effective pressure in the area surrounding it [which would be more like the $N$ controlling sliding for the whole grid cell]. Admittedly, this effect is likely small, but it would be better to acknowledge and dismiss than to ignore.

The assumption that the drainage density is uniform is rather separate. I appreciate the sensitivity tests showing that using different uniform values results in only limited sensitivity of the model, but I was questioning more about whether $l_c$ should change with $q$, and therefore would feed more impactfully into the way that $N$ changed with flux. The authors include quite a detailed description around l.235 about the change in morphology of the patchy film with $H$, which is exactly why I question that $l_c$, effectively the distance between separating clasts in this regime, is not also somehow a function of film thickness. Similarly one might consider that the distance between linked cavities is likely quite a bit smaller than the distance between subglacial channels (recorded by eskers). I would really appreciate if the authors spent more time with this assumption, either in section 2.2.1, around equation (4), or when discussing film geometry ($\sim$l. 235). If nothing else, there is a lot of great discussion in this paper highlighting the need for future investigation, so it would be helpful to highlight that future work could be performed here.

Specific comments

- Abstract: it could be nice to focus more on the fast aspect of the model.

- l.2 Perhaps just 'switch' rather than 'dynamic switch' since the hydrology is quasi-steady?

- l.9 Comma missing after 'itself'

- l.17 What is meant by 'plasticity' here? In the sense of variability, not in the sense of sediment having a plastic rheology? Perhaps rephrase.

- l.41 There are no processes listed in the previous sentence? Unless by distribution you mean the flow, rather than the final spatial distribution? Also, please add an example reference for this sentence.

- l.43 A[nother] limiting factor, rather?

- l.50 'allows [us?] to dynamically link' word missing.

- l.51 Perhaps state that the hydrology is quasi-static but temporally varying.

- l.92 Inefficient hydrological systems can still transport large fluxes of water if they need to, they just induce larger pressure gradients to do so. Maybe add 'with low gradients in $\phi$' to the end of the sentence.

- l.123 As noted above, split this point into two parts. Also, drainage density has not yet been defined in the paper at this point.

- l.131 You have your own synthetic and real geometries to estimate $\nabla\phi_0$ from, so it could be good to use those values.

- l.159 Suggest for clarity 'we choose not to do so as this allows us to decouple the water routing solver from the effective pressure calculation.'

- l.197 Not sure the sentence beginning 'Note that' belongs here.

- l.200 Clarify - drainage rate from where to what aquifer?

- l.212 Can commit to this analysis having been discussed already in 2.2.1 and skip straight to using $\nabla\phi_0$.

- l.222 Weaken to 'and we suggest the effective pressure is approximated over the whole domain by', since the functional form does not formally come from the boundary layer analysis.

- l.306 How are the fluctuations in meltwater forced, since there is (as I understand it) no direct meltwater input in the model?

- l.448 is split $\rightarrow$ can be split

- l.460 Confusingly worded - seems more like the second sentence is restating the first, rather than being implied by it.

- l.464 I don't understand how low $\tilde{\tau}_b$ causes a low $N$ - via ice sheet loss? Or is the implication the other way round and just via (12b)? Please clarify logical flow with a more explicit description of the mechanism.

- l.465 that $\rightarrow$ which

- l.481 Probably should use something less strong than 'On the contrary', such as 'However' or even 'Similarly' since the logic is the same. Then have a paragraph break or contrasting start to the next sentence to introduce the new logic of a spatially variable $C$.

---

## Author Response (AR3)

**Response to the Editor on "A fast and simplified subglacial hydrological model for the Antarctic ice sheet and outlet glaciers" by Kazmierczak, Gregov, Coulon & Pattyn (Rev. 2).**

Dear Dr. Joe MacGregor,

Once again, we would like to thank you and the reviewers for your comments, which have helped us to improve our article, as well as your handling of our paper.

The detailed description of the changes made to the manuscript can be found in the marked-up version of the revised manuscript, as well as in our responses to the Referees.

Best regards,

On behalf of the authors,
Thomas Gregov

**Response to Referee 1 on "A fast and simplified subglacial hydrological model for the Antarctic ice sheet and outlet glaciers" by Kazmierczak, Gregov, Coulon & Pattyn (Rev. 2).**

Dear Referee,

We would like to thank you for your review, and in particular for explaining in further details your questions with respect to the model assumptions. This has enabled us to clarify these, and should also allow us to ponder on the limitations of our model, thus paving the way for further interesting research associated with the development of simplified hydrological models.

You will find below, in blue, our responses to your comments.

Best regards,

On behalf of the authors,
Thomas Gregov

**Response to the Referee's comments**

General comment:

Thanks to the authors for their work restructuring and clarifying the paper, which is a very nice and interesting read. Most of my comments below are quite minor. However, I am still a little unhappy that there are two (rather different) assumptions lumped under 'key point 3' in their new section 2.2.1, where the only justification provided in the text is that the assumption 'follows from [their] modelling approach', which seems circular to me.

Thanks for your positive comment and your critical assessment of the hypotheses of the model. We have now separated the third assumption into two separate items:

3. *The drainage density, that is, the number of conduits per grid cell, is uniform.*

4. *The effective-pressure distribution is not calculated at the sub-grid (local) level.*

The other proposed modifications are described below the following paragraphs.

I do appreciate that for practical purposes one must prescribe a single value of $N$ for a given grid cell, to then use in the sliding law. My point was that it is usually the case (per high resolution models) that the effective pressure in a channel [as calculated with something like (5b)] is somewhat higher compared to the average effective pressure in the area surrounding it [which would be more like the $N$ controlling sliding for the whole grid cell]. Admittedly, this effect is likely small, but it would be better to acknowledge and dismiss than to ignore.

We agree with the Referee's remark. We have added the following sentence after the assumptions to acknowledge this effect:

"(...) *However, the effective pressure within a channel may well differ from its value away from the channel, which is something that is not taken into account. Consequently, these last assumptions are the most likely to be debatable.*"

Note that we are very much interested in tackling this issue of attributing a single effective-pressure value in large-scale ice-sheet codes despite the local heterogeneity in the subglacial hydrology, and this something that we wish to work on in the future in our research group.

The assumption that the drainage density is uniform is rather separate. I appreciate the sensitivity tests showing that using different uniform values results in only limited sensitivity of the model, but I was questioning more about whether $l_c$ should change with $\boldsymbol{q}_w$, and therefore

would feed more impactfully into the way that $N$ changed with flux. The authors include quite a detailed description around l. 235 about the change in morphology of the patchy film with $H$, which is exactly why I question that $l_c$, effectively the distance between separating clasts in this regime, is not also somehow a function of film thickness. Similarly one might consider that the distance between linked cavities is likely quite a bit smaller than the distance between subglacial channels (recorded by eskers). I would really appreciate if the authors spent more time with this assumption, either in section 2.2.1, around equation (4), or when discussing film geometry ($\sim$l. 235). If nothing else, there is a lot of great discussion in this paper highlighting the need for future investigation, so it would be helpful to highlight that future work could be performed here.

We also agree with this remark of the Referee. Following their suggestion, we have made several changes to the manuscript. Firstly, as mentioned above, a sentence has been added after the assumptions to highlight that the third and fourth assumptions are the ones that are the most tricky ones. Secondly, the following remark has been added after equation (4):

"*However, we acknowledge that this distance is likely to be a function of the drainage system, but leave this to be investigated in future work.*"

Finally, we have modified the model limitations subsection in the Discussion section to include this limitation into account. The beginning of the paragraph has been modified as follows:

"*(...) Our subglacial hydrology models do not include variations of drainage density or of effective pressure below the resolution of the ice-sheet discretization.*"

Specific comments:

Abstract: it could be nice to focus more on the fast aspect of the model.

We have added the following sentence to the abstract:

"*It does not explicitly simulate the details of water conduits at the local scale and assumes that subglacial hydrology is in quasi-static equilibrium with the ice sheet, which makes the computations very fast.*"

Together with the changed title, this should emphasize the fast aspect of the model.

Line 2: Perhaps just 'switch' rather than 'dynamic switch' since the hydrology is quasi-steady?

We have replaced 'dynamic' by 'automatic'. This suggests that the switch is made by the model itself, which is an important feature of the model.

Line 9: Comma missing after 'itself'.

Corrected.

Line 17: What is meant by 'plasticity' here? In the sense of variability, not in the sense of sediment having a plastic rheology? Perhaps rephrase.

In the sense of variability, but this variability (or lack thereof) could be explained by a plastic rheology of the bed, as mentioned directly after: 'which mainly depends on the bed rheology'.

Line 41: There are no processes listed in the previous sentence? Unless by distribution you mean the flow, rather than the final spatial distribution? Also, please add an example reference for this sentence.

Indeed. We have modified this sentence to the following: "*Furthermore, subglacial processes occur on time scales that can be as small as a few hours (e.g., Clarke, 2005)*".

Line 43: A[nother] limiting factor, rather?

Corrected.

Line 50: 'allows [us?] to dynamically link' word missing.

Corrected.

Line 51: Perhaps state that the hydrology is quasi-static but temporally varying.

Thanks for the suggestion. However, the details of the model follow in the next section. Here we just have a very broad statement on the temporal and spatial variability.

Line 92: Inefficient hydrological systems can still transport large fluxes of water if they need to, they just induce larger pressure gradients to do so. Maybe add 'with low gradients in $\phi$' to the end of the sentence.

We agree with the Referee that inefficient hydrological systems can also, in principle, transport large fluxes of water. However, inefficient systems are thought to be unstable when the input of water gets above a threshold. Therefore, it seems to us that it is fair to say that we expect efficient systems to be those that transport large fluxes of water in channels, while inefficient systems are characterized by a distributed transport.

To improve the clarity of the efficient/inefficient distinction, we have modified this sentence to the following:

*"Generally, efficient systems transport large water fluxes and are characterized by localized channelized flow, while inefficient systems take the form of distributed water flow."*

Line 123: As noted above, split this point into two parts. Also, drainage density has not yet been defined in the paper at this point.

Corrected.

Line 131: You have your own synthetic and real geometries to estimate $\nabla\phi_0$ from, so it could be good to use those values.

This is an interesting suggestion that we had also considered. However, it seemed to us that such a justification could take the form of a circular argument, since we would be relying on the results of our simulations to justify the assumptions of the model. So we preferred to rely on other pre-existing references for estimates.

Line 159: Suggest for clarity 'we choose not to do so as this allows us to decouple the water routing solver from the effective pressure calculation'.

Thanks for the suggestion, which we have followed.

Line 197: Not sure the sentence beginning 'Note that' belongs here.

We have removed the *"Note that"* which was not necessary here.

Line 200: Clarify - drainage rate from where to what aquifer?

This sentence has been modified to the following: *"Indeed, as the drainage rate from the till towards a subglacial aquifer is much smaller than the basal melt rate, (...)"*.

Line 212: Can commit to this analysis having been discussed already in 2.2.1 and skip straight to using $\nabla\phi_0$.

Corrected.

Line 222: Weaken to 'and we suggest the effective pressure is approximated over the whole domain by', since the functional form does not formally come from the boundary layer analysis.

Corrected.

Line 306: How are the fluctuations in meltwater forced, since there is (as I understand it) no direct meltwater input in the model?

It is true that there is no direct surface or englacial meltwater input to the model. Here, the subglacial meltwater oscillates thanks to variations in the meltrate $\dot{m}$ that is artificially set to a sinusoidal function (see figure 6a).

Line 448: is split $\rightarrow$ can be split.

Corrected.

Line 460: Confusingly worded - seems more like the second sentence is restating the first, rather than being implied by it.

Indeed. We have removed the implication: the sentence is now simply given by "*The zone of low effective pressure migrates with a migrating grounding line.*".

Line 464: I don't understand how low $\tilde{\boldsymbol{\tau}}_{\mathrm{b}}$ causes a low $N$ - via ice sheet loss? Or is the implication the other way round and just via (12b)? Please clarify logical flow with a more explicit description of the mechanism.

It seems that there was an error in this sentence, and that the explanation was not very clear overall. We have rewritten it as follows:

"*The zone of low effective pressure migrates with a migrating grounding line. Such migration obviously does not take place when the subglacial hydrological field is kept constant or when subglacial hydrology is not linked to basal sliding (or not considered; NON). For a retreating grounding line, such linkage actually amplifies grounding-line retreat, as the friction stress close to the grounding line is also reduced following this retreat, leading to a positive feedback mechanism. This reduction in $\boldsymbol{\tau}_{\mathrm{b}}$ stems from a reduction of $\tilde{\boldsymbol{\tau}}_{\mathrm{b}}$, but most importantly from a large value in the magnitude of $\Delta\tilde{\boldsymbol{\tau}}_{\mathrm{b}}$, which is typically negative.*"

We hope this clarifies the instability mechanism.

Line 465: that $\rightarrow$ which.

Corrected.

Line 481: Probably should use something less strong than 'On the contrary', such as 'However' or even 'Similarly' since the logic is the same. Then have a paragraph break or contrasting start to the next sentence to introduce the new logic of a spatially variable $C$.

Thanks for the suggestion; we have replaced "*On the contrary*" by "*However*".

**References**

Clarke, G. K. (2005). Subglacial Processes. *Annual Review of Earth and Planetary Sciences*, 33(1):247–276.

**Response to Referee 2 on "A fast and simplified subglacial hydrological model for the Antarctic ice sheet and outlet glaciers" by Kazmierczak, Gregov, Coulon & Pattyn (Rev. 2).**

Dear Referee,

We would like to thank you again for your detailed review and your numerous remarks.

You will find below, in blue, our responses to your comments.

Best regards,

On behalf of the authors,
Thomas Gregov

**Response to the Referee's comments**

Overview:

The authors of this manuscript have made substantial improvements addressing concerns and questions raised in the first round of reviews. They present a method for simulating subglacial hydrology on large scales (both in space and time) that captures several interesting qualities of potential drainage systems that may arise. The model assumes a steady state hydrological configuration and involves switches between different classifications of drainage types: efficient vs. inefficient, opening by sliding vs. opening by melt, close to the grounding line vs. far from the grounding line, hard bed vs. soft bed. While there are limitations, I think this is a creative approach with demonstrated practical use that has great potential for producing helpful advancements in understanding of the relationship between ice dynamics and subglacial hydrology on broad scales.

The revised manuscript is well organized and well written. I have a few relatively minor comments and questions, listed by line number below, along with a few typographical errors pointed out. With a bit of further refinement, this paper will make a nice contribution to the glaciological literature.

We would like to thank you for your positive assessment of our paper.

Specific Comments:

Title: I like the new title, but suggest "Antarctic Ice Sheet" to follow convention (capitalized when referring to a specific named ice sheet).

Corrected.

Line 41: extra "a" before "small".

Corrected.

Line 42: Some mountain glaciers can surge remarkably fast, advancing significantly in sub-year time scales, and Antarctic ice shelves can break apart rapidly. I suggest adding a qualifier here: "... several orders of magnitude smaller than the typical response time of glaciers and ice sheets..."

Corrected.

Line 65: ice sheet-ice shelf model (replace "-" with "–" for clarity, otherwise this reads as sheet-ice)

To avoid any confusion, we have replaced 'ice sheet-ice shelf' by 'ice sheet/ice shelf'.

Lines 91-92: This is a vague description of efficient criteria – perhaps a more quantitative statement would be better here. What qualifies as a large flux? Or you could describe physical characteristics of an efficient drainage system.

Following your advice, we have modified this sentence to the following: *"Generally, efficient systems transport large water fluxes and are characterized by localized channelized flow, while inefficient systems take the form of distributed water flow."*.

Line 122: This assumption may only be valid if there isn't much water generated. If you have a substantial amount of subglacial water, this won't necessarily be accurate.

Indeed. However, as mentioned later in this subsection, the meltwater supply in Antarctica (which is our primary application) is limited to subglacial meltwater.

Line 123: Does "sub-grid" here refer to your global or local grid scale?

This sentence was indeed ambiguous. We have replaced it by the following: *"The effective-pressure distribution is not calculated at the sub-grid (local) level."*.

Lines 149-150: Under what conditions was the dissipative melt negligible relative to the other melt terms? This can be an important source of basal meltwater depending on geometry etc., and in fact is the melt term usually responsible for triggering the channelization process. So, I don't think the general dismissal of this term is justified by this statement – and you may be missing an important piece in the hydrology physics if you ignore heat from dissipation. This deserves more acknowledgement.

We agree with the Referee that dissipative melt is an essential component in subglacial hydrology as it is a key element for channelization (e.g., Warburton et al., 2024). Dissipative melt of conduits impacts out model in two ways:

(i) In the subglacial water routing model, i.e., in the mass-balance equation (2).

(ii) In the equation for the effective pressure, i.e., in the balance of opening-closing balance equation (5).

In our model, the subglacial water flux is integrated across the entire grid area and distributed to a discrete number of conduits. Equation (4) describes how $q_{\mathrm{w}}$ is converted to $Q_{\mathrm{w}}$. Based on this approach, the location where the water is generated does not influence the calculation of $Q_{\mathrm{w}}$. Since we have determined that $\dot{m}_{\mathrm{w}}$ is relatively small compared to $\dot{m}$, it can be safely ignored in this calculation.

However, the opening due to melt must be included in the opening-closing balance equation (5), as it is the dominant process for channels. Accordingly, we have kept the dissipative melt in that equation.

To clarify a bit that the dissipative melt is ignored only in equation (3), we have modified lines 149-150 to the following:

*"However, we do not include this last term in the computation of the subglacial water in our simulations as it was found to be negligible compared to the other terms."*

Lines 176-177: It seems inconsistent to assume fully turbulent flow with large fluxes as described here, but neglect heat from dissipation in Eq. (3). This limitation should at least be mentioned and justified.

See our response to the previous comment.

Lines 219-220: Another interpretation of boundary conditions at the grounding line applies when it is actually grounded ice. Rather than $N = 0$ as is appropriate for floating ice, in the grounded case the subglacial water pressure could be equal to the pressure of the overlying water where it is discharged. This may be a small difference, but should ideally be tested or at least mentioned here.

We agree with the Referee that this boundary condition should also be considered. However, given that our paper specifically targets marine-terminated ice sheets we decided to keep only the 'floating ice' boundary condition. We have modified the description of the equation $N = 0$ as follows:

*"Finally, the third equation comes from the equality between the subglacial water pressure and the sea-water pressure at the grounding line (Drews et al., 2017), which holds because we are considering marine-terminated ice sheets"*.

It would be interesting to see how our model would cope with this change in the boundary condition. It seems that in this case a boundary layer still exists (see e.g. Fowler, 2011, end of page 662), so it is likely that our model could be adapted for such a case. We leave this for possible future work.

Lines 260-261: Melt from dissipation was indicated to be neglected above (text following Eq. 3), so how does large flux lead to higher melt in your model? This is not clear.

As mentioned previously, while melt from dissipation is ignored for the computation of the subglacial water, it is not ignored in the opening-closing balance equation (equation (5)). We hope this clarifies your question.

Line 262: Clarify that this is only because of the steady-state condition that ice creep must increase to counter increased melt with increased flux. In a dynamic system that is not at steady state, these opening and closing rates are not necessarily in balance.

Indeed. In the introduction to the system of equations (5) we now explicitly state that these equations are associated with a quasi-static regime. Since the explanation in line 262 is related to equation (5), this should now be more clear.

Line 270: I suggest rewriting slightly to clarify that these are modifications that may be imposed to force the model to produce an entirely efficient or entirely inefficient system. As it is currently written, it can lead to confusion about what is included in the model. Suggested rephrasing: "By removing the opening term associated with the sliding over obstacles, $\|\boldsymbol{u}_\mathrm{b}\|h_\mathrm{b}$, from equations (5b) and (6a), it is possible to force the model to produce an entirely efficient drainage system. In this case, we also set $Q_\mathrm{c} = \infty$, which guarantees that the conduit geometry is the one of an inefficient system for soft beds. Similarly, to force an entirely inefficient system, the efficient component, $Q_\mathrm{w}\|\nabla\phi\|/\rho_\mathrm{i}\mathcal{L}_\mathrm{w}$, can be removed from (5b), together with the condition that $Q_\mathrm{c} = 0$".

Thanks for the suggestion, which we have followed.

Lines 281-282: Why use a different sliding law with the hydrology simulations (regularized Coulomb) than what was used for generating the reference state (Weertman)? I imagine you could justify this because the effective pressure varies spatially in the hydrology simulations, while in the reference state spin-up you likely assumed some spatially uniform fraction of overburden or something. Please explain.

We used a regularized Coulomb siding law in our simulations because it allows for both a viscous and a plastic behavior as a function of the sliding velocity (see equation (1)). However, to generate the initial state, we used a Weertman sliding law. For the idealized experiments, this is motivated by the fact that we rely on the MISMIP set-up which considers a Weertman sliding law to get the initial state. We have slightly modified our manuscript to make this 'choice' more clear; the lines 281-282 have been modified to the following:

*"In our experiments, we use a regularized Coulomb friction law combined with hydrological models, while the reference state from the MISMIP set-up has been obtained with a Weertman friction law."*

For the Thwaites experiments, we also used a Weertman sliding law. Here, we chose this law for efficiency of the spin-up procedure, as it is also run without subglacial hydrology. Nevertheless, this has no impact on the results of the simulations that follow, given our scaling approach that is explained in the manuscript.

Lines 301-303: Suggested rephrasing for enhanced clarity: "We also compare the impact of the drainage efficiency, by comparing the cases where only efficient (eff) or inefficient (ineff) systems are allowed to develop. Note that, by default, the switch between both systems (efficient/inefficient) is determined based on the subglacial water flux magnitude".

Thanks for the suggestion, which we have followed.

Line 309: "correspond", not "corresponds".

Corrected.

Line 518: "when basal cavities are growing" should be "where basal cavities are growing" (your model is considering steady-state configurations, so this should be a spatial description, not temporal).

Corrected.

**References**

Drews, R., Pattyn, F., Hewitt, I. J., Ng, F. S. L., Berger, S., Matsuoka, K., Helm, V., Bergeot, N., Favier, L., and Neckel, N. (2017). Actively evolving subglacial conduits and eskers initiate ice shelf channels at an antarctic grounding line. *Nature Communications*, 8(1).

Fowler, A. (2011). *Mathematical Geoscience*. Springer London.

Warburton, K., Meyer, C., and Sommers, A. (2024). Numerical and physical instability of subglacial water flow. *EarthArXiv preprint*.